# Statistical inference for Linear Stochastic Approximation with Markovian Noise

**Sergey Samsonov**
HSE University
svsamsonov@hse.ru

**Marina Sheshukova**
HSE University
msheshukova@hse.ru

**Eric Moulines**
Ecole Polytechnique,
MBZUAI
eric.moulines@polytechnique.edu

**Alexey Naumov**
HSE University,
Steklov Mathematical Institute
of Russian Academy of Sciences
anaumov@hse.ru

## Abstract

In this paper we derive non-asymptotic Berry-Esseen bounds for Polyak-Ruppert averaged iterates of the Linear Stochastic Approximation (LSA) algorithm driven by the Markovian noise. Our analysis yields $\mathcal{O}(n^{-1/4})$ convergence rates to the Gaussian limit in the Kolmogorov distance. We further establish the non-asymptotic validity of a multiplier block bootstrap procedure for constructing the confidence intervals, guaranteeing consistent inference under Markovian sampling. Our work provides the first non-asymptotic guarantees on the rate of convergence of bootstrap-based confidence intervals for stochastic approximation with Markov noise. Moreover, we recover the classical rate of order $\mathcal{O}(n^{-1/8})$ up to logarithmic factors for estimating the asymptotic variance of the iterates of the LSA algorithm.

## 1 Introduction

Stochastic approximation (SA) has become foundational to modern machine learning, especially its reinforcement learning (RL) domain. Many classical RL algorithms, including $Q$-learning [80, 76], the actor–critic algorithm [35], and policy evaluation algorithms, such as temporal difference (TD) learning [76] are special instances of SA. Recent research has extensively studied both the asymptotic [14] and non-asymptotic behavior [49] of these algorithms. It is important not only to establish the convergence of SA estimators, but also to quantify their uncertainty, which is typically done through the asymptotic normality of corresponding estimates, see [57, 46]. Recent works have focused on deriving non-asymptotic convergence rates for SA methods [69, 67, 82]. Notably, most existing results consider settings with independent and identically distributed (i.i.d.) noise. In contrast, many practical SA applications involve dependent noise, often forming a Markov chain. This additionally complicates the problem. Indeed, even the problem of deriving precise Berry–Esseen type convergence rates for additive functionals of Markov chains is challenging compared to i.i.d. setting, where quantitative results are well-established, starting from Bentkus' influential work [7].

Asymptotic normality of SA estimates is particularly important in practice, as it allows one to construct approximate confidence intervals for the parameters of interest. Different approaches either directly utilize the asymptotic normality and aim to estimate the asymptotic covariance matrix directly (the plug-in or batch-mean methods, see [17, 15, 63]), or rely on the non-parametric methods, based on the bootstrap [27, 64]. When the latter approach is applied to the dependent observations, standard bootstrap methods need to be carefully adjusted to account for the dependence structure, see [29, 42].

In this paper, we study a linear stochastic approximation (LSA) procedure. This setting covers several important scenarios, such as the classical TD learning algorithm [76] with linear function approximation, while allowing for sharper theoretical analysis. The LSA procedure aims to find an approximate solution to the linear system $\bar{\mathbf{A}}\theta^\star = \bar{\mathbf{b}}$ with a unique solution $\theta^\star$, based on observations $\{(\mathbf{A}(Z_k), \mathbf{b}(Z_k))\}_{k \in \mathbb{N}}$. Here $\mathbf{A} : \mathsf{Z} \to \mathbb{R}^{d \times d}$ and $\mathbf{b} : \mathsf{Z} \to \mathbb{R}^d$ are measurable functions, and $\{Z_k\}_{k \in \mathbb{N}}$ is a sequence of noise variables taking values in a measurable space $(\mathsf{Z}, \mathcal{Z})$ with a distribution $\pi$ satisfying $\mathbb{E}_\pi[\mathbf{A}(Z_k)] = \bar{\mathbf{A}}$ and $\mathbb{E}_\pi[\mathbf{b}(Z_k)] = \bar{\mathbf{b}}$. We focus on the setting where $\{Z_k\}_{k \in \mathbb{N}}$ is an ergodic Markov chain. Given a sequence of decreasing step sizes $\{\alpha_k\}_{k \in \mathbb{N}}$ and an initial point $\theta_0 \in \mathbb{R}^d$, we consider the estimates $\{\bar{\theta}_n\}_{n \in \mathbb{N}}$ given by

$$\theta_k = \theta_{k-1} - \alpha_k\{\mathbf{A}(Z_k)\theta_{k-1} - \mathbf{b}(Z_k)\}, \ k \geq 1, \quad \bar{\theta}_n = n^{-1}\sum_{k=0}^{n-1}\theta_k, \ n \geq 1. \quad (1)$$

The sequence $\{\theta_k\}_{k \in \mathbb{N}}$ corresponds to the standard LSA iterates, while $\{\bar{\theta}_n\}_{n \in \mathbb{N}}$ corresponds to the Polyak-Ruppert (PR) averaged iterates [65, 57]. It is known that, under appropriate technical conditions on the step sizes $\{\alpha_k\}$ and the noisy observations $\{Z_k\}$ (see [57] and [31] for a discussion),

$$\sqrt{n}(\bar{\theta}_n - \theta^\star) \overset{d}{\to} \mathcal{N}(0, \Sigma_\infty). \quad (2)$$

Here the asymptotic covariance matrix $\Sigma_\infty$ is defined in Section 3; see (8). Recent works have provided a number of non-asymptotic guarantees for the averaged LSA iterates of $\bar{\theta}_n$, in particular, [39, 74, 47, 25, 48, 24], which study the mean-squared error, high-order moment bounds, and concentration bounds for $\sqrt{n}(\bar{\theta}_n - \theta^\star)$. However, these results are not enough to establish explicit convergence rates in (2) in some appropriate probability metric d:

$$\mathsf{d}(\sqrt{n}(\bar{\theta}_n - \theta^\star), \Sigma_\infty^{1/2}Y),$$

where $Y \sim \mathcal{N}(0, \mathrm{I})$. Notable exceptions are recent papers [67, 82, 70, 83], which study non-asymptotic convergence rates in the LSA and stochastic gradient descent (SGD) algorithms, as well as in the specific setting of the TD learning algorithm. Our paper aims to complement these findings by providing both a non-asymptotic analysis of the convergence rate in (2) and an analysis of an appropriate procedure for constructing confidence sets for $\theta^\star$. Our primary contributions are as follows:

- We derive a novel non-asymptotic bound for normal approximation for projected Polyak–Ruppert averaged LSA iterates $\sqrt{n}u^\top(\bar{\theta}_n - \theta^\star)$ under Markovian noise in (2). Here $u$ is a vector on the unit sphere $\mathcal{S}_{d-1}$. Precisely, we establish a convergence rate of order $\mathcal{O}(n^{-1/4})$ in Kolmogorov distance. The rate $\mathcal{O}(n^{-1/4})$ matches the recent result of [83], which considered the particular setting of the TD learning algorithm, and improves over the previous results from [73]. Our proof strategy differs from that of [83] and is based on a Poisson decomposition for Markov chains, combined with an appropriate version of the Berry–Esseen bound for martingales, building on the results of Fan [28] and Bolthausen [12].
- We provide a non-asymptotic analysis of the multiplier subsample bootstrap approach [43] for the LSA algorithm under Markovian noise. Our bounds imply that the coverage probabilities of the true parameter $\theta^\star$ can be approximated at a rate of $\mathcal{O}(n^{-1/10})$, where $n$ is the number of samples used in the procedure. To the best of our knowledge, this is the first non-asymptotic bound on the accuracy of bootstrap approximation for SA algorithms with Markov noise. As a byproduct of our analysis, we also recover the rate of $\mathcal{O}(n^{-1/8})$ (up to logarithmic factors) for estimating the asymptotic variance of projected LSA iterates using the overlapping batch means (OBM) estimator, previously obtained in [17, 79].
- We apply the proposed methodology to the temporal difference learning (TD) algorithm for policy evaluation in reinforcement learning.

The rest of the paper is organized as follows. In Section 2, we review the literature on Berry–Esseen type results for stochastic approximation (SA) algorithms and methods for constructing confidence intervals for the parameter $\theta^\star$ in these settings. In Section 3, we obtain a quantitative version of the Berry–Esseen theorem for the projected error of the Polyak–Ruppert averaged LSA under the Kolmogorov distance. In Section 4, we discuss the multiplier subsample bootstrap approach (as proposed in [43]) for LSA and provide non-asymptotic bounds on the accuracy of approximating the exact distribution of $\sqrt{n}u^\top(\bar{\theta}_n - \theta^\star)$ with its bootstrap counterpart. Then in Section 5 we apply the proposed methodology to the temporal difference learning (TD) algorithm, based on the sequence of states, which form a geometrically ergodic Markov chain. Proofs are postponed to appendix.

**Notations.** We denote by $\mathcal{P}(\mathsf{Z})$ the set of probability measures on a measurable space $(\mathsf{Z}, \mathcal{Z})$. For probability measures $\mu$ and $\nu$ on $(\mathsf{Z}, \mathcal{Z})$ we denote by $\mathrm{d}_{\mathrm{tv}}(\mu, \nu)$ the total variation distance between them $\mu$ and $\nu$, that is, $\mathrm{d}_{\mathrm{tv}}(\mu, \nu) = \sup_{C \in \mathcal{Z}} |\mu(C) - \nu(C)|$. For matrix $A \in \mathbb{R}^{d \times d}$ we denote by $\|A\|$ its operator norm. Given a sequence of matrices $\{A_\ell\}_{\ell \in \mathbb{N}}$, $A_\ell \in \mathbb{R}^{d \times d}$, we use the following convention for matrix products: $\prod_{\ell=m}^{k} A_\ell = A_k A_{k-1} \ldots A_m$, where $m \leq k$. For the matrix $Q \in \mathbb{R}^{d \times d}$, which is symmetric and positive-definite, and $x \in \mathbb{R}^d$, we define the corresponding norm $\|x\|_Q = \sqrt{x^\top Q x}$, and define the respective matrix $Q$-norm of the matrix $B \in \mathbb{R}^{d \times d}$ by $\|B\|_Q = \sup_{x \neq 0} \|Bx\|_Q / \|x\|_Q$. For sequences $a_n$ and $b_n$, we write $a_n \lesssim b_n$ if there exist a (numeric) constant $c > 0$ such that $a_n \leq c b_n$. For simplicity of presentation, we state the main results of the paper up to absolute constants. We also use a notation $a_n \lesssim_{\log n} b_n$, if there exist $\beta > 0$ and $c > 0$, such that $a_n \leq c(1 + \log n)^\beta b_n$ for any $n$. Additionally, we use the standard abbreviations "i.i.d. " for "independent and identically distributed" and "w.r.t." for "with respect to". We use index $k$ to present results which hold true for any intermediate iteration, and index $n$ denotes the total number of iterations of the algorithm.

## 2 Related works

The analysis of linear stochastic approximation (LSA) algorithms under Markovian noise has a long history, with classical works establishing almost sure convergence and asymptotic normality under broad conditions [57, 14, 38, 8]. These asymptotic results, however, lack explicit finite-sample error bounds. Recently, non-asymptotic performance analysis of stochastic approximation and gradient methods has gained attention. In the i.i.d. setting of stochastic gradient descent (SGD), finite-time error bounds were provided by [51, 58]. For Markovian LSA contexts, convergence rates were analyzed by [10, 39], focusing on temporal-difference learning, and instance-dependent bounds were derived by [48] and [24]. These studies establish guarantees on mean-square error or high-probability deviations but do not address the distributional approximation of the estimator.

Rates of convergence in CLT are widely studied in probability theory [7], primarily for sums of random variables or univariate martingale difference sequences [55, 12]. Berry-Esseen bounds and Edgeworth expansions for Markov chains under various ergodicity assumptions are also available [13, 11, 9, 34], though typically restricted to the 1-dimensional case. Our analysis leverages recent concentration results on quadratic forms of Markov chains [50], building upon martingale decomposition techniques for $U$-statistics developed in [5].

Recent studies have analyzed convergence rates of SA iterates to their limiting normal distributions. [67] derived an $\mathcal{O}(n^{-1/4})$ convergence rate in (2) in convex distance for TD learning under i.i.d. noise, improved to $\mathcal{O}(n^{-1/3})$ by [82] specifically for TD learning. [3] applied Stein's method to averaged SGD and LSA iterates in smooth Wasserstein distance, again under i.i.d. conditions. Recently, [73] analyzed rates for martingale CLT in 1-st order Wasserstein distance with Markovian samples but achieved slower convergence than our result (see Theorem 1 and discussion after it). Lastly, [1] provided last-iterate bounds for SGD for high-dimensional linear regression, also with i.i.d. noise.

Originally introduced for i.i.d. data [27], the bootstrap method has been adapted to complex settings, including high-dimensional tests [18, 19] and linear regression [72]. For SA methods, an online bootstrap for SGD was proposed in [29], with recent non-asymptotic analysis showing approximation rates up to $\mathcal{O}(n^{-1/2})$ for coverage probabilities in case of strongly convex objectives with independent noise [70]. A similar analysis for linear stochastic approximation (LSA) yields rates of order $\mathcal{O}(n^{-1/4})$ [67]. Extensions to Markovian settings by [59] proved inconsistent, as demonstrated in [42, Proposition 1]. Meanwhile, [42] proposed consistent mini-batch SGD estimators for $\varphi$-mixing noise. They also suggested a consistent procedure for constructing the confidence intervals, but studied only its asymptotic properties using the independent block trick [84]. Lastly, multiplier bootstrap techniques for online non-convex SGD were considered by [85].

Among other approaches for constructing confidence intervals for dependent data, we mention methods based on (asymptotically) pivotal statistics [40, 41]. The authors of [41] considered the Polyak–Ruppert averaged $Q$-learning algorithm under the i.i.d. noise assumption (generative model), while [40] generalized this approach to nonlinear stochastic approximation under Markov noise. Another group of methods are based on estimating the asymptotic covariance matrix $\Sigma_\infty$ appearing in the central limit theorem (2). Among these approaches we mention the plug-in estimator of

[17] and batch-mean estimators, see e.g. [15, 79, 63]. Specifically, [63] treats SGD with Markov noise, contrasted with the independent-noise SGD analyses in [17, 15, 79], and [85] investigates both multiplier bootstrap and batch-mean estimators for nonconvex objectives. These methods yield non-asymptotic error bounds in expectation, $\mathbb{E}[\|\hat{\Sigma}_n - \Sigma_\infty\|]$, but do not study convergence rates in (2). A notable exception is [82], which delivers a non-asymptotic analysis of TD-learning under i.i.d. noise, using plug-in covariance estimates to achieve an $\mathcal{O}(n^{-1/3})$ approximation rate for coverage probabilities of $\theta^\star$.

# 3 Accuracy of Gaussian approximation for LSA with Markov noise

## 3.1 Gaussian approximation for non-linear statistics

We first discuss a general scheme for proving quantitative normal approximation bounds for non-linear statistics. Consider the statistic $T(X_1, \ldots, X_n) \in \mathbb{R}^d$, which can be written as

$$T = W + D \, , \tag{3}$$

where $W$ is a linear statistic of the random variables $X_1, \ldots, X_n$, and $D$ is a nonlinear term, which is typically small. Multidimensional versions of CLT based on the decomposition (3) are well-studied when $X_1, \ldots, X_n$ are i.i.d. random variables [16, 69]. In this case, one can follow the randomized concentration approach [69], which requires only a finite 2-nd moment of the term $D$. To our knowledge, there are no such results readily available for Markov chains or martingales. Moreover, even the Berry-Esseen type result for $d$-dimensional linear statistics of Markov chains or martingale-differences with explicit dependence on $d$ remains an interesting and open question. There is no affirmative result comparable to the one of [7] for i.i.d. random vectors. This is the primary reason why we focus on the setting of 1-dimensional statistics and use the following proposition to obtain the rates of Gaussian approximation for $T$ given in (3):

**Proposition 1.** *Let $\Phi(x), x \in \mathbb{R}$ be the c.d.f. of the standard normal law. Then for any random variables $W$ and $D$ and any $p \geq 1$,*

$$\sup_{x \in \mathbb{R}} |\mathbb{P}(W + D \leq x) - \Phi(x)| \leq \sup_{x \in \mathbb{R}} |\mathbb{P}(W \leq x) - \Phi(x)| + 2\mathbb{E}^{1/(p+1)}[|D|^p] \, .$$

This result is classical and can be found e.g. in [71, 16]. For completeness, we provide the proof in Appendix F. The most involved part of applications of Proposition 1 is the proper bound on $\mathbb{E}[|D|^p]$. This requires to select large values of $p$, which later requires to introduce additional assumptions on the step size choice in the LSA procedure, see A3 in Section 3.2.

## 3.2 Gaussian approximation for LSA with Markov noise

When there is no risk of ambiguity, we use the simplified notations $\mathbf{A}_n = \mathbf{A}(Z_n)$ and $\mathbf{b}_n = \mathbf{b}(Z_n)$. Starting from the definition (1), we get with elementary transformations that

$$\theta_n - \theta^\star = (\mathrm{I} - \alpha_n \mathbf{A}_n)(\theta_{n-1} - \theta^\star) - \alpha_n \varepsilon_n \, , \tag{4}$$

where we have set $\varepsilon_n = \varepsilon(Z_n)$ with $\varepsilon(z) = \tilde{\mathbf{A}}(z)\theta^\star - \tilde{\mathbf{b}}(z)$, $\tilde{\mathbf{A}}(z) = \mathbf{A}(z) - \bar{\mathbf{A}}$, $\tilde{\mathbf{b}}(z) = \mathbf{b}(z) - \bar{\mathbf{b}}$. Here the random variable $\varepsilon(Z_n)$ can be viewed as a noise, measured at the optimal point $\theta^\star$. We also define the noise covariance matrix under the stationary distribution $\pi$:

$$\Sigma_\varepsilon = \mathbb{E}_\pi[\varepsilon(Z_0)\{\varepsilon(Z_0)\}^\top] + 2\sum_{\ell=1}^\infty \mathbb{E}_\pi[\varepsilon(Z_0)\{\varepsilon(Z_\ell)\}^\top] \, . \tag{5}$$

We now impose the following conditions:

**A1.** *The sequence $(Z_k)_{k \in \mathbb{N}}$ is a Markov chain taking values in a Polish space $(\mathsf{Z}, \mathcal{Z})$ with Markov kernel $\mathrm{P}$. Moreover, $\mathrm{P}$ admits $\pi$ as a unique invariant distribution and is uniformly geometrically ergodic, that is, there exists $t_{\mathrm{mix}} \in \mathbb{N}$, such that for any $k \in \mathbb{N}$, it holds that*

$$\Delta(\mathrm{P}^k) := \sup_{z,z' \in \mathsf{Z}} \mathsf{d}_{\mathrm{tv}}(\mathrm{P}^k(z, \cdot), \mathrm{P}^k(z', \cdot)) \leq (1/4)^{\lceil k/t_{\mathrm{mix}} \rceil} \, . \tag{6}$$

Parameter $t_{\mathrm{mix}}$ in (6) is referred to as *mixing time*, see [54]. We also impose the following assumptions on the noise variables $\varepsilon(z)$:

**A 2.** $\int_Z \mathbf{A}(z)\mathrm{d}\pi(z) = \bar{\mathbf{A}}$ *and* $\int_Z \mathbf{b}(z)\mathrm{d}\pi(z) = \bar{\mathbf{b}}$, *with the matrix* $-\bar{\mathbf{A}}$ *being Hurwitz. Moreover,* $\|\varepsilon\|_\infty = \sup_{z\in Z}\|\varepsilon(z)\| < +\infty$, *and the mapping* $z \to \mathbf{A}(z)$ *is bounded, that is,*

$$C_{\mathbf{A}} = \sup_{z\in Z}\|\mathbf{A}(z)\| \vee \sup_{z\in Z}\|\tilde{\mathbf{A}}(z)\| < \infty \ .$$

*Moreover, for* $\Sigma_\varepsilon$ *defined in* (5), *we assume that* $\lambda_{\min}(\Sigma_\varepsilon) > 0$.

To motivate the introduction of $\Sigma_\varepsilon$ in this particular form, note that under assumption A 2, $\mathbb{E}_\pi[\varepsilon(Z_0)] = 0$, the following central limit theorem holds (see e.g. [23, Chapter 21]):

$$\frac{1}{\sqrt{n}}\sum_{\ell=0}^{n-1}\varepsilon(Z_\ell) \to \mathcal{N}(0, \Sigma_\varepsilon) \ .$$

The fact that the matrix $-\bar{\mathbf{A}}$ is Hurwitz implies that the linear system $\bar{\mathbf{A}}\theta = \bar{\mathbf{b}}$ has a unique solution $\theta^\star$. Moreover, this fact is sufficient to show that the matrix $\mathrm{I} - \alpha\bar{\mathbf{A}}$ is a contraction in an appropriate matrix $Q$-norm for small enough $\alpha > 0$. Precisely, the following result holds:

**Proposition 2** (Proposition 1 in [67]). *Let* $-\bar{\mathbf{A}}$ *be a Hurwitz matrix. Then for any* $P = P^\top \succ 0$, *there exists a unique matrix* $Q = Q^\top \succ 0$, *satisfying the Lyapunov equation* $\bar{\mathbf{A}}^\top Q + Q\bar{\mathbf{A}} = P$. *Moreover, setting*

$$a = \frac{\lambda_{\min}(P)}{2\|Q\|} \ , \quad \text{and} \quad \alpha_\infty = \frac{\lambda_{\min}(P)}{2\kappa_Q\|\bar{\mathbf{A}}\|_Q^2} \wedge \frac{\|Q\|}{\lambda_{\min}(P)} \ , \tag{7}$$

*where* $\kappa_Q = \lambda_{\max}(Q)/\lambda_{\min}(Q)$, *it holds for any* $\alpha \in [0, \alpha_\infty]$ *that* $\alpha a \leq 1/2$, *and*

$$\|\mathrm{I} - \alpha\bar{\mathbf{A}}\|_Q^2 \leq 1 - \alpha a \ .$$

For completeness we provide the proof of this result in Appendix D. Now, we make an assumption about the form of step sizes $\alpha_k$ in (4) and the total number of observations $n$:

**A 3.** *Step sizes* $\{\alpha_k\}_{k\in\mathbb{N}}$ *have a form* $\alpha_k = c_0/(k + k_0)^\gamma$, *where* $\gamma \in [1/2; 1)$, *and the constant* $c_0 \leq 1/(2a)$. *Moreover, we assume that* $n$ *is sufficiently large and*

$$k_0 > g(a, t_{\mathrm{mix}}, c_0, C_{\mathbf{A}}\,\kappa_Q, \alpha_\infty)(\log n)^{1/\gamma}.$$

*Precise expressions for* $g(a, t_{\mathrm{mix}}, c_0, C_{\mathbf{A}}\,\kappa_Q, \alpha_\infty)$ *and for* $n$ *are given in Appendix A (see A' 3).*

The main aim of lower bounding $n$ is to ensure that the error related to the choice of initial condition $\theta_0$ becomes small enough. Note also that our bound of A 3 requires to known the number of observations $n$ in advance and adjust $k_0$ accordingly. The same problem can be traced in the existing high-probability results for LSA [47, 24, 82], in the regime when the confidence parameter $\delta$ in high-probability bounds scales with the number of iterations $n$.

It is known (see e.g. [31]), that the assumptions A 1 - A 3 guarantee that the sequence $\bar{\theta}_n$ is asymptotically normal, that is, $\sqrt{n}(\bar{\theta}_n - \theta^\star) \overset{d}{\to} \mathcal{N}(0, \Sigma_\infty)$, where the covariance matrix $\Sigma_\infty$ has a form

$$\Sigma_\infty = \bar{\mathbf{A}}^{-1}\Sigma_\varepsilon\bar{\mathbf{A}}^{-\top} \ . \tag{8}$$

For a fixed $u \in \mathbb{S}^{d-1}$ we consider projection of $\sqrt{n}(\bar{\theta}_n - \theta^\star)$ on $u$ and quantify the rate of convergence in the Kolmogorov distance, i.e., we consider the quantity

$$\mathsf{d}_K\big(\sqrt{n}u^\top(\bar{\theta}_n - \theta^\star)/\sigma(u), \mathcal{N}(0, 1)\big) = \sup_{x\in\mathbb{R}}|\mathbb{P}(\sqrt{n}u^\top(\bar{\theta}_n - \theta^\star)/\sigma(u) \leq x) - \Phi(x)| \ , \tag{9}$$

where $\Phi(x)$ is the c.d.f. of the standard normal law $\mathcal{N}(0, 1)$, and $\sigma^2(u) = u^\top\bar{\mathbf{A}}^{-1}\Sigma_\varepsilon\bar{\mathbf{A}}^{-\top}u$. To control the quantity (9), we will first present an auxiliary result. Define

$$G_{m:k} = \prod_{\ell=m}^{k}(\mathrm{I} - \alpha_\ell\bar{\mathbf{A}}), \ Q_\ell = \alpha_\ell\sum_{k=\ell}^{n-1}G_{\ell+1:k} \ , \ \Sigma_n = \frac{1}{n}\sum_{\ell=2}^{n-1}Q_\ell\Sigma_\varepsilon Q_\ell^\top \ , \ \sigma_n^2(u) = u^\top\Sigma_n u \ . \tag{10}$$

**Theorem 1.** *Assume A1, A2, and A3. Then for any* $u \in \mathbb{S}_{d-1}$, $\theta_0 \in \mathbb{R}^d$, *and initial distribution* $\xi$ *on* $(Z, \mathcal{Z})$, *it holds that*

$$\mathsf{d}_K\big(\sqrt{n}u^\top(\bar{\theta}_n - \theta^\star)/\sigma_n(u), \mathcal{N}(0, 1)\big) \leq \mathrm{B}_n \ ,$$

*where we set*

$$\mathrm{B}_n = \frac{C_{K,1}\log^{3/4}n}{n^{1/4}} + \frac{C_{K,2}\log n}{n^{1/2}} + \frac{C_1^{\mathsf{D}}\|\theta_0 - \theta^\star\| + C_2^{\mathsf{D}}}{\sqrt{n}} + C_3^{\mathsf{D}}\frac{(\log n)^2}{n^{\gamma-1/2}} + C_4^{\mathsf{D}}\frac{(\log n)^{5/2}}{n^{\gamma-1/2}} \ , \tag{11}$$

*where* $C_{K,1}$ *and* $C_{K,2}$ *are defined in Appendix F, and* $\{C_i^{\mathsf{D}}\}_{i=1}^4$ *are defined in Appendix B.*

*Proof.* We provide below the main elements of the proof and refer the reader to Appendix B for a complete derivations. We use the expansion (35) and (39), detailed in the appendix:

$$\theta_k - \theta^\star = \Gamma_{1:k}(\theta_0 - \theta^\star) + J_k^{(0)} + H_k^{(0)} \ ,$$

where we set $\Gamma_{m:k} = \prod_{\ell=m}^k (I - \alpha_\ell \mathbf{A}(Z_\ell))$ and for $m \leq k$,

$$J_k^{(0)} = -\sum_{\ell=1}^k \alpha_\ell G_{\ell+1:k}\varepsilon(Z_\ell) \ , \quad J_0^{(0)} = 0$$

$$H_k^{(0)} = -\sum_{\ell=1}^k \alpha_\ell \Gamma_{\ell+1:k}(\mathbf{A}(Z_\ell) - \bar{\mathbf{A}})J_{\ell-1}^{(0)} \ , \quad H_0^{(0)} = 0 \tag{12}$$

Taking average and changing the order of summation we get that $\sqrt{n}(\bar{\theta}_n - \theta^\star) = W + D_1$, with

$$W = -\frac{1}{\sqrt{n}}\sum_{\ell=1}^{n-1} Q_\ell \varepsilon(Z_\ell), \quad D_1 = \frac{1}{\sqrt{n}}\sum_{k=0}^{n-1}\Gamma_{1:k}(\theta_0 - \theta^\star) + \frac{1}{\sqrt{n}}\sum_{k=1}^{n-1} H_k^{(0)} \ . \tag{13}$$

Note that $W$ is a linear statistic of the Markov chain $\{Z_\ell\}_{\ell\in\mathbb{N}}$. We further expand it into a sum of martingale-differences and remainder terms. Note that under A1 the function $\hat{\varepsilon}(z) = \sum_{k=0}^\infty \mathrm{P}^k\varepsilon(z)$ is a solution to *Poisson equation* $\hat{\varepsilon}(z) - \mathrm{P}\hat{\varepsilon}(z) = \varepsilon(z)$. We rewrite the term $W$ in the following way $W = n^{-1/2}M + D_2$, with

$$M = -\sum_{\ell=2}^{n-1}\Delta M_\ell \ , \qquad \Delta M_\ell = Q_\ell\big(\hat{\varepsilon}(Z_\ell) - \mathrm{P}\hat{\varepsilon}(Z_{\ell-1})\big)$$

$$D_2 = -\frac{1}{\sqrt{n}}Q_1\hat{\varepsilon}(Z_1) + \frac{1}{\sqrt{n}}Q_{n-1}\mathrm{P}\hat{\varepsilon}(Z_{n-1}) + \sum_{\ell=1}^{n-2}(Q_\ell - Q_{\ell+1})\mathrm{P}\hat{\varepsilon}(Z_\ell) \ .$$

Note that $\{\Delta M_\ell\}_{\ell=2}^{n-1}$ is a martingale-difference sequence w.r.t. $\mathcal{F}_\ell = \sigma(Z_k, k \leq \ell)$. Note that

$$\mathbb{E}^{\mathcal{F}_{\ell-1}}[\Delta M_\ell \Delta M_\ell^\top] = Q_\ell\tilde{\varepsilon}(Z_{\ell-1})Q_\ell^\top, \quad \tilde{\varepsilon}(z) = \mathrm{P}\hat{\varepsilon}\hat{\varepsilon}^\top(z) - \mathrm{P}\hat{\varepsilon}(z)\mathrm{P}\hat{\varepsilon}^\top(z) \ . \tag{14}$$

Furthermore, we have $\pi(\tilde{\varepsilon}) = \Sigma_\varepsilon$ . The term $M$ is a martingale, whose quadratic characteristic is given by $\langle M \rangle_n = \sum_{\ell=1}^{n-2} Q_{\ell+1}\tilde{\varepsilon}(Z_\ell)Q_{\ell+1}^\top$. With these notations, we get

$$\sqrt{n}(\bar{\theta}_n - \theta^\star) = n^{-1/2}M + D \ ,$$

where we denote $D = D_1 + D_2$. Applying Proposition 14 with $X = n^{-1/2}u^\top M/\sigma_n(u)$ and $Y = u^\top D/\sigma_n(u)$, we obtain for any $p \geq 2$,

$$\mathsf{d}_K\big(\tfrac{\sqrt{n}u^\top(\bar{\theta}_n-\theta^\star)}{\sigma_n(u)}, \mathcal{N}(0,1)\big) \leq \mathsf{d}_K\big(\tfrac{u^\top M}{\sqrt{n}\sigma_n(u)}, \mathcal{N}(0,1)\big) + 2\big\{\mathbb{E}[|\tfrac{u^\top D}{\sigma_n(u)}|^p]\big\}^{1/(p+1)} \ . \tag{15}$$

To obtain the rate of Gaussian approximation for $\sqrt{n}u^\top(\bar{\theta}_n - \theta^\star)/\sigma_n(u)$ it remains to control the moments of the term $|u^\top D/\sigma_n(u)|$, which is done in Proposition 7, and to control $\mathsf{d}_K\big(u^\top M/(\sqrt{n}\sigma_n(u))\big)$. To bound the latter term we use a normal approximation result for sums of martingale-difference sequences, which builds upon the arguments of [12] and [28] - see Proposition 6, applied with $p = \log n$. $\qquad\square$

Note that the result of Theorem 1 yields an approximation of $\sqrt{n}u^\top(\bar{\theta}_n - \theta^\star)$ with $\mathcal{N}(0, \sigma_n^2(u))$, and not the limiting quantity $\sigma^2(u)$ from the CLT (2). In order to complete the result, we need an additional result on the Gaussian comparison between $\mathcal{N}(0, \sigma_n^2(u))$ and $\mathcal{N}(0, \sigma^2(u))$. This result is based is based on the quantitative estimates provided first in [6], and then revised in [21].

**Corollary 1.** *Under assumptions of Theorem 1 it holds, with* $\mathrm{B}_n$ *given in* (11)*, that*

$$\mathsf{d}_K\big(\sqrt{n}u^\top(\bar{\theta}_n - \theta^\star)/\sigma(u), \mathcal{N}(0,1)\big) \leq \mathrm{B}_n + C_\infty n^{\gamma-1} \ ,$$

*where* $C_\infty$ *is defined in* (30)*.*

**Remark 1.** *The bound of Corollary 1 predicts the optimal error of normal approximation for Polyak-Ruppert averaged estimates of order* $n^{-1/4}$ *up to a logarithmic factors in* $n$*, which is achieved with the step size* $\alpha_k = c_0/(k + k_0)^{3/4}$*, that is, when setting* $\gamma = 3/4$ *in* (11)*. In this case we obtain the optimized bound:*

$$\mathsf{d}_K\big(\sqrt{n}u^\top(\bar{\theta}_n - \theta^\star)/\sigma(u), \mathcal{N}(0,1)\big) \lesssim_{pr} \frac{(\log n)^{5/2}}{n^{1/4}} \ .$$

*where* $\lesssim_{pr}$ *stands for inequality up to absolute and problem-specific constants (such as* $C_\mathbf{A}, \kappa_Q, a, t_{\mathrm{mix}}$*), but not* $n$*.*

**Discussion.** The proof of Corollary 1 is given in Appendix B.3. Results similar to the one of Corollary 1 have been recently obtained in the literature in [67], [73] and [3]. [67] considered the LSA algorithm based on i.i.d. noise variables $\{Z_k\}_{1 \le k \le n}$ and a randomized concentration approach based on [69]. This result was later refined in [82]. However, this technique does not extend to Markovian noise $\{Z_k\}_{1 \le k \le n}$. While preparing this manuscript, we became aware of a recent paper [83], which provides a quantitative $d$-dimensional CLT for martingales. Given this work, a natural direction for further research is to use this result and generalize Theorem 1 to the $d$-dimensional setting. The authors in [83] applied their findings to the particular setting of the TD learning algorithm with Markov noise and obtained a bound of the form

$$\mathsf{d}_C(\sqrt{n}(\bar{\theta}_n - \theta^\star), \Sigma_\infty^{1/2}\eta) \lesssim \frac{\log n}{n^{1/4}} ,$$

where $\eta \sim \mathcal{N}(0, \mathrm{I}_d)$, and $\mathsf{d}_C(X, Y) = \sup_{B \in \mathrm{Conv}(\mathbb{R}^d)} \left| \mathbb{P}(X \in B) - \mathbb{P}(Y \in B) \right|$ denotes the convex distance. The authors in [73] obtained the convergence rate for the general LSA procedure with Markov noise

$$\mathsf{d}_W(\sqrt{n}(\bar{\theta}_n - \theta^\star), \Sigma_\infty^{1/2}\eta) \lesssim \frac{\sqrt{\log n}}{n^{1/6}} ,$$

where $\mathsf{d}_W$ is the 1-st order Wasserstein distance. This result implies, due to the classical relations of between Kolmogorov and Wasserstein distance (see [62]), the final rate approximation on Kolmogorov distance of order $\mathcal{O}(n^{-1/12})$, which is slower compared to the rate of Theorem 1. On the other hand, the result of [73] holds in $d$-dimensional setting, whereas our analysis is restricted to one-dimensional projections of the estimation error. Nevertheless, the essential part of our analysis, namely controlling the remainder term $D$ in (15), can be generalized to the $d$-dimensional case through bounds on $\mathbb{E}_\xi^{1/p}[\|D\|^p]$, following an approach similar to that presented in Appendix B.

## 4   Multiplier subsample bootstrap for LSA

We will apply the multiplier subsample bootstrap (MSB) procedure, a block-based approach that constructs the bootstrap statistic via a blockwise scheme; see [37, 43]. Below we describe in details the MSB approach, closely following [43]. Let $b_n$ be the length of block, and for each $t = 0, \ldots, n - b_n$, define $\bar{\theta}_{b_n, t} = (1/b_n) \sum_{\ell=t}^{t+b_n-1} \theta_\ell$, the "scale $b_n$" version of $\bar{\theta}_n$. To imitate $\bar{\theta}_n$, the MSB estimator of $\bar{\theta}_n$ is given by

$$\bar{\theta}_{n,b_n}(u) = \frac{\sqrt{b_n}}{\sqrt{n - b_n + 1}} \sum_{t=0}^{n-b_n} w_t (\bar{\theta}_{b_n, t} - \bar{\theta}_n)^\top u , \tag{16}$$

where $\Xi_n = \{Z_\ell\}_{\ell=1}^n$, and $\{w_\ell\}_{0 \le \ell \le n - b_n}$, the multiplier weights, are i.i.d. $\mathcal{N}(0, 1)$ random variables, which are independent of $\Xi_n$. We write, respectively, $\mathbb{P}^b = \mathbb{P}(\cdot | \Xi_n)$ and $\mathbb{E}^b = \mathbb{E}(\cdot | \Xi_n)$ for the corresponding conditional probability and expectation. For simplicity, we do not "subsample" the blocks: the theory extends readily, but we do not want to add another layer of notations. The key idea of the MSB procedure (16) is that the "bootstrap world" distribution $\mathbb{P}^b(\bar{\theta}_{n,b_n}(u) \le x)$ should be close to their "real world" counterparts $\mathbb{P}(\sqrt{n}(\bar{\theta}_n - \theta^\star) \le x)$ for any $x \in \mathbb{R}$. Formally, the procedure (16) is said to be *asymptotically valid*, if the quantity

$$\sup_{x \in \mathbb{R}} |\mathbb{P}(\sqrt{n}(\bar{\theta}_n - \theta^\star)^\top u \le x) - \mathbb{P}^b(\bar{\theta}_{n,b_n}(u) \le x)| \tag{17}$$

converges to 0 in $\mathbb{P}$-probability. Typically the authors consider the asymptotic validity of the procedures for constructing the confidence intervals (either with multiplier bootstrap [29, 43] or with direct estimation of the asymptotic covariance [17, 79]). Our aim in this section is to provide *fully non-asymptotic* bounds on the rate at which the supremum in (17) decays as a function of $n$.

The MSB estimator (16) $\bar{\theta}_{n,b_n}(u)$ is normally distributed w.r.t. $\mathbb{P}^b$, that is,

$$\bar{\theta}_{n,b_n}(u) \sim \mathcal{N}(0, \hat{\sigma}_\theta^2(u)) ,$$

with the variance $\hat{\sigma}_\theta^2(u)$ given by

$$\hat{\sigma}_\theta^2(u) = \frac{b_n}{n - b_n + 1} \sum_{t=0}^{n-b_n} ((\bar{\theta}_{b_n, t} - \bar{\theta}_n)^\top u)^2 . \tag{18}$$

The parameter $b_n$ is commonly referred to as *lag window*, or *bandwidth* (see [30] and references therein). Under the bootstrap probability $\mathbb{P}^b$, $\bar{\theta}_{n,b_n}(u)$ (see (16)) is a Gaussian approximation of $\bar{\theta}_n$

with estimated variance $\hat{\sigma}_\theta^2(u)$. Up to a multiplicative factor tending to 1 as $n$ goes to $\infty$, the variance formula in (18) coincides with the *overlapping batch mean* estimator (OBM), a well-known technique for estimating the asymptotic variance of the Markov chain, suggested in [45]. The properties of OBM estimator are studied, see e.g. [81, 20, 30]. In particular, it is known (see [30]), that the OBM is a consistent estimator of the asymptotic variance of a Markov chain under suitable ergodicity assumptions, provided that $b_n \to \infty$ as $n \to \infty$.

Note that even if $\{Z_k\}_{k\in\mathbb{N}}$ is a Markov chain, the iterates $\{\theta_k\}_{k\in\mathbb{N}}$ alone do not form a Markov chain (one must rather consider the joint process $(\theta_k, Z_k)$). Consequently, the classical consistency results for overlapping-batch-means variance estimators do not apply directly to (18). Fortunately, we show below that applying the block bootstrap for the sequence $\{\theta_\ell\}$, is equivalent, up to a suitable correction, to the block bootstrap procedure applied to the non-observable random variables $\{\varepsilon(Z_\ell)\}$. To make this precise, we define for each block start $t \in \{0, \ldots, n - b_n\}$ the quantities:

$$\bar{W}_{b_n,t} = \frac{1}{b_n} \sum_{\ell=t+1}^{t+b_n-1} \bar{\mathbf{A}}^{-1}\varepsilon(Z_\ell) \ , \quad \bar{W}_n = \frac{1}{n}\sum_{\ell=1}^{n-1}\bar{\mathbf{A}}^{-1}\varepsilon(Z_\ell),$$

$$\hat{\sigma}_\varepsilon^2(u) = \frac{b_n}{n - b_n + 1}\sum_{t=0}^{n-b_n}((\bar{W}_{b_n,t} - \bar{W}_n)^\top u)^2 \ .$$

Then the following proposition holds:

**Proposition 3.** *Assume A1, A2, and A3. Then for any $u \in \mathbb{S}_{d-1}$, it holds that*

$$\hat{\sigma}_\theta^2(u) = \hat{\sigma}_\varepsilon^2(u) + \mathcal{R}_{var}(u) \ ,$$

*where $\mathcal{R}_{var}(u)$ is a remainder term defined in Appendix E.1 (see (63)). Moreover, for any $2 \leq p \leq \log n$, and any initial distribution $\xi$ on $(\mathsf{Z}, \mathcal{Z})$, it holds that*

$$\mathbb{E}_\xi^{1/p}\big[|\mathcal{R}_{var}(u)|^p\big] \lesssim M_1 p b_n^{1/2} n^{\gamma/2-1} + M_2 p^4 (\log n) b_n^{1/2} n^{-\gamma} \tag{19}$$
$$+ M_3 p b_n^{-1/2} n^{\gamma/2} + M_4 p^4 (\log n) n^{-1} + M_5 p n^{2\gamma-2}$$

*and the constants $M_i$ are defined in Appendix E.1, equation (66).*

The proof of Proposition 3 is given in Appendix E.1. The bound of Proposition 3 has some noteworthy properties. First, our bound requires that the block size $b_n$ to grow at least like $n^\gamma$ to ensure that the residual term $\mathcal{R}_{var}(u)$ is small. Careful inspection of the bootstrap estimator $\bar{\theta}_{n,b_n}(u)$ from (16) explains this dependence. Indeed, one can expect that the decay rate of the covariance between $\theta_k$ and $\theta_{k+b_n}$ depends on the quantity $\sum_{k=t+1}^{t+b_n}\alpha_k$, see e.g. [22]. At the same time, for $b_n \simeq n^\beta, 0 < \beta < 1$, and $\alpha_k = c_0/(k_0 + k)^\gamma$, which is set according to A3, we obtain that

$$\sum_{k=t+1}^{t+b_n}\alpha_k \lesssim b_n/t^\gamma \ .$$

Hence, if $b_n \ll n^\gamma$, the sequence $\{\theta_k\}$ does not mix within each block $t \leq k \leq t + b_n$.

Given that the remainder term $\mathcal{R}_{var}(u)$ in (19) is negligible, it remains to analyze the concentration properties of $\sigma_\varepsilon^2(u)$. Indeed, under A1 and A2, the (normalized) linear statistic $n^{-1/2}\sum_{\ell=0}^{n-1}u^\top\varepsilon(Z_\ell)$ is asymptotically normal with the asymptotic variance equal to $\sigma^2(u)$ defined in (9). Moreover, $\hat{\sigma}_\varepsilon^2(u)$ coincides with the overlapping batch mean estimator of $\sigma^2(u)$. In order to prove concentration bounds for $\hat{\sigma}_\varepsilon^2(u)$ around $\sigma^2(u)$, we apply the result of [50, Theorem 1].

**Proposition 4.** *Assume A1 and A2. Then for any $p \geq 2$, and $n \geq 2b_n + 1$, and any initial distribution $\xi$ on $(\mathsf{Z}, \mathcal{Z})$, it holds that*

$$\mathbb{E}_\xi^{1/p}[|\hat{\sigma}_\varepsilon^2(u) - \sigma^2(u)|^p] \lesssim \frac{pt_{\mathrm{mix}}^3\|\varepsilon\|_\infty^2}{\sqrt{n}} + \frac{p^2 t_{\mathrm{mix}}^2\sqrt{b_n}\|\varepsilon\|_\infty^2}{\sqrt{n}} + \frac{pt_{\mathrm{mix}}^2\|\varepsilon\|_\infty^2}{\sqrt{b_n}} \ .$$

The result above is based on martingale decomposition, associated with the Poisson equation for quadratic forms of Markov chains, as introduced in [5]. Now, combining the estimates of Propositions 3 and 4 and applying Markov's inequality with $p = \log n$, we obtain the following result:

**Corollary 2.** *Let $n$ be large enough. Set $b_n = \lceil n^{3/4} \rceil$, $\varepsilon \in (0; 1/\log n)$, and let $\alpha_k = c_0/(k_0 + k)^{1/2+\varepsilon}$. Then, with probability at least $1 - n^{-1}$, it holds that*

$$\left|\hat{\sigma}_\theta^2(u) - \sigma^2(u)\right| \lesssim_{\log n} n^{-1/8+\varepsilon/2} \ . \tag{20}$$

**Discussion.** The version of Corollary 2 with explicit constants and explicit power of $\log n$ is provided in Appendix E.3, together with the proof of Corollary 2. Note that the fastest decay rate in the r.h.s of (20) is achieved when we set $b_n = \mathcal{O}(n^{3/4})$ and aggressive step sizes $\alpha_k = c_0/(k_0 + k)^{1/2+\varepsilon}$. The same choice of hyperparameters appears to be optimal in the recent work of [63] for batch-mean estimators of the asymptotic variance for the SGD algorithm, even in case of controlled Markov chain. We also recover the rate $n^{-1/8}$ (up to logarithmic factors), that previously appeared in [63]. The authors of [17, 79] also considered the batch mean estimators $\widehat{\Sigma}_n$ of the asymptotic variance $\Sigma_\infty$ in case of SGD algorithms with independent noise and obtained the same (up to logarithmic factors) convergence rate $\mathbb{E}[\|\widehat{\Sigma}_n - \Sigma_\infty\|] \lesssim n^{-1/8}$. Moreover, the optimal rate for recovering $\Sigma_\infty$ in [17, Corollary 4.5] and [79, Corollary 3.4] is attained for the step sizes $\alpha_k = 1/k^\gamma$ with $\gamma \to 1/2$. This is on par with our findings of Corollary 2. At the same time, one should note that this choice of step sizes yields extremely slow convergence rates in the CLT in Theorem 1. This introduces an additional trade-off, that needs to be taken into account when considering the decay rate of (17). Namely, one needs to balance not only the right-hand sides of Corollary 2 and Proposition 3, but also to take into account the convergence rate in Theorem 1. The respective trade-off yields

$$b_n = \lceil n^{4/5} \rceil , \quad \alpha_k = c_0/(k_0 + k)^{3/5} . \tag{21}$$

The corresponding main theorem writes as follows:

**Theorem 2.** *Assume A1, A2, and A3, let $n$ be large enough, set $b_n = \lceil n^{4/5} \rceil$, $\alpha_k = c_0/(k_0 + k)^{3/5}$. Then for any $u \in \mathbb{S}_{d-1}$, and any initial distribution $\xi$ on $(\mathsf{Z}, \mathcal{Z})$, it holds with $\mathbb{P}$ – probability at least $1 - 1/n$ that*

$$\sup_{x\in\mathbb{R}} |\mathbb{P}(\sqrt{n}(\bar{\theta}_n - \theta^\star)^\top u \le x) - \mathbb{P}^{\mathsf{b}}(\bar{\theta}_{n,b_n}(u) \le x)| \lesssim_{\log n} n^{-1/10} .$$

*Proof.* We provide the detailed proof of Theorem 2 in Appendix E.4 together with the explicit form of condition on $n$. The proof is based on the following scheme:

$$\text{Real world:}\sqrt{n}u^\top(\bar{\theta}_n - \theta^\star) \xleftarrow{\text{Gaussian approximation, Cor. 1}} \xi \sim \mathcal{N}(0, \sigma^2(u))$$
$$\Big\updownarrow {\scriptstyle\text{Gaussian comparison}} \tag{22}$$
$$\text{Bootstrap world:} \quad \bar{\theta}_{n,b}(u) \xleftarrow{\text{exactly matches the distribution}} \xi^{\mathsf{b}} \sim \mathcal{N}(0, \hat{\sigma}_\theta^2(u))$$

Due to Corollary 1, it holds that

$$\mathsf{d}_K\big(\sqrt{n}u^\top(\bar{\theta}_n - \theta^\star), \xi\big) \lesssim_{\log n} n^{-1/4} + n^{1/2-\gamma} + n^{\gamma-1} ,$$

where $\xi \sim \mathcal{N}(0, 1)$. This result allows for the first horizontal bar above. We now use the Gaussian comparison (see [6], [21]), which states that for $\xi_i \sim \mathcal{N}(0, \sigma_i^2)$, $i = 1, 2$, are such that $|\sigma_1^2/\sigma_2^2 - 1| \le \delta$, for some $\delta \ge 0$, $\sup_{x\in\mathbb{R}} |\mathbb{P}(\xi_1 \le x) - \mathbb{P}(\xi_2 \le x)| \le \frac{3}{2}\delta$. We apply the Gaussian comparison between the limiting Gaussian $\xi \sim \mathcal{N}(0, \sigma^2(u))$ and $\xi^b \sim \mathcal{N}(0, \hat{\sigma}_\theta^2(u))$. For this purpose, we need to obtain a high probability bound for $|\hat{\sigma}_\theta^2(u) - \sigma^2(u)|/\sigma^2(u)$. Combining Proposition 3, Proposition 4 and Markov's inequality, we get that with probability at least $1 - 1/n$,

$$|\hat{\sigma}_\theta^2(u) - \sigma^2(u)|/\sigma^2(u) \lesssim_{\log n} b_n^{1/2}n^{\gamma/2-1} + b_n^{1/2}n^{-\gamma} + b_n^{-1/2}n^{\gamma/2} + n^{2\gamma-2} + b^{1/2}n^{-1/2} .$$

Now note that by construction $\bar{\theta}_{n,b} \sim \mathcal{N}(0, \hat{\sigma}_\theta^2(u))$ under the bootstrap probability. Hence, with probability at least $1 - 1/n$, it holds that

$$\mathsf{d}_K\big(\sqrt{n}u^\top(\bar{\theta}_n - \theta^\star), \bar{\theta}_{n,b}\big) \lesssim_{\log n} \frac{b_n^{1/2}}{n^{1-\gamma/2}} + \frac{b_n^{1/2}}{n^{\gamma/2}} + \frac{b_n^{1/2}}{n^{1/2}} + \frac{1}{n^{1/4}} + \frac{1}{n^{\gamma-1/2}} + \frac{1}{n^{1-\gamma}} .$$

To complete the proof, it remains to optimize our choice of $\gamma$ and $b_n$, which yields to (21). $\qquad\square$

**Discussion.** Non-asymptotic analysis of coverage probabilities has been carried out for the modifications of multiplier bootstrap approach of [29] in recent papers [67, 70]. These approaches showed that coverage probabilities of $\theta^\star$ can be approximated by their bootstrap counterparts with the order up to $\mathcal{O}(n^{-1/2})$. Yet for this bootstrap approach it is crucial to work in the independent noise setting. The attempt of [59] to generalize it for the case of Markovian noise yields inconsistent procedure, as

shown in [42, Proposition 1]. That is why we prefer to start from the asymptotically consistent MSB procedure of [43]. Our bootstrap validity proof relies on direct approximation of $\sqrt{n}u^\top(\bar{\theta}_n - \theta^\star)$ by the limiting Gaussian distribution $\mathcal{N}(0, \sigma^2(u))$. At the same time, it is known (see [70]) that under i.i.d. conditions, it may be more advantageous in the proof scheme 22 of the non-asymptotic validity of the bootstrap procedure for $\sqrt{n}u^\top(\bar{\theta}_n - \theta^\star)$ using another Gaussian distribution $\mathcal{N}(0, \sigma_n^2(u))$ and its "bootstrap" analogue. It remains an open question if this reasoning can be applied in case of dependent random variables.

## 5   Application to the TD learning

We illustrate our findings via the temporal-difference (TD) learning algorithm [75, 76] for policy evaluation in a discounted Markov Reward Process (MRP) $(\mathcal{S}, \mathrm{P}, \mathcal{R}, \lambda)$. Here, $\mathcal{S}$ is a complete metric space with its Borel $\sigma$-algebra; $\mathrm{P}(\cdot \mid s)$ denotes the state transition kernel; $\mathcal{R} : \mathcal{S} \to [0, 1]$ is the scalar reward function; $\lambda \in [0, 1)$ is the discount factor. The value function is given by

$$V(s) = \mathbb{E}[\sum_{k=0}^{\infty} \lambda^k \mathcal{R}(S_k) \mid S_0 = s] \text{ where } S_{k+1} \sim \mathrm{P}(\cdot \mid S_k) \,.$$

We approximate $V(s)$ in a linear feature space: $V_\theta(s) = \varphi(s)^\top \theta$, where $\varphi : \mathcal{S} \to \mathbb{R}^d$. We impose two standard assumptions:

**TD 1.** P *is uniformly geometrically ergodic with unique invariant distribution $\mu$ and mixing time $\tau$.*

We also define the design matrix $\Sigma_\varphi = \mathbb{E}_\mu[\varphi(S)\varphi(S)^\top]$ and require that it is non-degenerate:

**TD 2.** $\Sigma_\varphi$ *is non-degenerate, i.e. $\lambda_{\min}(\Sigma_\varphi) > 0$. Moreover, $\sup_{s \in \mathcal{S}} \|\varphi(s)\| \leq 1$.*

Then TD learning is an instance of LSA with

$$\mathbf{A}_k = \varphi(S_k)\big[\varphi(S_k) - \lambda\varphi(S_{k+1})\big]^\top , \quad \bar{\mathbf{A}} = \mathbb{E}_\mu[\mathbf{A}_k] , \quad \mathbf{b}_k = \varphi(S_k)\,\mathcal{R}(S_k) \,.$$

Detailed derivations are given in Appendix H. Refining constants following [53, 68], one shows:

**Proposition 5.** *Under **TD 1** and **TD 2**, the TD updates satisfy the noise level condition A 2 with $\mathrm{C}_{\mathbf{A}} = 2(1+\lambda)$ and $\|\varepsilon\|_\infty = 2(1+\lambda)(\|\theta_\star\|+1)$. Moreover, for step size $\alpha \leq \alpha_\infty = (1-\lambda)/(1+\lambda)^2$, it holds that $\|\mathrm{I} - \alpha\bar{\mathbf{A}}\|^2 \leq 1 - \alpha\,a$, where $a = (1 - \lambda)\,\lambda_{\min}(\Sigma_\varphi)$.*

By [67, Proposition 2], this ensures that Proposition 2 holds with $Q = \mathrm{I}$, and hence Theorem 2 applies directly to the TD scheme. We provide numerical simulations in Appendix H.

## 6   Conclusion

We presented the non-asymptotic Berry–Esseen bounds for Polyak–Ruppert averaged iterates of linear stochastic approximation algorithm under Markovian noise, achieving convergence rates of order up to $n^{-1/4}$ in Kolmogorov distance. Additionally, we established the theoretical validity of a multiplier subsample bootstrap procedure, enabling reliable uncertainty quantification in the setting of the LSA algorithm with Markovian noise. Our paper suggest a number of further research directions. One of them is related with the generalizations of Theorem 1 to the setting of non-linear SA algorithms, as well as with obtaining multivariate version of it. It is also an interesting and important question if our non-asymptotic bounds on coverage probabilities provided in Theorem 2 can be further improved.

## Acknowledgment

The work was supported by the grant for research centers in the field of AI provided by the Ministry of Economic Development of the Russian Federation in accordance with the agreement 000000C313925P4E0002 and the agreement with HSE University № 139-15-2025-009. This research was supported in part through computational resources of HPC facilities at HSE University [36].

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

# Contents

## A  Extended version of A3

In this section, we present the full version of A3 with explicit constants.

**A' 3.** *The sequence of step sizes* $\{\alpha_k\}_{k\in\mathbb{N}}$ *has a form* $\alpha_k = c_0/(k+k_0)^\gamma$, *where* $\gamma \in [1/2; 1)$ *and, with*

$$h = \left\lceil \frac{16 t_{\mathrm{mix}} \kappa_Q^{1/2} \, \mathrm{C_A}}{a} \right\rceil, \tag{23}$$

*it holds that*

$$c_0 \leq \frac{1}{2a} \, . \tag{24}$$

*Moreover, the number of observations* $n$ *and parameter* $k_0$ *satisfy*

$$
\begin{aligned}
& n^{1-\gamma} \geq \max\Big(\frac{2C'_\infty}{\lambda_{\min}(\Sigma_\infty)}, e^{2(1-\gamma)}\Big) \,, \\
& k_0 \geq \max\Big\{ \Big(\frac{24}{ac_0}\Big)^{1/(1-\gamma)}, c_0^{1/\gamma}, \Big(c_0(h+1)\max(\alpha_\infty, \kappa_Q^{1/2}\,\mathrm{C_A}, 6e\kappa_Q\,\mathrm{C_A^2}/a)\Big)^{1/\gamma}, \\
& \qquad\qquad \frac{8 t_{\mathrm{mix}}\kappa_Q^{1/2}\,\mathrm{C_A}}{a}, \Big(\frac{12(h+1)(\log n)c_0\,\mathrm{C_\sigma^2}}{a}\Big)^{1/\gamma} \Big\} \,, \\
& k_0^{1-\gamma} \geq \frac{2}{c_0 b}\Big( \log\{\frac{c_0}{b(2\gamma-1)2^\gamma}\} + \gamma\log\{k_0\} \Big) \,,
\end{aligned}
\tag{25}
$$

**Discussion.** The lower bound on the sample size $n$ in (25) is required to guarantee the non-degeneracy of the empirical covariance matrix $\Sigma_n$ (see its definition in (10) and Lemma 10). This condition ensures that $\lambda_{\min}(\Sigma_n)$ remains bounded away from zero, which is crucial for the proof of rate of convergence in CLT for LSA problem.

Furthermore, the constraint on $k_0$, which scales as $k_0 \gtrsim (\log n)^{1/\gamma}$, is imposed to control higher-order moments of the product of random matrices appearing in the recursion (see Proposition 12). In particular, this ensures that moments up to order $p \sim \log n$ remain finite, which is necessary for establishing the Gaussian approximation bounds used in the main result.

While this assumption could, in principle, be relaxed, doing so would lead to a degradation in the Gaussian approximation rate from $n^{-1/4}$ to $n^{-1/4+\varepsilon(p)}$, where the function $\varepsilon(p)$ depends on the number of finite moments available for the random matrix product.

## B  Proof of Section 3

Recall that in the proof of Theorem 1 (see (15)), we obtained that

$$\mathsf{d}_K\big(\frac{\sqrt{n}u^\top(\bar\theta_n - \theta^\star)}{\sigma_n(u)}, \mathcal{N}(0,1)\big) \leq \mathsf{d}_K\big(\frac{u^\top M}{\sqrt{n}\sigma_n(u)}, \mathcal{N}(0,1)\big) + 2\big\{\mathbb{E}[\big|\frac{u^\top D}{\sigma_n(u)}\big|^p]\big\}^{1/(p+1)} \, .$$

Hence, to conclude with the proof it remains to

    (i)  control the Kolmogorov distance $\mathsf{d}_K\big(\frac{u^\top M}{\sqrt{n}\sigma_n(u)}\big)$;

    (ii)  bound the moments of $\big|\frac{u^\top D}{\sigma_n(u)}\big|$.

We present the required result for item (i) in Appendix B.1 (see Proposition 6) and for (ii) in Appendix B.1 below.

## B.1  Gaussian approximation for sums of martingale-difference sequences

In this section we establish rate of convergence sums of martingale-difference to the gaussian limit. We follow the approach of [28], which builds upon the work [12]. Recall that

$$M = -\sum_{\ell=2}^{n-1} \Delta M_\ell , \quad \Delta M_\ell = Q_\ell\big(\hat{\varepsilon}(Z_\ell) - \mathrm{P}\hat{\varepsilon}(Z_{\ell-1})\big) .$$

Note that the quadratic characteristic

$$\mathbb{E}[|u^\top \Delta M_\ell|^2 | Z_{\ell-1}] = u^\top Q_\ell \tilde{\varepsilon}(Z_{\ell-1}) Q_{\ell+1}^\top u , \quad \tilde{\varepsilon}(z) = \mathrm{P}\hat{\varepsilon}\hat{\varepsilon}^\top(z) - \mathrm{P}\hat{\varepsilon}(z)\mathrm{P}\hat{\varepsilon}^\top(z)$$

is not constant a.s.. Hence, one can not directly apply the results on martingale CLT due to [12, Theorem 2]. Below we first formulate this result, and then demonstrate how to mitigate the problem of time-varying quadratic characteristic following the approach of [66, 28].

**CLT for martingales with constant quadratic characteristic.** Let $X = (X_1,\ldots,X_n)$ be a sequence of real valued random variables which are square integrable and satisfy $\mathbb{E}[X_i|\mathcal{F}_{i-1}] = 0$, a.s. for $1 \le i \le n$, where $\mathcal{F}_i = \sigma(X_1,\ldots,X_i)$. We denote the class of such sequences of length $n$ by $\mathcal{M}_n$. Denote

$$\sigma_j^2 = \mathbb{E}[X_j^2|\mathcal{F}_{j-1}], \quad \hat{\sigma}_j^2 = \mathbb{E}[X_j^2],$$

$$s_n^2 = \sum_{j=1}^{n} \hat{\sigma}_j^2,$$

$$\|X\|_p = \max_{1 \le j \le n} \|X_j\|_p, \quad 1 \le p \le \infty ,$$

and, with $1 \le k \le n$, we have

$$V_k^2 = \sum_{j=1}^{k} \sigma_j^2/s_n^2, \quad S_k = \sum_{j=1}^{k} X_j .$$

**Theorem 3** (Theorem 2 in [12])**.** *Let $0 < \varkappa < \infty$. There exists a constant $0 < L(\varkappa) < \infty$ depending only on $\varkappa$, such that for all $X \in \mathcal{M}_n, n \ge 2$, satisfying*

$$\|X\|_\infty \le \varkappa, \quad V_n^2 = 1 \text{ a.s. },$$

*the following bound holds*

$$\mathsf{d}_K\big(S_n/s_n\big) \le L(\varkappa)n \log n/s_n^3.$$

The constant $L(\varkappa)$ can be quantified following the work of Adrian Röllin, see [66], but exact calculations are beyond the scope of this paper. Now we adapt the arguments from [28] based on [12] to provide the counterpart of this result with time-varying quadratic characteristic. We specify the constants and dependence on $p$ since we need to have possibility to take $p$ of logarithmic order with respect to $n$.

**Lemma 1.** *Let $0 < \varkappa < \infty$. Then, with $L(\varkappa)$ defined in Theorem 3, for all $X \in \mathcal{M}_n, n \ge 2$, satisfying $\|X\|_\infty \le \varkappa$, the following bound holds for any $p \ge 1$*

$$\mathsf{d}_K\big(S_n/s_n\big) \le \frac{L(\varkappa)(2n+1)\log(2n+1)}{s_n^3}$$
$$+ C_1\sqrt{p}s_n^{-\frac{2p}{2p+1}}\big(\mathbb{E}|\sum_{i=1}^{n}\sigma_i^2 - s_n^2|^p\big)^{1/(2p+1)} + C_2 s_n^{-\frac{2p}{2p+1}}p\varkappa^{2p/(2p+1)}.$$

*where we have defined the constants $C_1 = 2\sqrt{2}\,\mathrm{C}_{\mathrm{Rm},1}, C_2 = 4(\mathrm{C}_{\mathrm{Rm},1} + \mathrm{C}_{\mathrm{Rm},2})$.*

*Proof.* Consider the following stopping time $\tau = \sup\{0 \le k \le n : V_k^2 \le 1\}$. Denote

$$r = \lfloor (s_\tau^2 - \sum_{j=1}^{\tau} \sigma_j^2)/\varkappa^2 \rfloor .$$

Note that $r \leq n$. Let $N = 2n + 1$. Construct the following sequence $X' = (X'_1, \ldots, X'_N)$:

$$X'_i = X_i, i \leq \tau; \quad X'_i = \varkappa \eta_i, \tau + 1 \leq i \leq \tau + r; \quad X'_i = (s_n^2 - \sum_{i=1}^{\tau} \sigma_i^2 - r \varkappa^2)^{1/2} \eta_i, i = \tau + r + 1;$$

and $X'_i = 0, i \geq \tau + r + 2$. Here $\eta_i$ are Rademacher random variables, which are independent of all other r.v.'s. Then $X'$ is a vector of martingale increments w.r.t. an extended filtration $\mathcal{F}'_i = \sigma(X_j, \eta_j, j \leq i)$, and, by construction

$$\sum_{i=1}^{N} \mathbb{E}[(X'_i)^2 | \mathcal{F}'_{i-1}] = \sum_{i=1}^{\tau} \sigma_i^2 + r \varkappa^2 + s_n^2 - \sum_{i=1}^{\tau} \sigma_i^2 - r \varkappa^2 = s_n^2.$$

Applying Proposition 14 with $X = S_n/s_n$ and $Y = S'_N/s_n$, we get

$$d_K(S_n/s_n) \leq d_K(S'_N/s_n) + 2 s_n^{-\frac{2p}{2p+1}} (\mathbb{E}[|S_n - S'_N|^{2p}])^{1/(2p+1)}. \tag{26}$$

We control the term $d_K(S'_N/s_n)$ with [12, Theorem 2]:

$$d_K(S'_N/s_n) \leq \frac{L(\varkappa) N \log N}{s_n^3}.$$

In order to control the second term in the right-hand side of (26), we notice that

$$S_n - S'_N = \sum_{i \geq \tau + 1} (X_i - X'_i) = \sum_{i=1}^{N} \mathbf{1}_{\tau \leq i-1}(X_i - X'_i).$$

Since $\tau$ is a stopping time, we can condition on it and get by the Rosenthal's inequality

$$\mathbb{E}[|S_n - S'_N|^{2p}] \leq C_{\mathsf{Rm},1}^{2p} p^p \mathbb{E}[|\sum_{i=\tau+1}^{N} \mathbb{E}[(X_i - X'_i)^2 | \mathcal{F}'_{i-1}]|^p] + C_{\mathsf{Rm},2}^{2p} p^{2p} \mathbb{E}[\max_{\tau+1 \leq i \leq N} |X_i - X'_i|^{2p}]$$

It is easy to see that

$$\sum_{i=\tau+1}^{N} \mathbb{E}[(X_i - X'_i)^2 | \mathcal{F}'_{i-1}] = \sum_{i=\tau+1}^{n} \mathbb{E}[X_i^2 | \mathcal{F}'_{i-1}] + \sum_{i=\tau+1}^{N} \mathbb{E}[(X'_i)^2 | \mathcal{F}'_{i-1}] = \sum_{i=1}^{n} \sigma_i^2 + s_n^2 - 2 \sum_{i=1}^{\tau} \sigma_i^2.$$

Note that

$$s_n^2 - \varkappa^2 \leq \sum_{i=1}^{\tau} \sigma_i^2 \leq s_n^2.$$

Hence, it holds that

$$\sum_{i=\tau+1}^{N} \mathbb{E}[(X_i - X'_i)^2 | \mathcal{F}'_{i-1}] \leq \sum_{i=1}^{n} \sigma_i^2 - s_n^2 + 2 \varkappa^2.$$

Finally,

$$\mathbb{E}[|S_n - S'_N|^{2p}] \leq 2^{p-1} C_{\mathsf{Rm},1}^{2p} p^p (\mathbb{E}|\sum_{i=1}^{n} \sigma_i^2 - s_n^2|^p + 2^p \varkappa^{2p}) + C_{\mathsf{Rm},2}^{2p} (2p)^{2p} \varkappa^{2p})^{1/(2p+1)}.$$

Substituting the above bound into (26), we obtain

$$d_K(S_n/s_n) \leq d_K(S'_N/s_n)$$
$$+ 2 s_n^{-\frac{2p}{2p+1}} (2^{p-1} C_{\mathsf{Rm},1}^{2p} p^p (\mathbb{E}|\sum_{i=1}^{n} \sigma_i^2 - s_n^2|^p + 2^p \varkappa^{2p}) + C_{\mathsf{Rm},2}^{2p} (2p)^{2p} \varkappa^{2p})^{1/(2p+1)},$$

and the statement follows. $\qquad \square$

We now state a lemma that allows us to specify the result of Lemma 1 to the martingale $M$ and its quadratic characteristic $\langle M \rangle_n$, given in (14).

**Lemma 2.** *Assume A1. Then for any $u \in \mathbb{S}_{d-1}$, probability measure $\xi$ on $(\mathsf{Z}, \mathcal{Z})$, and $p \geq 2$:*

$$\mathbb{E}_{\xi}^{1/p}[|u^{\top}\langle M \rangle_n u - n\sigma_n^2(u)|^p] \leq 32n^{1/2}p^{1/2}\mathcal{L}_Q^2 \|\varepsilon\|_{\infty}^2 t_{\text{mix}}^{5/2}$$

*Proof.* Using the definition (14), we get:

$$u^{\top}\langle M \rangle_n u - \sigma_n^2(u) = \sum_{\ell=1}^{n-2}\{h_{\ell}(Z_{\ell}) - \pi(h_{\ell})\} \quad \text{where} \quad h_{\ell}(z) = u^{\top}Q_{\ell+1}\tilde{\varepsilon}(z)Q_{\ell+1}^{\top}u \,.$$

Note that using Lemma 3, for any $z, z' \in \mathsf{Z}$ and any $\ell \in [1, \ldots, n-2]$ we have

$$|h_{\ell}(z) - h_{\ell}(z')| \leq 2\mathcal{L}_Q^2 \|\tilde{\varepsilon}(z)\| \leq 4\mathcal{L}_Q^2 \|\hat{\varepsilon}(z)\|^2 \leq 8\mathcal{L}_Q^2 \|\varepsilon\|_{\infty}^2 (\sum_{k=0}^{\infty}\Delta(\mathrm{P}^k))^2$$

$$\leq 8\mathcal{L}_Q^2 \|\varepsilon\|_{\infty}^2 (\sum_{k=0}^{\infty}\sum_{r=0}^{t_{\text{mix}}-1}(1/4)^{\lceil(kt_{\text{mix}}+r)/t_{\text{mix}}\rceil})^2 \leq 2(8/3)^2 \mathcal{L}_Q^2 \|\varepsilon\|_{\infty}^2 t_{\text{mix}}^2$$

We then apply [54][Corollary 2.11], showing that for all $t \geq 0$,

$$\mathbb{P}_{\xi}(|\sum_{\ell=1}^{n-1}\{h_{\ell}(Z_{\ell}) - \pi(h_{\ell})\}| \geq t) \leq 2\exp\left(-\frac{t^2}{2u_n^2}\right), \text{where} \quad u_n = (64/3)\mathcal{L}_Q^2 \|\varepsilon\|_{\infty}^2 t_{\text{mix}}^{5/2} n^{1/2}$$

We conclude by using [24, Lemma 7]. $\qquad\square$

By combining Lemma 1 and Lemma 2, we finally get the following result:

**Proposition 6.** *Assume A1, A2, and A3. Then for any $u \in \mathbb{S}_{d-1}$ and $p \geq 1$:*

$$\mathsf{d}_K\left(\frac{u^{\top}M}{\sqrt{n}\sigma_n(u)}\right) \leq \frac{L(\varkappa)(2n+1)\log(2n+1)C_{\Sigma}^3}{n^{3/2}}$$

$$+ 4\sqrt{2}C_1\left(\mathcal{L}_Q \|\varepsilon\|_{\infty} C_{\Sigma}\right)^{\frac{2p}{2p+1}} t_{\text{mix}}^{5/4} p^{3/4} n^{-\frac{p}{2(2p+1)}} + C_2 n^{-\frac{p}{2p+1}}\left(\frac{16\mathcal{L}_Q \|\varepsilon\|_{\infty} C_{\Sigma}}{3}\right)^{\frac{2p}{2p+1}} pt_{\text{mix}} \,,$$

*where $\varkappa = (16/3)\|\varepsilon\|_{\infty}\mathcal{L}_Q t_{\text{mix}}$, constants $C_1$, $C_2$ are defined in Lemma 1, $\mathcal{L}_Q$ is defined in Lemma 3, and $C_{\Sigma}$ is defined in Lemma 10. Moreover, setting $p = \log n$, we obtain that*

$$\mathsf{d}_K\left(\frac{u^{\top}M}{\sqrt{n}\sigma_n(u)}\right) \leq \frac{C_{K,1}\log^{3/4}n}{n^{1/4}} + \frac{C_{K,2}\log n}{n^{1/2}} \,,$$

*where the constants $C_{K,1}$ and $C_{K,2}$ are given by*

$$C_{K,1} = 4\sqrt{2}e^{1/8}C_1 t_{\text{mix}}^{5/4}\mathcal{L}_Q \|\varepsilon\|_{\infty} C_{\Sigma} \,,$$

$$C_{K,2} = \frac{16e^{1/4}t_{\text{mix}}C_2\mathcal{L}_Q \|\varepsilon\|_{\infty} C_{\Sigma}}{3} + 3L(\varkappa)C_{\Sigma}^3 + 3L(\varkappa)\log 3C_{\Sigma}^3 \,.$$

*Proof of Proposition 6.* Using Lemma 1 with $s_n^2 = n\sigma_n^2(u)$, we obtain

$$\mathsf{d}_K\left(\frac{u^{\top}M}{\sqrt{n}\sigma_n(u)}\right) \leq \frac{L(\varkappa)(2n+1)\log(2n+1)}{n^{3/2}\sigma_n^3(u)}$$

$$+ C_1\sqrt{p}(n\sigma_n^2(u))^{-\frac{2p}{2p+1}}\left(\mathbb{E}|u^{\top}\langle M \rangle_n u - n\sigma_n^2(u)|^p\right)^{1/(2p+1)} + C_2(n\sigma_n^2(u))^{-\frac{p}{2p+1}} p\varkappa^{2p/(2p+1)} \,,$$

where the particular form of $\varkappa$ follows from (70). To complete the proof, it remains to control the $p$-th moment of $u^{\top}\langle M \rangle_n u - n\sigma_n^2(u)$, which is established in Lemma 2. Finally, we note that $\sigma_n^{-1}(u)$ is bounded, as follows from Lemma 10, which concludes the argument. $\qquad\square$

## B.2 Bound for the remainder term $D$

Recall that $D = D_1 + D_2$, where

$$D_1 = \frac{1}{\sqrt{n}} \sum_{k=0}^{n-1} \Gamma_{1:k}(\theta_0 - \theta^\star) + \frac{1}{\sqrt{n}} \sum_{k=1}^{n-1} H_k^{(0)} \varepsilon(Z_\ell) \,,$$

$$(27)$$

$$D_2 = -\frac{1}{\sqrt{n}} Q_1 \hat{\varepsilon}(Z_1) + \frac{1}{\sqrt{n}} Q_{n-1} \mathrm{P} \hat{\varepsilon}(Z_{n-1}) + \sum_{\ell=1}^{n-2} (Q_\ell - Q_{\ell+1}) \mathrm{P} \hat{\varepsilon}(Z_\ell) \,,$$

and $H_k^{(0)}$ are defined in (12). Then we obtain the following bound on the moments of $D$:

**Proposition 7.** *Assume A1, A2, and A3 and $k_0 \geq \max((\frac{24}{ac_0})^{1/(1-\gamma)}, c_0^{1/\gamma})$. Then for any $2 \leq p \leq \log n$, $u \in \mathbb{S}_{d-1}$, and any initial distribution $\xi$ on $(\mathsf{Z}, \mathcal{Z})$, it holds that*

$$\mathbb{E}_\xi^{1/(p+1)} \left[ |(u^\top D)/\sigma_n(u)|^p \right] \lesssim (\mathrm{C}_{1,1}^\mathsf{D} \|\theta_0 - \theta^\star\| + \mathrm{C}_{1,2}^\mathsf{D}) n^{p/(2p+2)} + \mathrm{C}_{1,3}^\mathsf{D} \{p^2 n^{1/2-\gamma}\}^{p/(p+1)}$$
$$+ \mathrm{C}_{1,4}^\mathsf{D} \{p^2 \sqrt{\log n} n^{1/2-\gamma}\}^{p/(p+1)} \,,$$

*where*

$$\mathrm{C}_{1,1}^\mathsf{D} = C_\Sigma C_\Gamma d^{1/\log n} \frac{k_0^\gamma}{ac_0(1-\gamma)} \,,$$

$$\mathrm{C}_{1,2}^\mathsf{D} = C_\Sigma \|\varepsilon\|_\infty \mathcal{L}_Q t_{\mathrm{mix}} \,,$$

$$\mathrm{C}_{1,3}^\mathsf{D} = \frac{C_\Sigma c_0 t_{\mathrm{mix}}}{1-\gamma} \left( \mathcal{L}_{Q,2} \|\varepsilon\|_\infty + \sqrt{\log \frac{k_0^\gamma}{c_0} \mathsf{D}_3^{(\mathrm{M})}} \right) \,,$$

$$\mathrm{C}_{1,4}^\mathsf{D} = \frac{C_\Sigma \mathsf{D}_3^{(\mathrm{M})} c_0 t_{\mathrm{mix}} \sqrt{\gamma}}{1-\gamma} \,.$$

*Moreover setting $p = \log n$ we have*

$$\mathbb{E}_\xi^{1/(p+1)} \left[ |(u^\top D)/\sigma_n(u)|^p \right] \lesssim (\mathrm{C}_1^\mathsf{D} \|\theta_0 - \theta^\star\| + \mathrm{C}_2^\mathsf{D}) n^{-1/2} + \mathrm{C}_3^\mathsf{D} (\log n)^2 n^{1/2-\gamma}$$
$$+ \mathrm{C}_4^\mathsf{D} (\log n)^{5/2} n^{1/2-\gamma} \,,$$

*and*

$$\mathrm{C}_i^\mathsf{D} = \sqrt{\mathrm{e}} \, \mathrm{C}_{1,i}^\mathsf{D} \qquad \text{for } i \in \{1, 2, 3, 4\}$$

*Proof.* We first note that $\sigma_n^{-1}(u)$ is bounded, as follows from Lemma 10, that is,

$$\mathbb{E}_\xi^{1/p} \left[ |(u^\top D)/\sigma_n(u)|^p \right] \leq C_\Sigma \mathbb{E}_\xi^{1/p} [|u^\top D|^p] \,.$$

In order to bound $\mathbb{E}_\xi^{1/p}[|u^\top D|^p]$, we bound the terms $D_1$ and $D_2$ from (27) separately. Note that

$$u^\top D_1 = D_{11} + D_{12} \,, \quad D_{11} = \frac{1}{\sqrt{n}} \sum_{k=0}^{n-1} u^\top \Gamma_{1:k}(\theta_0 - \theta^\star), \quad D_{12} = \frac{1}{\sqrt{n}} \sum_{k=1}^{n-1} u^\top H_k^{(0)} \,.$$

Now we use the result of Proposition 12 to bound $\mathbb{E}_\xi[\|\Gamma_{1:k}\|^p]$, and obtain that

$$\mathbb{E}_\xi^{1/p}[|D_{11}|^p] \lesssim C_\Gamma d^{1/\log n} \frac{1}{\sqrt{n}} \sum_{k=0}^{n-1} \exp\left\{ -(a/12) \sum_{j=1}^{k} \alpha_j \right\} \|\theta_0 - \theta^\star\| \,.$$

Applying Lemma 24 and Lemma 31,

$$\mathbb{E}_\xi^{1/p}[|D_{11}|^p] \lesssim C_\Gamma d^{1/\log n} \frac{1}{\sqrt{n}} \sum_{k=0}^{n-1} \exp\left\{ -\frac{ac_0}{24(1-\gamma)}((k+k_0)^{1-\gamma} - k_0^\gamma) \right\} \|\theta_0 - \theta^\star\| \quad (28)$$

$$\lesssim C_\Gamma d^{1/\log n} \frac{1}{\sqrt{n}} \frac{k_0^\gamma}{ac_0(1-\gamma)} \|\theta_0 - \theta^\star\| \,.$$

It follows from Minkowski's inequality and Proposition 10 that

$$
\mathbb{E}_\xi^{1/p}[|D_{12}|^p] \leq \frac{1}{\sqrt{n}} \sum_{k=1}^{n-1} \mathbb{E}_\xi^{1/p}[|u^\top H_k^{(0)}|^p] \leq \frac{\mathsf{D}_3^{(M)} t_{\mathrm{mix}} p^2}{\sqrt{n}} \sum_{k=1}^{n-1} \alpha_k \sqrt{\log(1/\alpha_k)}
$$

$$
\lesssim \frac{\mathsf{D}_3^{(M)} t_{\mathrm{mix}} p^2 c_0}{1-\gamma} \sqrt{\log \frac{(n+k_0-1)^\gamma}{c_0}} \frac{((n-1+k_0)^{1-\gamma} - k_0^{1-\gamma})}{\sqrt{n}} \tag{29}
$$

$$
\lesssim \frac{\mathsf{D}_3^{(M)} t_{\mathrm{mix}} p^2 c_0}{1-\gamma} \left(\sqrt{\gamma \log n} + \sqrt{\frac{k_0^\gamma}{c_0}}\right) n^{1/2-\gamma} .
$$

Similarly, $u^\top D_2 = D_{21} + D_{22} + D_{32}$, where

$$
D_{21} = -\frac{1}{\sqrt{n}} u^\top Q_1 \hat{\varepsilon}(Z_1) ,
$$

$$
D_{22} = \frac{1}{\sqrt{n}} u^\top Q_{n-1} \mathrm{P} \hat{\varepsilon}(Z_{n-1}) ,
$$

$$
D_{32} = \sum_{\ell=1}^{n-2} u^\top (Q_\ell - Q_{\ell+1}) \mathrm{P} \hat{\varepsilon}(Z_\ell) .
$$

For $D_{21}$ and $D_{22}$ we use Lemma 3 to bound $\|Q_\ell\|$ together with the upper bound $\|\hat{\varepsilon}(z)\| \leq (8/3) t_{\mathrm{mix}} \|\varepsilon\|_\infty$. Then we obtain

$$
\mathbb{E}_\xi^{1/p}[|D_{21}|^p] + \mathbb{E}_\xi^{1/p}[|D_{22}|^p] \lesssim \frac{t_{\mathrm{mix}} \|\varepsilon\|_\infty \mathcal{L}_Q}{\sqrt{n}} .
$$

It remains to bound $\mathbb{E}_\xi^{1/p}[|D_{23}|^p]$. Using Lemma 6 to bound the difference $Q_\ell - Q_{\ell+1}$, and Minkowski's inequality, we get

$$
\mathbb{E}_\xi^{1/p}[|D_{23}|^p] \lesssim \frac{\mathcal{L}_{Q,2} t_{\mathrm{mix}} \|\varepsilon\|_\infty}{\sqrt{n}} \sum_{\ell=2}^{n-1} \alpha_\ell \lesssim \frac{\mathcal{L}_{Q,2} t_{\mathrm{mix}} \|\varepsilon\|_\infty c_0}{1-\gamma} n^{1/2-\gamma} .
$$

It remains to combine the above bounds. $\qquad\square$

### B.3 Proof of Corollary 1

*Proof.* Let $\Phi(x)$ is the c.d.f. of the standard normal law $\mathcal{N}(0,1)$, $\Phi_\sigma(x)$ is the c.d.f. of the normal law $\mathcal{N}(0, \sigma(u))$ and $\Phi_{\sigma_n}(x)$ is the c.d.f. of the normal law $\mathcal{N}(0, \sigma_n(u))$. Then we have

$$
\begin{aligned}
\mathsf{d}_K\big(\sqrt{n} u^\top(\bar{\theta}_n - \theta^\star)/\sigma(u)\big) &= \sup_{x \in \mathbb{R}} |\mathbb{P}(\sqrt{n} u^\top(\bar{\theta}_n - \theta^\star)/\sigma(u) \leq x) - \Phi(x)| \\
&= \sup_{x \in \mathbb{R}} |\mathbb{P}(\sqrt{n} u^\top(\bar{\theta}_n - \theta^\star) \leq x) - \Phi_\sigma(x)| \\
&\leq \sup_{x \in \mathbb{R}} |\mathbb{P}(\sqrt{n} u^\top(\bar{\theta}_n - \theta^\star) \leq x) - \Phi_{\sigma_n}(x)| + \sup_{x \in \mathbb{R}} |\Phi_\sigma(x) - \Phi_{\sigma_n}(x)| \\
&= \mathsf{d}_K\big(\sqrt{n} u^\top(\bar{\theta}_n - \theta^\star)/\sigma_n(u)\big) + \sup_{x \in \mathbb{R}} |\Phi_\sigma(x) - \Phi_{\sigma_n}(x)|
\end{aligned}
$$

Using Lemma 9 to bound $|\sigma^2(u) - \sigma_n^2(u)|$ and Lemma 10 to bound $\sigma_n^{-2}(u)$ we obtain

$$
|\sigma^2(u)/\sigma_n^2(u) - 1| \leq \frac{C'_\infty C_\Sigma^2}{n^{1-\gamma}} .
$$

Setting

$$
C_\infty = \frac{3C'_\infty}{\lambda_{\min}(\Sigma_\infty)} , \text{ where } C'_\infty \text{ is defined in (31)} \tag{30}
$$

and applying the Gaussian comparison inequality (68) we conclude the proof. $\qquad\square$

## B.4 Technical bounds

**Lemma 3.** *Assume A1, A2, and A3. Then for any $\ell \in \mathbb{N}$ it holds that*

$$\|Q_\ell\| \leq \mathcal{L}_Q \,,$$

*where*

$$\mathcal{L}_Q = \kappa_Q^{1/2}\big(c_0 + \frac{4}{a(1-\gamma)}\big)$$

*and $\kappa_Q$ is defined in Proposition 2.*

*Proof.* Using the definition of $Q_\ell$ from (10) and Proposition 2, we get

$$\|Q_\ell\| \leq \kappa_Q^{1/2}\alpha_\ell \sum_{k=\ell}^{n-1} \|G_{\ell+1:k}\|_Q \leq \kappa_Q^{1/2}\alpha_\ell \sum_{k=\ell}^{n-1} \prod_{j=\ell+1}^{k} \sqrt{1 - a\alpha_j}$$

$$\leq \kappa_Q^{1/2}\alpha_\ell \sum_{k=\ell}^{n-1} \prod_{j=\ell+1}^{k} (1 - (a/2)\alpha_j) \leq \mathcal{L}_Q \,,$$

where in the last bound we applied Lemma 23 with $b = a/2$. $\qquad\square$

**Lemma 4.** *The following identity holds*

$$Q_\ell - \bar{\mathbf{A}}^{-1} = S_\ell - \bar{\mathbf{A}}^{-1}G_{\ell:n-1} \,, \text{ where } S_\ell = \sum_{j=\ell+1}^{n-1} (\alpha_\ell - \alpha_j)G_{\ell+1:j-1} \,,$$

*and*

$$\sum_{i=1}^{n-1}(Q_i - \bar{\mathbf{A}}^{-1}) = -\bar{\mathbf{A}}^{-1}\sum_{j=1}^{n-1} G_{1:j} \,.$$

*Proof.* See [82, pp. 26-30]. $\qquad\square$

**Lemma 5.** *Let $c_0 \in (0, \alpha_\infty]$ and $\ell \in \mathbb{N}$. Then under A1, A2, and A3, it holds*

$$\|S_\ell\| \leq \sqrt{\kappa_Q} \cdot C_{\gamma,\beta}^{(S)} \cdot (\ell + k_0)^{\gamma-1} \,,$$

*where*

$$C_{\gamma,a}^{(S)} = 2c_0 \exp\bigg\{\frac{ac_0}{2k_0^\gamma}\bigg\} \bigg(2^{\gamma/(1-\gamma)}\frac{2}{ac_0} + (\frac{2}{ac_0})^{1/(1-\gamma)}\Gamma(\frac{1}{1-\gamma})\bigg) \,.$$

*Proof.* For simplicity we define $m_i^j = \sum_{k=i}^{j}(k + k_0)^{-\gamma}$ and $\beta = a/2$. Note that

$$\|\sum_{j=i+1}^{n-1} (\alpha_i - \alpha_j)G_{i+1:j-1}^{(\alpha)}\| \leq \sqrt{\kappa_Q} \sum_{j=i}^{n-2} \frac{c_0}{(j + k_0 + 1)^\gamma}\bigg(\bigg(\frac{j + k_0 + 1}{i + k_0}\bigg)^\gamma - 1\bigg)\exp\{-\beta c_0 m_{i+1}^j\}$$

Following the proof of [82, Lemma A.7], we have

$$\bigg(\frac{j + k_0 + 1}{i + k_0}\bigg)^\gamma - 1 \leq (i + k_0)^{\gamma-1}\bigg(1 + (1 - \gamma)m_i^j\bigg)^{\gamma/(1-\gamma)}$$

Hence, we obtain

$$
\frac{\|S_i\|}{\sqrt{\kappa_Q}} \le c_0(i+k_0)^{\gamma-1} \sum_{j=i}^{n-2} \frac{1}{(j+k_0+1)^{\gamma}} \left(1+(1-\gamma)m_i^j\right)^{\gamma/(1-\gamma)} \exp\{-\beta c_0 m_{i+1}^j\}
$$

$$
\le c_0(i+k_0)^{\gamma-1} \sum_{j=i}^{n-2} \frac{1}{(j+k_0)^{\gamma}} \left(1+(1-\gamma)m_i^j\right)^{\gamma/(1-\gamma)} \exp\{\beta c_0(k_0+i)^{-\gamma}\} \exp\{-\beta c_0 m_i^j\}
$$

$$
\le c_0 \exp\{\frac{\beta c_0}{k_0^{\gamma}}\}(i+k_0)^{\gamma-1} \sum_{j=i}^{n-2} (m_i^j - m_i^{j-1}) \left(1+(1-\gamma)m_i^j\right)^{\gamma/(1-\gamma)} \exp\{-\beta c_0 m_i^j\}
$$

$$
\le 2c_0 \exp\{\frac{\beta c_0}{k_0^{\gamma}}\}(i+k_0)^{\gamma-1} \int_0^{+\infty} \left(1+(1-\gamma)m\right)^{\gamma/(1-\gamma)} \exp\{-\beta c_0 m\}\mathrm{d}m
$$

$$
\le 2c_0 \exp\{\frac{\beta c_0}{k_0^{\gamma}}\}(i+k_0)^{\gamma-1} \left(2^{\gamma/(1-\gamma)}\frac{1}{\beta c_0} + (\frac{1}{\beta c_0})^{1/(1-\gamma)}\Gamma(\frac{1}{1-\gamma})\right) .
$$

$\qquad\qquad\qquad\qquad\qquad\qquad\qquad\qquad\qquad\qquad\qquad\qquad\qquad\qquad\qquad\qquad\qquad\qquad\qquad$ $\square$

**Lemma 6.** *Let $c_0 \in (0, \alpha_\infty]$ and $\ell \in \mathbb{N}$. Then under A1, A2, and A3, it holds*

$$
\|Q_{\ell+1} - Q_\ell\| \le \mathcal{L}_{Q,2} \cdot \alpha_{\ell+1} ,
$$

*where*

$$
\mathcal{L}_{Q,2} = \sqrt{\kappa_Q}\left(2^{\gamma} + (2\,\mathrm{C_A} + a/4)(c_0 + \frac{4}{a(1-\gamma)})\right) .
$$

*Proof.* Using the definition of $Q_\ell$ we have

$$
Q_{\ell+1} - Q_\ell = \alpha_{\ell+1} \sum_{k=l+1}^{n-1} G_{\ell+2:k} - \alpha_\ell \sum_{k=l}^{n-1} G_{\ell+1:k} = \alpha_{\ell+1} \sum_{k=l+1}^{n-1} G_{\ell+2:k+1} - \alpha_\ell \sum_{k=l+1}^{n-1} G_{\ell+1:k} - \alpha_\ell
$$

$$
= -\alpha_\ell + \sum_{k=l+1}^{n-1} G_{l+2:k}(\alpha_{\ell+1}I - \alpha_\ell I + \alpha_{\ell+1}\alpha_\ell \bar{\mathbf{A}}) .
$$

Hence, using Lemma 27 with $r = a/4$ and Lemma 3 we get

$$
\|Q_{\ell+1} - Q_\ell\| \le \alpha_\ell + \sum_{k=l+1}^{n-1} \alpha_{\ell+1}^2 \|G_{l+2:k}\|(2\,\mathrm{C_A}+a/4) \le (2^{\gamma} + \mathcal{L}_Q(2\,\mathrm{C_A}+a/4))\alpha_{\ell+1} .
$$

$\qquad\qquad\qquad\qquad\qquad\qquad\qquad\qquad\qquad\qquad\qquad\qquad\qquad\qquad\qquad\qquad\qquad\qquad\qquad\qquad$ $\square$

**Lemma 7.** *Let $c_0 \in (0, \alpha_\infty]$, $k_0^{1-\gamma} > bc_0$ and $\ell \in \mathbb{N}$. Then under A1, A2, and A3 it holds*

$$
\|S_{\ell+1} - S_\ell\| \le \sqrt{\kappa_Q} \cdot C_{\gamma,a}^{(S,2)} \cdot \alpha_{\ell+1} ,
$$

*where*

$$
C_{\gamma,a}^{(S,2)} = (c_0 + \frac{4}{a(1-\gamma)})(a/2 + 3\,\mathrm{C_A}) .
$$

*Proof.* Using Lemma 4 we obtain

$$
S_{\ell+1} - S_\ell = Q_{\ell+1} - Q_\ell + \bar{\mathbf{A}}^{-1}(G_{\ell+1:n-1} - G_{\ell:n-1})
$$

$$
= \alpha_{\ell+1} \sum_{k=l+1}^{n-1} G_{\ell+2:k} - \alpha_\ell \sum_{k=l}^{n-1} G_{\ell+1:k} + \bar{\mathbf{A}}^{-1}(I - (I - \alpha_\ell\bar{\mathbf{A}}))G_{\ell+1:n-1}
$$

$$
= \alpha_{\ell+1} \sum_{k=l}^{n-2} G_{\ell+2:k+1} - \alpha_\ell \sum_{k=l}^{n-2} G_{\ell+1:k}
$$

$$
= (\alpha_{\ell+1} - \alpha_\ell)I + \sum_{k=l+1}^{n-2} (\alpha_{\ell+1}(I - \alpha_{k+1}\bar{\mathbf{A}}) - \alpha_\ell(I - \alpha_{\ell+1}\bar{\mathbf{A}}))G_{l+2:k} .
$$

Hence, using Lemma 27 with $r = a/4$ we get

$$\|S_{\ell+1} - S_\ell\| \leq (a/4)\alpha_{\ell+1}^2 + \sum_{k=l+1}^{n-2} ((a/4)\alpha_{\ell+1}^2 + C_{\mathbf{A}}\,\alpha_{\ell+1}^2 + 2^\gamma\,C_{\mathbf{A}}\,\alpha_{\ell+1}^2)\|G_{l+2:k}\|$$

$$\leq (\mathcal{L}_Q((a/4) + C_{\mathbf{A}} + 2^\gamma\,C_{\mathbf{A}}) + (a/4))\alpha_{\ell+1}\,,$$

where in the last inequality we applying lemma 3. Since $\mathcal{L}_Q \geq 1$ and $2^\gamma < 2$ we complete the proof. $\qquad\square$

**Lemma 8.** *Let $c_0 \in (0, \alpha_\infty]$. Then under A2 for any $m \in \mathbb{N}$ it holds*

$$\sum_{t=1}^{n-1} \|G_{t:n-1}\|^m \leq \frac{\kappa_Q^{m/2}}{1 - (1 - c_0(a/2)(n + k_0 - 2)^{-\gamma})^m}$$

*Proof.* Note that

$$\sum_{t=1}^{n-1} \|G_{t:n-1}\|^m \leq \kappa_Q^{m/2} \sum_{l=1}^{n-1} \prod_{i=t}^{n-1} (1 - (a/2)\alpha_i)$$

$$= \frac{\kappa_Q^{m/2}}{(1 - (1 - (a/2)\alpha_{n-2})^m)} \sum_{l=1}^{n-1} (1 - (1 - (a/2)\alpha_{t-1})^m) \prod_{i=t}^{n-1} (1 - (a/2)\alpha_i)^m$$

$$\leq \frac{\kappa_Q^{m/2}}{(1 - (1 - (a/2)\alpha_{n-2})^m)}\,.$$

$\qquad\square$

**Lemma 9.** *Let $c_0 \in (0, \alpha_\infty]$. Then under A1, A2, and A3 it holds*

$$|\sigma_n^2(u) - \sigma^2(u)| \leq C'_\infty n^{\gamma-1}\,,$$

*where*

$$C'_\infty = \|\Sigma_\varepsilon\| \frac{\kappa_Q (C_{\gamma,a}^{(S)})^2}{2\gamma - 1} + \|\Sigma_\infty\| \left( \frac{2^{\gamma+1} 5 + 3}{ac_0(1 - \gamma)} k_0^\gamma + \frac{4\,C_{\mathbf{A}}\,\kappa_Q C_{\gamma,a}^{(S)}}{ac_0} k_0^{2\gamma-1} \right) \qquad (31)$$

*Proof.* Note that

$$|\sigma_n^2(u) - \sigma^2(u)| \leq \|\Sigma_n - \Sigma_\infty\|\,.$$

Then, we express the $\Sigma_n - \Sigma_\infty$ in the following form:

$$\Sigma_n - \Sigma_\infty = \underbrace{\frac{1}{n} \sum_{t=2}^{n-1} (Q_t - \bar{\mathbf{A}}^{-1})\Sigma_\varepsilon \bar{\mathbf{A}}^{-\top} + \frac{1}{n} \sum_{t=2}^{n-1} \bar{\mathbf{A}}^{-1}\Sigma_\varepsilon (Q_t - \bar{\mathbf{A}}^{-1})^\top}_{R_1}$$

$$+ \underbrace{\frac{1}{n} \sum_{t=2}^{n-1} (Q_t - \bar{\mathbf{A}}^{-1})\Sigma_\varepsilon (Q_t - \bar{\mathbf{A}}^{-1})^\top}_{R_2} - \frac{2}{n}\Sigma_\infty\,.$$

First, we bound $R_1$, using Lemma 23 we obtain

$$\|\frac{1}{n} \sum_{t=2}^{n-1} (Q_t - \bar{\mathbf{A}}^{-1})\Sigma_\varepsilon \bar{\mathbf{A}}^{-\top}\| = \|\frac{1}{n}\bar{\mathbf{A}}^{-1} \sum_{j=2}^{n-1} G_{1:j}\Sigma_\varepsilon \bar{\mathbf{A}}^{-\top}\|$$

$$\leq \|n^{-1}\Sigma_\infty \sum_{j=2}^{n-1} G_{1:j}\| \leq n^{-1}\|\Sigma_\infty\| \cdot \sum_{j=1}^{n-1} \|G_{1:j}\|$$

$$\leq n^{-1}\|\Sigma_\infty\|(1 + \frac{4}{ac_0(1 - \gamma)})(1 + k_0)^\gamma$$

$$\leq 2^\gamma n^{\gamma-1}\|\Sigma_\infty\|(1 + \frac{4}{ac_0(1 - \gamma)})k_0^\gamma\,.$$

Hence, we get that

$$\|R_1\| \leq 2^{\gamma+1} n^{\gamma-1} \|\Sigma_\infty\| (1 + \frac{4}{ac_0(1-\gamma)}) k_0^\gamma .$$

Now, we rewrite term $R_2$ as follows:

$$n^{-1} \sum_{t=2}^{n-1} (Q_t - \bar{\mathbf{A}}^{-1}) \Sigma_\varepsilon (Q_t - \bar{\mathbf{A}}^{-1})^\top$$

$$= n^{-1} \sum_{t=2}^{n-1} \left( S_t - \bar{\mathbf{A}}^{-1} \prod_{k=t}^{n-1} (\mathbf{I} - \alpha_k \bar{\mathbf{A}}) \right) \Sigma_\varepsilon \left( S_t - \bar{\mathbf{A}}^{-1} \prod_{k=t}^{n-1} (\mathbf{I} - \alpha_k \bar{\mathbf{A}}) \right)^\top$$

$$= \underbrace{n^{-1} \sum_{t=2}^{n-1} S_t \Sigma_\varepsilon S_t^\top}_{R_{21}} + \underbrace{n^{-1} \sum_{t=2}^{n-1} \bar{\mathbf{A}}^{-1} \prod_{k=t}^{n-1} (\mathbf{I} - \alpha_k \bar{\mathbf{A}}) \Sigma_\varepsilon \bar{\mathbf{A}}^{-\top} \prod_{k=t}^{n-1} (\mathbf{I} - \alpha_k \bar{\mathbf{A}})^\top}_{R_{22}}$$

$$\underbrace{- n^{-1} \sum_{t=2}^{n-1} \bar{\mathbf{A}}^{-1} \prod_{k=t}^{n-1} (\mathbf{I} - \alpha_k \bar{\mathbf{A}}) \cdot \Sigma_\varepsilon S_t^\top}_{R_{23}} \underbrace{- n^{-1} \sum_{t=2}^{n-1} S_t \Sigma_\varepsilon \bar{\mathbf{A}}^{-\top} \prod_{k=t}^{n-1} (\mathbf{I} - \alpha_k \bar{\mathbf{A}})^\top}_{R_{24}} .$$

To bound $\|R_{21}\|$ we use Lemma 5 and obtain

$$\|R_{21}\| = \|n^{-1} \sum_{t=2}^{n-1} S_t \Sigma_\varepsilon S_t^\top\| \leq n^{-1} \sum_{t=2}^{n-1} \|\Sigma_\varepsilon\| \|S_t\|^2$$

$$\leq n^{-1} \|\Sigma_\varepsilon\| \sum_{t=2}^{n-1} \kappa_Q \left( C_{\gamma,a}^{(S)} \right)^2 (t + k_0)^{2(\gamma-1)}$$

$$\leq n^{-1} \|\Sigma_\varepsilon\| \kappa_Q \left( C_{\gamma,a}^{(S)} \right)^2 \frac{(n + k_0 - 1)^{2\gamma-1} - (k_0 + 1)^{2\gamma-1}}{2\gamma - 1}$$

$$\leq n^{2(\gamma-1)} \frac{\|\Sigma_\varepsilon\| \kappa_Q \left( C_{\gamma,a}^{(S)} \right)^2}{2\gamma - 1} ,$$

where the last inequality holds since $(n + k_0 - 1)^{2\gamma-1} \leq n^{2\gamma-1} + (k_0 + 1)^{2\gamma-1}$ for $\gamma \in (1/2, 1)$.
The bound for $R_{22}$ follows from Lemma 8 and simple inequality $(n + k_0 - 2)^\gamma \leq (k_0 n)^\gamma$:

$$\|R_{22}\| = \|n^{-1} \sum_{i=2}^{n-1} G_{i:n-1} \bar{\mathbf{A}}^{-1} \Sigma_\varepsilon \bar{\mathbf{A}}^{-\top} G_{i:n-1}^\top\| \leq n^{-1} \|\Sigma_\infty\| \sum_{i=1}^{n-1} \|G_{i:n-1}\|^2$$

$$\leq n^{-1} \frac{\|\Sigma_\infty\|}{c_0 a (n + k_0 - 2)^{-\gamma} - c_0^2 (a^2/4)(n + k_0 - 2)^{-2\gamma}} \leq 2 \|\Sigma_\infty\| k_0^\gamma \frac{n^{\gamma-1}}{c_0 a} .$$

Since $R_{23} = R_{24}^\top$, we concentrate on $\|D_{24}\|$. Lemma 8 immediately imply

$$\|R_{24}\| \leq n^{-1} \|\Sigma_\varepsilon \bar{\mathbf{A}}^{-\top}\| \sum_{i=1}^{n-1} \|S_i\| \|G_{i:n-1}\|$$

$$\leq n^{-1} \|\bar{\mathbf{A}} \Sigma_\infty\| \kappa_Q C_{\gamma,a}^{(S)} \sum_{i=1}^{n-1} (i + k_0)^{\gamma-1} \prod_{k=i}^{n-1} (1 - \frac{ac_0}{2(k + k_0)^\gamma})$$

$$\leq n^{-1} \|\bar{\mathbf{A}} \Sigma_\infty\| \kappa_Q C_{\gamma,a}^{(S)} \sum_{i=1}^{n-1} (i + k_0)^{2\gamma-1} (i + k_0)^{-\gamma} \prod_{k=i+1}^{n-1} (1 - \frac{ac_0}{2(k + k_0)^\gamma})$$

$$\leq C_{\mathbf{A}} \|\Sigma_\infty\| \kappa_Q C_{\gamma,a}^{(S)} k_0^{2\gamma-1} \frac{2n^{2(\gamma-1)}}{ac_0}$$

By combining all the inequalities, we complete the proof. $\qquad\square$

**Lemma 10.** *Let $c_0 \in (0, \alpha_\infty]$ and $C'_\infty n^{\gamma-1} \leq \lambda_{\min}(\Sigma_\infty)/2$. Then under A1, A2, and A3 it holds*

$$\sigma_n^{-1}(u) \leq C_\Sigma \,,$$

*where $C_\Sigma = \sqrt{\frac{2}{\lambda_{\min}(\Sigma_\infty)}}$.*

*Proof.* Note that $\sigma_n^2(u) > \lambda_{\min}(\Sigma_n)$. Using Lidskii's inequality, we obtain

$$\lambda_{\min}(\Sigma_n) \geq \lambda_{\min}(\Sigma_\infty) - \|\Sigma_n - \Sigma_\infty\| \geq \lambda_{\min}(\Sigma_\infty)/2 \,,$$

where in the last inequality we use $C'_\infty n^{\gamma-1} \leq \lambda_{\min}(\Sigma_\infty)/2$. $\qquad\square$

## C  Last iterate bound

The main result of this section is the following bound on the $p$-th moment of the LSA iterate $\theta_k - \theta^\star$.

**Proposition 8.** *Assume A1, A2, and A3. Then for any $2 \leq p \leq \log n$, $k \geq 1$, $u \in \mathbb{S}_{d-1}$, and any initial distribution $\xi$ on $(\mathsf{Z}, \mathcal{Z})$, it holds that*

$$\mathbb{E}_\xi^{1/p}\left[|u^\top(\theta_k - \theta^\star)|^p\right] \lesssim \mathsf{D}_1^{(\mathrm{M})} t_{\mathrm{mix}} \sqrt{p\alpha_k} + C_\Gamma d^{1/\log n} \exp\left\{-(a/12) \sum_{\ell=1}^{k} \alpha_\ell\right\} \|\theta_0 - \theta^\star\| \,, \quad (32)$$

*where the constant $\mathsf{D}_1^{(\mathrm{M})}$ is given by*

$$\mathsf{D}_1^{(\mathrm{M})} = d^{1/2+1/\log n}\left(C_\Gamma \, \mathsf{C_A} \, \mathsf{D}_2^{(\mathrm{M})}/a\right)\|\varepsilon\|_\infty + \|\varepsilon\|_\infty(\kappa_Q/a)^{1/2}(3 + 4\,\mathsf{C_A}/a) \,,$$

*and $\mathsf{D}_2^{(\mathrm{M})}$ is defined in (37).*

We provide the result above only for 1-dimensional projections of the error $u^\top(\theta_n - \theta^\star)$, as previously considered in [47]. Note that the scaling of the right-hand side of (32) with a $t_{\mathrm{mix}}$ factor can be suboptimal. However, this scaling corresponds to the second-order (in $n$) terms. Tighter analysis of this term using the Rosenthal-type inequality should reveal a $\sqrt{t_{\mathrm{mix}}}$ dependence of the leading term, however, we leave this improvement for future work.

Our proof of the last iterate bound is based on the perturbation-expansion framework [2], see also [24]. Within this framework, we expand the error recurrence (4), using the notation $\Gamma_{m:k}^{(\alpha)}$ for the product of random matrices:

$$\Gamma_{m:k}^{(\alpha)} = \prod_{i=m}^{k}(\mathrm{I} - \alpha_i \mathbf{A}(Z_i)) \,, \quad m, k \in \mathbb{N}, \quad m \leq k \,, \quad (33)$$

with the convention that $\Gamma_{m:k}^{(\alpha)} = \mathrm{I}$ for $m > k$. Using the recurrence (4), we arrive at the following decomposition of the LSA error:

$$u^\top(\theta_k - \theta^\star) = u^\top\tilde{\theta}_k^{(\mathrm{tr})} + u^\top\tilde{\theta}_k^{(\mathrm{fl})} \,, \quad (34)$$

where we have defined

$$\tilde{\theta}_k^{(\mathrm{tr})} = u^\top\Gamma_{1:k}^{(\alpha)}\{\theta_0 - \theta^\star\} \,, \quad \tilde{\theta}_k^{(\mathrm{fl})} = -\sum_{j=1}^{k} u^\top\Gamma_{j+1:k}^{(\alpha)}\alpha_j\varepsilon(Z_j) \,.$$

Here $\tilde{\theta}_k^{(\mathrm{tr})}$ is a transient term (reflecting the forgetting of the initial condition) and $\tilde{\theta}_k^{(\mathrm{fl})}$ is a fluctuation term (reflecting misadjustement noise). We treat the $\tilde{\theta}_k^{(\mathrm{tr})}$ and $\tilde{\theta}_k^{(\mathrm{fl})}$ terms separately. In particular, we control $\tilde{\theta}_k^{(\mathrm{tr})}$ using Proposition 12. For estimating $\tilde{\theta}_k^{(\mathrm{fl})}$ we use the decomposition

$$u^\top\tilde{\theta}_k^{(\mathrm{fl})} = u^\top J_k^{(0)} + u^\top H_k^{(0)} \,, \quad (35)$$

where the latter terms are defined by the following pair of recursions

$$J_k^{(0)} = \left(\mathrm{I} - \alpha_k \bar{\mathbf{A}}\right) J_{k-1}^{(0)} - \alpha_k\varepsilon(Z_k) \,, \qquad\qquad J_0^{(0)} = 0 \,,$$

$$H_k^{(0)} = (\mathrm{I} - \alpha_k \mathbf{A}(Z_k)) H_{k-1}^{(0)} - \alpha_k\tilde{\mathbf{A}}(Z_k)J_{k-1}^{(0)} \,, \qquad H_0^{(0)} = 0 \,.$$

For notation convenience we introduce, for $m \leq k$, the deterministic counterpart of the product of random matrices (33), that is,

$$G_{m:k} = \prod_{i=m}^{k} (I - \alpha_i \bar{\mathbf{A}}) ,$$

keeping the convention $G_{m:k} = I$ if $m > k$. Thus we obtain that

$$J_k^{(0)} = -\sum_{j=1}^{k} \alpha_j G_{j+1:k} \varepsilon(Z_j), \quad H_k^{(0)} = -\sum_{j=1}^{k} \alpha_j \Gamma_{j+1:k} \tilde{\mathbf{A}}(Z_j) J_{j-1}^{(0)} , \tag{36}$$

and we analyze these terms above separately. We first bound the term $J_n^{(0)}$:

**Proposition 9.** *Assume A 1, A 2, and A 3. Let $k_0 \geq \{\frac{16\gamma}{ac_0}\}^{1/1-\gamma}$. Then, for any $p \geq 2$, initial probability measure $\xi$ on $(\mathsf{Z}, \mathcal{Z})$, and $k \geq 1$, it holds that*

$$\mathbb{E}_\xi^{1/p}\big[|u^\top J_k^{(0)}|^p\big] \leq \mathsf{D}_2^{(\mathrm{M})} t_{\mathrm{mix}} \sqrt{p \alpha_k} ,$$

*where*

$$\mathsf{D}_2^{(\mathrm{M})} = (32/3)\|\varepsilon\|_\infty (\kappa_Q/a)^{1/2}(3 + 4\,\mathsf{C}_{\mathbf{A}}/a) . \tag{37}$$

*Proof.* Note that $J_k^{(0)}$ is an additive functional of $\{\varepsilon(Z_j)\}_{j=1}^k$. Using the representation (36), we obtain that

$$u^\top J_k^{(0)} = -\sum_{j=1}^{k} \alpha_j u^\top G_{j+1:k} \varepsilon(Z_j) .$$

Applying Proposition 2, we get that

$$\|G_{j+1:k}\| \leq \kappa_Q^{1/2} \prod_{\ell=j+1}^{k} \sqrt{1 - a\alpha_\ell} .$$

Using Lemma 20 with $A_j = \alpha_j G_{j+1:k}$, we obtain that

$$\mathbb{E}_\xi^{1/p}[|u^T J_k^{(0)}|^p] \leq (16/3)p^{1/2}t_{\mathrm{mix}}\|\varepsilon\|_\infty \Big(\sum_{j=2}^{k} \alpha_j^2 \|G_{j+1:k}\|^2\Big)^{1/2}$$

$$+ (8/3)t_{\mathrm{mix}}\Big(\alpha_1\|G_{2:k}\| + \alpha_k + \sum_{j=1}^{k-1} \|\alpha_j G_{j+1:k} - \alpha_{j-1}G_{j:k}\|\Big)\|\varepsilon\|_\infty .$$

Using Lemma 28 with $b = a$, we get

$$\sum_{j=1}^{k} \alpha_j^2 \|G_{j+1:k}\|^2 \leq \kappa_Q \sum_{j=1}^{k} \alpha_j^2 \prod_{\ell=j+1}^{k} (1 - a\alpha_\ell) \leq 4(\kappa_Q/a)\alpha_k .$$

Applying Lemma 26 with $b = a/2$, we get $\alpha_1\|G_{2:k}\| \leq \alpha_k$. Moreover, using Lemma 27 with $r = a/4$, we get that

$$\|\alpha_j G_{j+1:k} - \alpha_{j-1}G_{j:k}\| = \|G_{j+1:k}(\alpha_j I - \alpha_{j-1}I + \alpha_{j-1}\alpha_j \bar{\mathbf{A}})\| \leq \alpha_j^2 \|G_{j+1:k}\|(2\,\mathsf{C}_{\mathbf{A}} + a/4) .$$

Then, applying Lemma 28 with $b = a/2$, we get

$$\alpha_1\|G_{2:k}\| + \alpha_k + \sum_{j=2}^{k} \|\alpha_j G_{j+1:k} - \alpha_{j-1}G_{j:k}\| \leq 2\alpha_k + \kappa_Q^{1/2}(a/4 + 2\,\mathsf{C}_{\mathbf{A}})\sum_{j=1}^{k} \alpha_j^2 \prod_{l=j+1}^{k} (1 - a\alpha_\ell/2)$$

$$\leq \alpha_k(2 + 8\kappa_Q^{1/2}(a/4 + 2\,\mathsf{C}_{\mathbf{A}})/a) .$$

Combining the above results and using that $p \geq 2$ yields (37). □

## C.1 Proof of Proposition 8

Proceeding as in (34) and (35), we obtain that

$$\mathbb{E}_\xi^{1/p}\left[|u^\top(\theta_k - \theta^\star)|^p\right] \leq \mathbb{E}_\xi^{1/p}\left[|u^\top\Gamma_{1:k}^{(\alpha)}\{\theta_0 - \theta^\star\}|^p\right] + \mathbb{E}_\xi^{1/p}\left[|u^\top J_k^{(0)}|^p\right] + \mathbb{E}_\xi^{1/p}\left[|u^\top H_k^{(0)}|^p\right] .$$

The first two terms are bounded using Proposition 12 and Proposition 9, respectively. In order to bound the term, corresponding to $H_k^{(0)}$, we apply Minkowski's inequality together with Hölder's inequlity to (36):

$$\mathbb{E}_\xi^{1/p}\left[|u^\top H_k^{(0)}|^p\right] \leq \sum_{j=1}^k \alpha_j \, \mathsf{C_A}\{\mathbb{E}_\xi\left[\|\Gamma_{j+1:k}^{(\alpha)}\|^{2p}\right]\}^{1/2p}\{\mathbb{E}_\xi\left[|J_{j-1}^{(0)}|^{2p}\right]\}^{1/2p} .$$

Note that

$$\{\mathbb{E}_\xi\left[\|J_{j-1}^{(0)}\|^{2p}\right]\}^{1/p} = \{\mathbb{E}_\xi\left[(\sum_{r=1}^d |e_r^T J_{j-1}^{(0)}|^2)^p\right]\}^{1/p}$$
$$\leq \sum_{r=1}^d \{\mathbb{E}_\xi\left[|e_r^T J_{j-1}^{(0)}|^{2p}\right]\}^{1/p} . \tag{38}$$

Using Proposition 9 and proposition 12 with a simple inequality $\mathrm{e}^{-x} \leq 1 - x/2$, valid for $x \in [0,1]$, we get

$$\mathbb{E}_\xi^{1/p}\left[|u^\top H_k^{(0)}|^p\right] \lesssim C_\Gamma \, \mathsf{C_A} \, \mathsf{D}_2^{(\mathrm{M})}\|\varepsilon\|_\infty t_{\mathrm{mix}}\sqrt{p}d^{1/2+1/\log n}\sum_{j=1}^k \alpha_j^{3/2} \prod_{\ell=j+1}^k (1 - a\alpha_\ell/24)$$

$$\lesssim d^{1/2+1/\log n}\left(C_\Gamma \, \mathsf{C_A} \, \mathsf{D}_2^{(\mathrm{M})}/a\right)\|\varepsilon\|_\infty t_{\mathrm{mix}}\sqrt{p\alpha_k} ,$$

where in the last inequality we use Lemma 28.

## C.2 Proof of $J_n^{(1)}$ bound

In order to proceed further, we need to obtain the tighter bound on $H_k^{(0)}$, which requires a tighter moment bound on the quality $J_k^{(1)}$.

**Proposition 10.** *Assume A1, A2, and A3. Then, for any $p \geq 2$, initial probability measure $\xi$ on $(\mathsf{Z}, \mathcal{Z})$, and $k \geq 1$, it holds that*

$$\mathbb{E}_\xi^{1/p}\left[|u^\top H_k^{(0)}|^p\right] \lesssim \mathsf{D}_3^{(\mathrm{M})}t_{\mathrm{mix}}p^2\alpha_k\sqrt{\log(1/\alpha_k)} ,$$

*where*

$$\mathsf{D}_3^{(\mathrm{M})} = \mathsf{D}_4^{(\mathrm{M})}(1 + \frac{d^{1/2+1/\log n}C_\Gamma \, \mathsf{C_A}}{a}) ,$$

*and the constant $\mathsf{D}_4^{(\mathrm{M})}$ is defined in (42).*

*Proof.* It is known (see e.g. [2] and [24]), that the term $H_k^{(0)}$ can be further decomposed as follows:

$$H_k^{(0)} = \sum_{\ell=1}^L J_k^{(\ell)} + H_k^{(L)} . \tag{39}$$

Here the parameter $L \geq 1$ control the depth of expansion, and the terms $J_k^{(\ell)}$ and $H_k^{(\ell)}$ are given by the following recurrences:

$$J_k^{(\ell)} = \left(\mathrm{I} - \alpha_k\bar{\mathbf{A}}\right) J_{k-1}^{(\ell)} - \alpha_k\tilde{\mathbf{A}}(Z_k)J_{k-1}^{(\ell-1)} , \qquad J_0^{(\ell)} = 0 ,$$
$$H_k^{(\ell)} = (\mathrm{I} - \alpha_k\mathbf{A}(Z_k)) H_{k-1}^{(\ell)} - \alpha_k\tilde{\mathbf{A}}(Z_k)J_{k-1}^{(\ell)} , \qquad H_0^{(\ell)} = 0 . \tag{40}$$

The expansion depth $L$ here controls the desired approximation accuracy. For our further results it is enough to take $L = 1$ and estimate the respective terms $J_k^{(1)}$ and $H_k^{(1)}$. Now the rest of the proof follows from Proposition 11 (see the bounds (41) and (43) for $\mathbb{E}_\xi^{1/p}\left[|u^\top J_k^{(1)}|^p\right]$ and $\mathbb{E}_\xi^{1/p}\left[|u^\top H_k^{(1)}|^p\right]$, respectively), and Minkoski's inequality. $\qquad\square$

**Proposition 11.** *Assume A1, A2, and A3. Then, for any $p \geq 2$, initial probability measure $\xi$ on $(\mathsf{Z}, \mathcal{Z})$, and $k \geq 1$, it holds that*

$$\mathbb{E}_\xi^{1/p}\left[|u^\top J_k^{(1)}|^p\right] \lesssim \mathsf{D}_4^{(\mathrm{M})} t_{\mathrm{mix}} p^2 \alpha_k \sqrt{\log(1/\alpha_k)}, \tag{41}$$

*where*

$$\mathsf{D}_4^{(\mathrm{M})} = \frac{\kappa_Q^{3/2}\, \mathrm{C_A}\, \|\varepsilon\|_\infty}{a} + \frac{\kappa_Q^{3/2}\, \mathrm{C_A}\, \|\varepsilon\|_\infty}{a^{3/2}} 3^{\gamma/2} + \frac{\kappa_Q^{3/2}\, \mathrm{C_A}\, \|\varepsilon\|_\infty}{a} 3^\gamma \left(\frac{8\gamma}{ac_0}\right)^{\gamma/(1-\gamma)}. \tag{42}$$

*Moreover,*

$$\mathbb{E}_\xi^{1/p}\left[|u^\top H_k^{(1)}|^p\right] \lesssim \mathsf{D}_5^{(\mathrm{M})} t_{\mathrm{mix}} p^2 \alpha_k \sqrt{\log(1/\alpha_k)}, \tag{43}$$

*where*

$$\mathsf{D}_5^{(\mathrm{M})} = \frac{d^{1/2+1/\log n} \mathsf{D}_4^{(\mathrm{M})} C_\Gamma\, \mathrm{C_A}}{a}. \tag{44}$$

*Proof.* Solving the recursion in (40) yields the double summation:

$$J_k^{(1)} = -\sum_{\ell=1}^k \alpha_\ell G_{\ell+1:k}\tilde{\mathbf{A}}(Z_\ell)J_{\ell-1}^{(0)} = \sum_{\ell=1}^k \alpha_\ell \sum_{j=1}^{\ell-1} \alpha_j G_{\ell+1:k}\tilde{\mathbf{A}}(Z_\ell)G_{j+1:\ell-1}\varepsilon(Z_j).$$

Changing the order of summation yields

$$J_k^{(1)} = \sum_{j=1}^{k-1} \alpha_j \left\{\sum_{\ell=j+1}^k \alpha_\ell G_{\ell+1:k}\tilde{\mathbf{A}}(Z_\ell)G_{j+1:\ell-1}\right\}\varepsilon(Z_j) = \sum_{j=1}^{k-1} \alpha_j S_{j+1:k}\varepsilon(Z_j),$$

where for $j \leq k$ we have defined

$$S_{j:k} := \sum_{\ell=j}^k \alpha_\ell G_{\ell+1:k}\tilde{\mathbf{A}}(Z_\ell)G_{j:\ell-1}.$$

Fix a constant $m \in \mathbb{N}$, $m \leq k$ (to be determined later). Then we can rewrite $S_{j+1:k}$ as

$$S_{j+1:k} = \sum_{\ell=j+1}^{j+m} \alpha_\ell G_{\ell+1:k}\tilde{\mathbf{A}}(Z_\ell)G_{j+1:\ell-1} + \sum_{\ell=j+m+1}^k \alpha_\ell G_{\ell+1:k}\tilde{\mathbf{A}}(Z_\ell)G_{j+1:\ell-1}$$

$$= G_{j+m+1:k}S_{j+1:j+m} + S_{j+m+1:k}G_{j+1:j+m}.$$

Let $N := \lfloor k/m \rfloor$. In these notations, we can express $J_k^{(1)}$ as a sum of three terms:

$$J_k^{(1)} = \underbrace{\sum_{j=1}^{m(N-1)} \alpha_j G_{j+m+1:k}S_{j+1:j+m}\varepsilon(Z_j)}_{T_1} + \underbrace{\sum_{j=1}^{m(N-1)} \alpha_j S_{j+m+1:k}G_{j+1:j+m}\varepsilon(Z_j)}_{T_2}$$

$$+ \underbrace{\sum_{j=m(N-1)+1}^{k-1} \alpha_j S_{j+1:k}\varepsilon(Z_j)}_{T_3}.$$

Consider the first term $T_1$. Applying Minkowski's inequality, Lemma 11, and Lemma 28, we get that

$$\mathbb{E}_\xi^{1/p}\left[|u^\top T_1|^p\right] \stackrel{(a)}{\lesssim} \kappa_Q^{3/2}\, \mathrm{C_A}\, p^{1/2}t_{\mathrm{mix}}^{1/2}\|\varepsilon\|_\infty \sum_{j=1}^{(m-1)N} \alpha_j \Big(\sum_{r=j+1}^{j+m} \alpha_r^2\Big)^{1/2} \prod_{\ell=j+1}^k \sqrt{1-a\alpha_\ell}$$

$$\lesssim \kappa_Q^{3/2}\, \mathrm{C_A}\, \sqrt{mpt_{\mathrm{mix}}}\|\varepsilon\|_\infty \sum_{j=1}^{(m-1)N} \alpha_j^2 \prod_{\ell=j+1}^k \sqrt{1-a\alpha_\ell}$$

$$\stackrel{(b)}{\lesssim} \mathrm{C}_1 \sqrt{mpt_{\mathrm{mix}}}\alpha_k,$$

where we set
$$\mathrm{C}_1 = (\kappa_Q^{3/2}/a)\,\mathrm{C}_{\mathbf{A}}\,\|\varepsilon\|_\infty\;.$$
In the line (a) above we used Lemma 11, and in (b) we used Lemma 28. Similarly, using the same lemmas, we bound the term $T_3$:
$$\mathbb{E}_\xi^{1/p}\big[\big|u^\top T_3\big|^p\big] \lesssim \mathrm{C}_1\,\sqrt{mp}t_{\mathrm{mix}}\alpha_k\;.$$
Note that the second term $T_2$ is non-zero only for $N \geq 2$ and it can be rewritten as $T_2 = T_{21} + T_{22}$, where

$$T_{21} := \sum_{j=0}^{N-2}\sum_{i=1}^m \alpha_{jm+i}S_{(j+1)m+i+1:k}G_{jm+i+1:(j+1)m+i}\varepsilon(Z_{jm+i}^*),$$

$$T_{22} := \sum_{j=0}^{N-2}\sum_{i=1}^m \alpha_{jm+i}S_{(j+1)m+i+1:k}G_{jm+i+1:(j+1)m+i}(\varepsilon(Z_{jm+i}) - \varepsilon(Z_{jm+i}^*))\;.$$

The set of random variables $Z_{jm+i}^*$ is constructed for each $i \in [1,m]$, with $\{Z_{jm+i}^*\}_{j=0}^{N-2}$ having the following properties:

1. $Z_{jm+i}^*$ is independent of $\mathfrak{F}_{(j+1)m+i}^k := \sigma\{Z_{(j+1)m+i},\ldots,Z_k\}$;
2. $\mathbb{P}_\xi(Z_{jm+i}^* \neq Z_{jm+i}) \leq 2\,(1/4)^{\lceil m/t_{\mathrm{mix}}\rceil}$;     (45)
3. $Z_{jm+i}^*$ and $Z_{jm+i}$ have the same distribution,

The existence of the random variables $Z_{jm+i}^*$ is guaranteed by Berbee's lemma, see e.g [60, Lemma 5.1], together with the fact that uniformly geometrically ergodic Markov chains are a special instance of $\beta$-mixing processes. We control $\beta$-mixing coefficient via total variation distance, see [23, Theorem F.3.3]. In order to analyze the term $T_{21}$ we use Minkowski's and Burkholder's inequality [52, Theorem 8.6], and obtain that:

$$\mathbb{E}_\xi^{1/p}\big[\big|u^\top T_{21}\big|^p\big] \leq \sum_{i=1}^m \mathbb{E}_\xi^{1/p}\bigg[\bigg|\sum_{j=0}^{N-2}\alpha_{jm+i}u^\top S_{(j+1)m+i+1:k}G_{jm+i+1:(j+1)m+i}\varepsilon(Z_{jm+i}^*)\bigg|^p\bigg]$$

$$\leq p\sum_{i=1}^m\bigg(\sum_{j=0}^{N-2}\alpha_{jm+i}^2\mathbb{E}^{2/p}\big[\big|u^\top S_{(j+1)m+i+1:k}G_{jm+i+1:(j+1)m+i}\varepsilon(Z_{jm+i}^*)\big|^p\big]\bigg)^{1/2}$$

$$\leq p\sqrt{m}\bigg(\sum_{j=1}^k\alpha_j^2\mathbb{E}^{2/p}\big[\big|u^\top S_{j+m+1:k}G_{j+1:j+m}\varepsilon(Z_j^*)\big|^p\big]\bigg)^{1/2}\;.$$

Applying now Lemma 11, we arrive at the bound

$$\mathbb{E}_\xi^{1/p}\big[\big|u^\top T_{21}\big|^p\big] \lesssim \kappa_Q^{3/2}\,\mathrm{C}_{\mathbf{A}}\,\|\varepsilon\|_\infty t_{\mathrm{mix}}^{1/2}p^{3/2}\sqrt{m}\bigg(\sum_{j=1}^k\alpha_j^2\big(\sum_{\ell=j+m+1}^k\alpha_\ell^2\big)\prod_{\ell=j+1}^k(1-\alpha_\ell a)\bigg)^{1/2}\;. \quad (46)$$

In order to bound the term $T_{22}$ we first note that

$$\mathbb{E}_\xi^{1/p}\big[\big|u^\top S_{(j+1)m+i+1:k}G_{jm+i+1:(j+1)m+i}(\varepsilon(Z_{jm+i}) - \varepsilon(Z_{jm+i}^*))\big|^p\big]$$
$$\leq \mathbb{E}_\xi^{1/p}\big[\|G_{jm+i+1:(j+1)m+i}\{\varepsilon(Z_{jm+i})-\varepsilon(Z_{jm+i}^*)\}\|^p\big]\sup_{u,v\in\mathbb{S}_{d-1},\xi'}\mathbb{E}_{\xi'}\big[\big|u^\top S_{(j+1)m+i+1:k}v\big|^p\big]\big]\;.$$

Hence, using Minkowski's inequality for $T_{22}$, we obtain that

$$\mathbb{E}_\xi^{1/p}\big[\big|u^\top T_{22}\big|^p\big] \leq \sqrt{\kappa_Q}\sum_{j=0}^{N-2}\sum_{i=1}^m\alpha_{jm+i}\sup_{u,v\in\mathbb{S}_{d-1},\xi'}\mathbb{E}_{\xi'}^{1/p}\big[\big|u^\top S_{(j+1)m+i+1:k}v\big|^p\big]\mathbb{E}_\xi^{1/p}\big[\|\varepsilon(Z_{jm+i}) - \varepsilon(Z_{jm+i}^*)\|^p\big]$$

$$\times\prod_{\ell=jm+i+1}^{(j+1)m+i}(\sqrt{1-\alpha_\ell a}).$$

Using the definition of $Z^*_{km+i}$ and the Cauchy-Schwartz inequality,

$$\mathbb{E}_\xi^{1/p}[\|\varepsilon(Z_{km+i}) - \varepsilon(Z^*_{km+i})\|^p] = \mathbb{E}_\xi^{1/p}[\|\varepsilon(Z_{km+i}) - \varepsilon(Z^*_{km+i}))\mathbb{1}\{Z^*_{km+i} \neq Z_{km+i}\}\|^p]$$
$$\stackrel{(a)}{\lesssim} \|\varepsilon\|_\infty (1/4)^{m/(2pt_{\mathrm{mix}})}.$$

where in (a) we used (45). The last two inequalities together with Lemma 11 imply

$$\mathbb{E}_\xi^{1/p}[|u^\top T_{22}|^p] \lesssim \kappa_Q^{3/2} \, \mathrm{C_A} \, \|\varepsilon\|_\infty t_{\mathrm{mix}}^{1/2} p^{1/2} (1/4)^{m/(2pt_{\mathrm{mix}})} \sum_{j=1}^k \alpha_j \big( \sum_{\ell=j+1}^k \alpha_\ell^2 \big)^{1/2} \prod_{\ell=j+1}^k \sqrt{1 - a\alpha_\ell} \, . \tag{47}$$

Combining now (46) and (47), we obtain

$$\mathbb{E}_\xi^{1/p}[|u^\top J_k^{(1)}|^p] \lesssim \mathrm{C_1} \sqrt{mp} t_{\mathrm{mix}}^{1/2} \alpha_k$$
$$+ \kappa_Q^{3/2} \, \mathrm{C_A} \, \|\varepsilon\|_\infty t_{\mathrm{mix}}^{1/2} p^{3/2} \sqrt{m} \bigg( \sum_{j=1}^k \alpha_j^2 \big( \sum_{\ell=j+1}^k \alpha_\ell^2 \big) \prod_{\ell=j+1}^k \sqrt{1 - \alpha_\ell a} \bigg)^{1/2}$$
$$+ \kappa_Q^{3/2} \, \mathrm{C_A} \, \|\varepsilon\|_\infty t_{\mathrm{mix}}^{1/2} p^{1/2} (1/4)^{m/(2pt_{\mathrm{mix}})} \sum_{j=1}^k \alpha_j \big( \sum_{\ell=j+1+m}^k \alpha_\ell^2 \big)^{1/2} \prod_{\ell=j+1}^k \sqrt{1 - a\alpha_\ell} \, .$$

Applying now Lemma 29, we arrive at the bound:

$$\mathbb{E}_\xi^{1/p}[|u^\top J_k^{(1)}|^p] \lesssim \mathrm{C_1} \sqrt{mp} t_{\mathrm{mix}}^{1/2} \alpha_k$$
$$+ (\kappa_Q^{3/2} \, \mathrm{C_A} \, \|\varepsilon\|_\infty /a) t_{\mathrm{mix}}^{1/2} p^{3/2} \sqrt{m} 3^\gamma \left( \frac{8\gamma}{ac_0} \right)^{\gamma/(1-\gamma)} \alpha_k$$
$$(\kappa_Q^{3/2} \, \mathrm{C_A} \, \|\varepsilon\|_\infty /a^{3/2}) t_{\mathrm{mix}}^{1/2} p^{1/2} (1/4)^{m/(4pt_{\mathrm{mix}})} 3^{\gamma/2} \sqrt{\alpha_k} \, .$$

Hence, it remains to set the block size $m$ as

$$(1/4)^{m/(4pt_{\mathrm{mix}})} \leq \sqrt{\alpha_k}, \text{ i.e. } m = \left\lceil \frac{2pt_{\mathrm{mix}} \log(1/\alpha_k)}{\log 4} \right\rceil .$$

With this choice of $m$ we obtain from the above inequality that

$$\mathbb{E}_\xi^{1/p}[|u^\top J_k^{(1)}|^p] \lesssim \mathrm{C_1} \, pt_{\mathrm{mix}} \sqrt{\log(1/\alpha_k)} \alpha_k$$
$$+ (\kappa_Q^{3/2} \, \mathrm{C_A} \, \|\varepsilon\|_\infty /a) t_{\mathrm{mix}} p^2 \sqrt{\log(1/\alpha_k)} 3^\gamma \left( \frac{8\gamma}{ac_0} \right)^{\gamma/(1-\gamma)} \alpha_k$$
$$+ (\kappa_Q^{3/2} \, \mathrm{C_A} \, \|\varepsilon\|_\infty /a^{3/2}) t_{\mathrm{mix}}^{1/2} p^{1/2} 3^{\gamma/2} \alpha_k$$
$$\lesssim \mathrm{C_3} \, p^2 t_{\mathrm{mix}} \sqrt{\log(1/\alpha_k)} \alpha_k$$

where we have defined

$$\mathrm{C_3} = \frac{\kappa_Q^{3/2} \, \mathrm{C_A} \, \|\varepsilon\|_\infty}{a} + \frac{\kappa_Q^{3/2} \, \mathrm{C_A} \, \|\varepsilon\|_\infty}{a^{3/2}} 3^{\gamma/2} + \frac{\kappa_Q^{3/2} \, \mathrm{C_A} \, \|\varepsilon\|_\infty}{a} 3^\gamma \left( \frac{8\gamma}{ac_0} \right)^{\gamma/(1-\gamma)} ,$$

and the bound (41) holds. To estimate $H_k^{(1)}$ we rewrite it as follows

$$H_k^{(1)} = -\sum_{j=1}^k \alpha_j \Gamma_{j+1:k}^{(\alpha)} \tilde{\mathbf{A}}(Z_j) J_{j-1}^{(1)} \, .$$

Using Minkowski's inequality together with Hölder's inequality, we get

$$\mathbb{E}_\xi^{1/p}[|u^\top H_k^{(1)}|^p] \leq \sum_{j=1}^k \mathrm{C_A} \, \alpha_j \big\{ \mathbb{E}_\xi \big[ \|\Gamma_{j+1:k}^{(\alpha)}\|^{2p} \big] \big\}^{1/2p} \big\{ \mathbb{E}_\xi \big[ \|J_{j-1}^{(1)}\|^{2p} \big] \big\}^{1/2p} \, .$$

Applying similar argument to (38) together with Proposition 12 and an elementary inequality $\mathrm{e}^{-x} \leq 1 - x/2$, valid for $x \in [0, 1]$, we obtain the bound (41), we get

$$\mathbb{E}_\xi^{1/p}[|u^\top H_k^{(1)}|^p] \lesssim d^{1/2+1/\log n} C_\Gamma \, \mathrm{C_A} \, \mathrm{D}_4^{(\mathrm{M})} t_{\mathrm{mix}} p^2 \sum_{j=1}^k \alpha_j^2 \sqrt{\log(1/\alpha_j)} \prod_{\ell=j+1}^k (1 - a\alpha_\ell/24)$$

$$\lesssim d^{1/2+1/\log n} C_\Gamma \, \mathrm{C_A} \, \mathrm{D}_4^{(\mathrm{M})} t_{\mathrm{mix}} p^2 \sqrt{\log(1/\alpha_k)} \sum_{j=1}^k \alpha_j^2 \prod_{\ell=j+1}^k (1 - a\alpha_\ell/24)$$

$$\overset{(a)}{\lesssim} \mathrm{D}_5^{(\mathrm{M})} t_{\mathrm{mix}} p^2 \alpha_k \sqrt{\log(1/\alpha_k)} \,,$$

where in (a) we have additionally used Lemma 28 and used the definition of $\mathrm{D}_5^{(\mathrm{M})}$ from (44). $\qquad\square$

## C.3 Technical bounds related to $J_k^{(1)}$.

Recall that $S_{\ell+1:\ell+m}$ is defined, for $\ell, m \in \mathbb{N}$, as

$$S_{\ell+1:\ell+m} = \sum_{k=\ell+1}^{\ell+m} \alpha_k \mathbf{B}_k(Z_k) \,, \quad \text{with } \mathbf{B}_k(z) = G_{k+1:\ell+m} \tilde{\mathbf{A}}(z) G_{\ell+1:k-1} \,. \tag{48}$$

**Lemma 11.** *Under the assumptions of Proposition 11, it holds for any vector $u \in \mathbb{S}_{d-1}$, $\ell, m \in \mathbb{N}$, and any initial distribution $\xi$ on $(\mathsf{Z}, \mathcal{Z})$, that*

$$\mathbb{E}_\xi^{1/p}[|u^\top S_{\ell+1:\ell+m}\varepsilon(Z_\ell)|^p] \leq C_S \Big(\sum_{r=\ell+1}^{\ell+m} \alpha_r^2\Big)^{1/2} \prod_{k=\ell+1}^{\ell+m} \sqrt{1 - a\alpha_k} \,,$$

*where*

$$C_S = 7\kappa_Q \, \mathrm{C_A} \, p^{1/2} t_{\mathrm{mix}}^{1/2} \|\varepsilon\|_\infty \,.$$

*Proof.* We first prove the auxiliary inequality for deterministic vectors $u, v \in \mathcal{S}_{d-1}$. Indeed, using (48), we obtain that

$$u^\top S_{\ell+1:\ell+m} v = \sum_{k=\ell+1}^{\ell+m} h_k(Z_k) \,, \quad \text{where } h_k(z) := \alpha_k u^\top \mathbf{B}_k(z) v \,.$$

It is easy to check that $u^\top S_{\ell+1:\ell+m} v$ satisfies the bounded differences property, since for any $z, z' \in \mathsf{Z}$, and $r \in \{\ell+1, \ldots, \ell+m\}$, it holds that

$$|h_r(z) - h_r(z')| \leq 2\kappa_Q \, \mathrm{C_A} \, \alpha_r \prod_{k=\ell+1, k\neq r}^{\ell+m} \sqrt{1 - a\alpha_k} \leq 2^{3/2} \kappa_Q \, \mathrm{C_A} \, \alpha_r \prod_{k=\ell+1}^{\ell+m} \sqrt{1 - a\alpha_k} \,.$$

In the last inequality we have additionally used the fact that $\alpha_k a \leq 1/2$ for any $k \in \mathbb{N}$. Applying the bounded differences inequality from [54][Corollary 2.11], we get that for any $t \geq 0$,

$$\mathbb{P}_\xi(|u^\top S_{j+1:j+m} v| \geq t) \leq 2 \exp\left\{-\frac{2(t - |\mathbb{E}_\xi[u^\top S_{j+1:j+m} v]|)^2}{72 t_{\mathrm{mix}} \kappa_Q^2 \, \mathrm{C_A^2} (\sum_{r=\ell+1}^{\ell+m} \alpha_r^2) \prod_{k=\ell+1}^{\ell+m} (1 - a\alpha_k)}\right\} \,.$$

It remains to upper bound $\mathbb{E}_\xi[u^\top S_{j+1:j+m} v]$. Note that

$$(\mathbb{E}_\xi[u^\top S_{j+1:j+m} v])^2 \leq \mathbb{E}_\xi\Big[\Big(\sum_{k=l+1}^{l+m} h_k(Z_k)\Big)^2\Big] = \sum_{k=l+1}^{l+m} \mathbb{E}_\xi[h_k(Z_k)^2] + 2\sum_{k=l+1}^{l+m} \sum_{j=1}^{l+m-k} \mathbb{E}_\xi[h_k(Z_k) h_{k+j}(Z_{k+j})] \,.$$

Using that $\pi(h_k) = 0$ and A1, we obtain

$$|\mathbb{E}_\xi[h_k(Z_k) h_{k+j}(Z_{k+j})]| = |\int_\mathsf{Z} h_k(z)(\mathrm{P}^j h_{k+j}(z) - \pi(h_{k+j}))\xi \mathrm{P}^k(\mathrm{d}z)| \leq \|h_k\|_\infty \|h_{k+j}\|_\infty \Delta(\mathrm{P}^j)$$

$$\leq 2\alpha_k \alpha_{k+j} \kappa_Q^2 (2\, \mathrm{C_A})^2 \prod_{t=l+1}^{l+m} (1 - a\alpha_t)(1/4)^{\lceil j/t_{\mathrm{mix}}\rceil} \,,$$

and

$$\mathbb{E}_\xi[h_k(Z_k)^2] \le 2\alpha_k^2 \kappa_Q^2 (2\,\mathrm{C_A})^2 \prod_{t=l+1}^{l+m}(1-a\alpha_t)\,.$$

Combining inequalities above, we obtain

$$(\mathbb{E}_\xi[u^\top S_{j+1:j+m}v])^2 \le \frac{32}{3} t_{\mathrm{mix}} \kappa_Q^2\,\mathrm{C_A^2}\Big(\sum_{k=l+1}^{l+m}\alpha_k^2\Big)\prod_{t=l+1}^{l+m}(1-a\alpha_t)\,.$$

Note that for $t \ge |\mathbb{E}_\xi[u^\top S_{j+1:j+m}v]|$, we have

$$\mathbb{P}_\xi\big(\big|u^\top S_{j+1:j+m}v\big| \ge t\big) \le 2\exp\bigg\{-\frac{t^2}{49 t_{\mathrm{mix}}\kappa_Q^2\,\mathrm{C_A^2}\big(\sum_{r=\ell+1}^{\ell+m}\alpha_r^2\big)\prod_{k=\ell+1}^{\ell+m}(1-a\alpha_k)}\bigg\}\,.$$

And for $t \le |\mathbb{E}_\xi[u^\top S_{j+1:j+m+m}v]|$, the right side of the inequality is greater than 1, so this inequality is also true for $t \le |\mathbb{E}_\xi[u^\top S_{j+1:j+m+m}v]|$. Hence, using [24, Lemma 7], we obtain that

$$\mathbb{E}_\xi^{1/p}\big[\big|u^\top S_{j+1:j+m}v\big|^p\big] \le 7p^{1/2}\kappa_Q\,\mathrm{C_A}\,t_{\mathrm{mix}}^{1/2}\Big(\sum_{r=\ell+1}^{\ell+m}\alpha_r^2\Big)^{1/2}\prod_{k=\ell+1}^{\ell+m}\sqrt{1-a\alpha_k}\,. \qquad (49)$$

Now, with $\mathcal{F}_\ell = \sigma\{Z_j, j \le \ell\}$, it holds that

$$\mathbb{E}_\xi^{1/p}[|u^\top S_{\ell+1:\ell+m}\varepsilon(Z_\ell)|^p] = \mathbb{E}_\xi^{1/p}\big[\|\varepsilon(Z_\ell)\|^p\mathbb{E}^{\mathcal{F}_\ell}\big[\big|u^\top S_{\ell+1:\ell+m}\varepsilon(Z_\ell)\big|^p/\|\varepsilon(Z_\ell)\|^p\big]\big]$$

$$\le \|\varepsilon\|_\infty \sup_{u,v\in\mathbb{S}_{d-1},\,\xi'\in\mathcal{P}(\mathsf{Z})}\mathbb{E}_{\xi'}^{1/p}\big[\big|u^\top S_{\ell+1:\ell+m}v\big|^p\big]\,.$$

Combining the above bounds with (49) and A2 yields the statement. $\qquad\square$

## D  Proofs for stability of matrix products

### D.1  Stability

We first provide a result on the product of dependent random matrices. Our proof technique is based on the approach of [33] and the results, previously obtained in [24]. Let $(\Omega, \mathfrak{F}, \{\mathfrak{F}_\ell\}_{\ell\in\mathbb{N}}, \mathbb{P})$ be a filtered probability space. For the matrix $B \in \mathbb{R}^{d\times d}$ we denote by $(\sigma_\ell(B))_{\ell=1}^d$ its singular values. For $q \ge 1$, the Shatten $q$-norm is denoted by $\|B\|_q = \{\sum_{\ell=1}^d \sigma_\ell^q(B)\}^{1/q}$. For $q, p \ge 1$ and a random matrix $\mathbf{X}$ we write $\|\mathbf{X}\|_{q,p} = \{\mathbb{E}[\|\mathbf{X}\|_q^p]\}^{1/p}$. The main result of this section is stated below:

**Proposition 12.** *Assume A1, A2, and A3. Then, for any $2 \le p \le \log n$, $n \in \mathbb{N}$, and probability distribution $\xi$ on $(\mathsf{Z}, \mathcal{Z})$, it holds that*

$$\mathbb{E}_\xi^{1/p}\Big[\|\Gamma_{j:n}^{(\alpha)}\|^p\Big] \le C_\Gamma d^{1/\log n}\exp\bigg\{-(a/12)\sum_{k=j}^n\alpha_k\bigg\}\,,$$

*where*

$$C_\Gamma = \sqrt{\kappa_Q}\mathrm{e}^2\,.$$

The proof is given in Appendix D.2. It is based on a simplification of the arguments in [26] together with a new result about the matrix concentration for the product of random matrices, using a proof method introduced in [33]. We first state a result from [24]:

**Proposition 13** (Proposition 15 in [24])**.** *Let $\{\mathbf{Y}_\ell\}_{\ell\in\mathbb{N}}$ be a sequence of random matrices adapted to the filtration $\{\mathfrak{F}_\ell\}_{\ell\in\mathbb{N}}$ and $P$ be a positive definite matrix. Assume that for each $\ell \in \mathbb{N}^*$ there exist $\mathrm{m}_\ell \in (0, 1]$ and $\sigma_\ell > 0$ such that*

$$\|\mathbb{E}^{\mathfrak{F}_{\ell-1}}[\mathbf{Y}_\ell]\|_P^2 \le 1 - \mathrm{m}_\ell \text{ and } \|\mathbf{Y}_\ell - \mathbb{E}^{\mathfrak{F}_{\ell-1}}[\mathbf{Y}_\ell]\|_P \le \sigma_\ell \quad \mathbb{P}\text{-a.s.}\,.$$

*Define $\mathbf{Z}_n = \prod_{\ell=0}^n \mathbf{Y}_\ell = \mathbf{Y}_n \mathbf{Z}_{n-1}$, for $n \ge 1$. Then, for any $2 \le p \le q$ and $n \ge 1$,*

$$\|\mathbf{Z}_n\|_{q,p}^2 \le \kappa_P \prod_{\ell=1}^n (1 - \mathrm{m}_\ell + (q-1)\sigma_\ell^2)\|P^{1/2}\mathbf{Z}_0 P^{-1/2}\|_{q,p}^2\,,$$

*where $\kappa_P = \lambda_{\max}(P)/\lambda_{\min}(P)$ and $\lambda_{\max}(P), \lambda_{\min}(P)$ correspond to the largest and smallest eigenvalues of $P$.*

Now we fix some $N \in \mathbb{N}$, block size $h \in \mathbb{N}$ defined in (23), and a sequence $0 = j_0 < j_1 < \ldots < j_N = 2n$, where $j_\ell = h\ell$, $\ell \leq N - 1$. Note that it is possible that $j_N - j_{n-1} < h$. Now we set

$$\mathbf{Y}_\ell = \prod_{i=j_{\ell-1}}^{j_\ell - 1} \left( \mathrm{I} - \alpha_i \mathbf{A}(Z_i) \right) .$$

Then the following lemma holds:

**Lemma 12.** *Assume A1, A2, and A3. Then for any $\ell \in \{1, \ldots, N-1\}$, and any probability measure $\xi$ on $(\mathsf{Z}, \mathcal{Z})$, it holds that*

$$\| \mathbb{E}_\xi[\mathbf{Y}_\ell] \|_Q \leq 1 - \Big( \sum_{k=j_{\ell-1}}^{j_\ell - 1} \alpha_k \Big) a/6 .$$

*Proof.* We decompose the matrix product $\mathbf{Y}_\ell$ as follows:

$$\mathbf{Y}_\ell = \mathrm{I} - \Big( \sum_{k=j_{\ell-1}}^{j_\ell - 1} \alpha_k \Big) \bar{\mathbf{A}} - \mathbf{S}_\ell + \mathbf{R}_\ell . \tag{50}$$

Here $\mathbf{S}_\ell = \sum_{k=j_{\ell-1}}^{j_\ell - 1} \alpha_k \big\{ \mathbf{A}(Z_k) - \bar{\mathbf{A}} \big\}$ is a linear statistics in $\{\mathbf{A}(Z_k)\}_{k=j_{\ell-1}}^{j_\ell - 1}$, and the remainder term $\mathbf{R}_\ell$ is defined as

$$\mathbf{R}_\ell = \sum_{r=2}^{h} (-1)^r \sum_{(i_1, \ldots, i_r) \in \mathsf{I}_r^\ell} \prod_{u=1}^{r} \{ \alpha_{i_u} \mathbf{A}(Z_{i_u}) \} ,$$

where $\mathsf{I}_r^\ell = \{ (i_1, \ldots, i_r) \in \{1, \ldots, h\}^r : i_1 < \cdots < i_r \}$. Since $\|M\|_Q = \|Q^{1/2} M Q^{-1/2}\|$, it is straightforward to check that $\mathbb{P}$-a.s. it holds

$$\| \mathbf{R}_\ell \|_Q \leq \frac{(\sum_{k=j_{\ell-1}}^{j_\ell - 1} \alpha_k)^2 \kappa_Q \, \mathrm{C}_{\mathbf{A}}^2}{2} \exp\big\{ \kappa_Q^{1/2} \, \mathrm{C}_{\mathbf{A}} \sum_{k=j_{\ell-1}}^{j_\ell - 1} \alpha_k \big\} =: T_2 . \tag{51}$$

On the other hand, using A1 and A2, we have for any $k \in \mathbb{N}$, that

$$\| \mathbb{E}_\xi[\mathbf{A}(Z_k) - \bar{\mathbf{A}}] \| = \sup_{u,v \in \mathbb{S}^{d-1}} [\mathbb{E}_\xi[u^\top \mathbf{A}(Z_k) v] - u^\top \bar{\mathbf{A}} v] \leq \mathrm{C}_{\mathbf{A}} \, \Delta(\mathrm{P}^k) .$$

Hence, with the triangle inequality we obtain that

$$\| \mathbb{E}_\xi[\mathbf{S}_\ell] \|_Q \leq \kappa_Q^{1/2} \sum_{k=j_{\ell-1}}^{j_\ell - 1} \alpha_k \| \mathbb{E}_\xi[\mathbf{A}(Z_k) - \bar{\mathbf{A}}] \| \leq \kappa_Q^{1/2} \, \mathrm{C}_{\mathbf{A}} \sum_{k=j_{\ell-1}}^{j_\ell - 1} \alpha_k \Delta(\mathrm{P}^k)$$

$$\leq (4/3) \alpha_{j_{\ell-1}} t_{\mathrm{mix}} \kappa_Q^{1/2} \, \mathrm{C}_{\mathbf{A}} =: T_1 .$$

This result combined with (51) in (50) implies that

$$\| \mathbb{E}_\xi[\mathbf{Y}_\ell] \|_Q \leq \| \mathrm{I} - \sum_{k=j_{\ell-1}}^{j_\ell - 1} \alpha_k \bar{\mathbf{A}} \|_Q + T_1 + T_2 .$$

First, by definition (23) of $h$ (see details in Lemma 13), we have

$$T_1 \leq \Big( \sum_{k=j_{\ell-1}}^{j_\ell - 1} \alpha_k \Big) a/6 . \tag{52}$$

Second, with the definition of $c_0$ in (24), we obtain that

$$T_2 \leq (\kappa_Q^{1/2} \, \mathrm{C}_{\mathbf{A}} \sum_{k=j_{\ell-1}}^{j_\ell - 1} \alpha_k)^2 \mathrm{e} \leq \Big( \sum_{k=j_{\ell-1}}^{j_\ell - 1} \alpha_k \Big) a/6 . \tag{53}$$

Finally, Proposition 2 implies that, for $\sum_{k=j_{\ell-1}}^{j_\ell - 1} \alpha_k \leq \alpha_\infty$, it holds that

$$\| \mathrm{I} - \Big( \sum_{k=j_{\ell-1}}^{j_\ell - 1} \alpha_k \Big) \bar{\mathbf{A}} \|_Q \leq 1 - (a/2) \sum_{k=j_{\ell-1}}^{j_\ell - 1} \alpha_k . \tag{54}$$

Combining (52), (53), and (54) yield that

$$\| \mathbb{E}_\xi[\mathbf{Y}_1] \|_Q \leq 1 - \Big( \sum_{k=j_{\ell-1}}^{j_\ell - 1} \alpha_k \Big) a/6 ,$$

and the statement follows. $\qquad\square$

**Lemma 13.** *Assume A3. Then, for the block size $h$ defined in (23), and any $\ell \in \{1, \ldots, 2n - h\}$, it holds that*

$$\sum_{k=\ell}^{\ell+h} \alpha_k \leq \alpha_\infty , \quad \kappa_Q^{1/2} \, C_{\mathbf{A}} \sum_{k=\ell}^{\ell+h} \alpha_k \leq 1 , \quad \kappa_Q \, C_{\mathbf{A}}^2 \, e \sum_{k=\ell}^{\ell+h} \alpha_k \leq a/6 ,$$

$$(4/3)\alpha_\ell t_{\mathrm{mix}}\kappa_Q^{1/2} \, C_{\mathbf{A}} \leq \big(\sum_{k=\ell}^{\ell+h} \alpha_k\big)a/6 , \tag{55}$$

$$(\log n) \, C_\sigma^2 \sum_{k=\ell}^{\ell+h} \alpha_k \leq \frac{a}{12} . \tag{56}$$

*Proof.* First three inequalities above are easy to check, since

$$\sum_{k=\ell}^{\ell+h} \alpha_k \leq \sum_{k=1}^{1+h} \frac{c_0}{(k+k_0)^\gamma} \leq c_0 \int_{k_0}^{1+k_0+h} \frac{dx}{x^\gamma} = \frac{c_0\{(1+h+k_0)^{1-\gamma} - k_0^{1-\gamma}\}}{1-\gamma}$$

$$= \frac{c_0 k_0^{1-\gamma}((1+\frac{h+1}{k_0})^{1-\gamma} - 1)}{1-\gamma} \leq c_0 k_0^{-\gamma}(h+1) ,$$

hence, in order to satisfy these inequalities, it is enough to choose $h$ in such a manner that

$$k_0^\gamma \geq c_0(h+1) \max(\alpha_\infty, \kappa_Q^{1/2} \, C_{\mathbf{A}}, 6e\kappa_Q \, C_{\mathbf{A}}^2 /a) ,$$

which is guaranteed by our choice of $c_0$ in A3 and (23). Now note that

$$\sum_{k=\ell}^{\ell+h} \alpha_k \geq c_0 \int_{\ell+k_0}^{\ell+k_0+h+1} \frac{dx}{x^\gamma} = \frac{c_0\{(\ell+k_0+h+1)^{1-\gamma} - (\ell+k_0)^{1-\gamma}\}}{1-\gamma} ,$$

hence, in order to check (55), we need to ensure that

$$\frac{4c_0 t_{\mathrm{mix}}\kappa_Q^{1/2} \, C_{\mathbf{A}}}{3(\ell+k_0)^\gamma} \leq \frac{c_0 a\{(\ell+k_0+h+1)^{1-\gamma} - (\ell+k_0)^{1-\gamma}\}}{6(1-\gamma)} .$$

Equivalently, it is enough to set $h$ in such a way, that

$$(\ell+k_0+h+1)^{1-\gamma} - (\ell+k_0)^{1-\gamma} \geq \frac{C_1}{(\ell+k_0)^\gamma} ,$$

where we set $C_1 = 8(1-\gamma)t_{\mathrm{mix}}\kappa_Q^{1/2} \, C_{\mathbf{A}} /a$. Hence, (55) will be satisfied if

$$h \geq (\ell+k_0)\Big(1 + \frac{C_1}{\ell+k_0}\Big)^{1/(1-\gamma)} - (\ell+k_0+1) .$$

Note that, with $\alpha > 1$, it holds that $(1+x)^\alpha \leq 1 + 2\alpha x$ for $0 < x \leq 1/\alpha$. Hence, provided that $k_0 > C_1/(1-\gamma)$, or, equivalently,

$$k_0 \geq \frac{8t_{\mathrm{mix}}\kappa_Q^{1/2} \, C_{\mathbf{A}}}{a} ,$$

it is enough to set

$$h \geq (\ell+k_0)\big(1 + \frac{2C_1}{(\ell+k_0)(1-\gamma)}\big) - (\ell+k_0+1) = \frac{16t_{\mathrm{mix}}\kappa_Q^{1/2} \, C_{\mathbf{A}}}{a} - 1 .$$

Now it remains to check (56), that is,

$$(\log n) \, C_\sigma^2 \sum_{k=\ell}^{\ell+h} \frac{c_0}{k^\gamma} \leq \frac{a}{12} .$$

Since

$$\sum_{k=\ell}^{\ell+h} \frac{1}{(k+k_0)^\gamma} \leq \sum_{k=1}^{h+1} \frac{1}{(k+k_0)^\gamma} \leq \int_{k_0}^{h+1+k_0} \frac{\mathrm{d}x}{x^\gamma} = \frac{(h+k_0+1)^{1-\gamma} - (k_0)^{1-\gamma}}{1-\gamma} ,$$

it is enough to choose $k_0$ in such a way that

$$(\log n)\, \mathrm{C}_\sigma^2\, c_0 \frac{(h+k_0+1)^{1-\gamma} - (k_0)^{1-\gamma}}{1-\gamma} \leq \frac{a}{12} ,$$

or, equivalently,

$$(k_0)^{1-\gamma}((1 + \frac{h+1}{k_0})^{1-\gamma} - 1) \leq \frac{a(1-\gamma)}{12(\log n)\, \mathrm{C}_\sigma^2\, c_0} . \tag{57}$$

Since $(1+x)^{1-\gamma} \leq 1 + (1-\gamma)x$ for $x \geq -1$, (57) holds if

$$k_0^{-\gamma}(1-\gamma)(h+1) \leq \frac{a(1-\gamma)}{12(\log n)\, \mathrm{C}_\sigma^2\, c_0} ,$$

or, equivalently,

$$k_0 \geq \left( \frac{12(h+1)(\log n)c_0\, \mathrm{C}_\sigma^2}{a} \right)^{1/\gamma}$$

which is guaranteed by the condition (25). $\qquad\square$

**Lemma 14.** *Assume A1, A2, and A3. Then, for any probability $\xi$ on $(\mathsf{Z}, \mathcal{Z})$, and any $\ell \in \{1, \ldots, N-1\}$, we have*

$$\|\mathbf{Y}_\ell - \mathbb{E}_\xi[\mathbf{Y}_\ell]\|_Q \leq \mathrm{C}_\sigma \Big( \sum_{k=j_{\ell-1}}^{j_\ell - 1} \alpha_k \Big) , \text{ where } \mathrm{C}_\sigma = 2(\kappa_Q^{1/2}\, \mathrm{C}_\mathbf{A} + a/6) , \tag{58}$$

*and $h$ is given in* (23).

*Proof.* Using the decomposition (50), we obtain

$$\|\mathbf{Y}_\ell - \mathbb{E}_\xi[\mathbf{Y}_\ell]\|_Q \leq \sum_{k=j_{\ell-1}}^{j_\ell - 1} \alpha_k \|\mathbf{A}(Z_k) - \mathbb{E}_\xi[\mathbf{A}(Z_k)]\|_Q + \|\mathbf{R}_\ell - \mathbb{E}_\xi[\mathbf{R}_\ell]\|_Q .$$

Applying the definition of $\mathbf{R}_\ell$ in (51), the definition of $h, \alpha_\infty^{(\mathrm{M})}$, and $T_2$ in (53), we get from the above inequalities that

$$\|\mathbf{Y}_\ell - \mathbb{E}_\xi[\mathbf{Y}_\ell]\|_Q \leq 2\kappa_Q^{1/2}\, \mathrm{C}_\mathbf{A} \Big( \sum_{k=j_{\ell-1}}^{j_\ell - 1} \alpha_k \Big) + (a/3) \Big( \sum_{k=j_{\ell-1}}^{j_\ell - 1} \alpha_k \Big) ,$$

and the statement follows. $\qquad\square$

We have now all ingredients required to prove Proposition 12.

### D.2 Proof of Proposition 12

Denote by $h \in \mathbb{N}$ a block length, the value of which is determined later. Define the sequence $j_0 = j$, $j_{\ell+1} = \min(j_\ell + h, n)$. By construction $j_{\ell+1} - j_\ell \leq h$. Let $N = \lceil (n-j)/h \rceil$. Now we introduce the decomposition

$$\Gamma_{j:n}^{(\alpha)} = \prod_{\ell=1}^{N} \mathbf{Y}_\ell , \quad \text{where} \quad \mathbf{Y}_\ell = \begin{cases} \prod_{i=j_{\ell-1}}^{j_\ell - 1} (\mathrm{I} - \alpha \mathbf{A}(Z_i)) , & \ell \in \{1, \ldots, N-1\} , \\ \prod_{i=j_{N-1}}^{n} (\mathrm{I} - \alpha \mathbf{A}(Z_i)) , & \ell = N . \end{cases} \tag{59}$$

The last block, $Y_N$, can be of smaller size than $h$. Now we apply the bound

$$\|\mathbf{Y}_N\| \leq \prod_{k=j_{N-1}}^{n} (1 + \alpha_k\, \mathrm{C}_\mathbf{A}) \leq \exp\Big\{ \mathrm{C}_\mathbf{A} \sum_{k=j_{N-1}}^{N} \alpha_k \Big\} \leq \mathrm{e} ,$$

where the last bound follows from the relation (24). Hence, substituting into (59), we get the following bound:

$$\mathbb{E}_\xi^{1/p}[\|\Gamma_{j:n}^{(\alpha)}\|^p] \le e\mathbb{E}_\xi^{1/p}[\|\textstyle\prod_{\ell=1}^{N-1} \mathbf{Y}_\ell\|^p] \ .$$

Now we bound $\mathbb{E}_\xi^{1/p}[\|\prod_{\ell=1}^{N-1} \mathbf{Y}_\ell\|^p]$ using the results from Proposition 13. To do so, we define, for $\ell \in \{1, \dots, N-1\}$, the filtration $\mathcal{H}_\ell = \sigma(Z_k \, : \, k \le j_\ell)$ and establish almost sure bounds on $\|\mathbb{E}_\xi^{\mathcal{H}_{\ell-1}}[\mathbf{Y}_\ell]\|_Q$ and $\|\mathbf{Y}_\ell - \mathbb{E}_\xi^{\mathcal{H}_{\ell-1}}[\mathbf{Y}_\ell]\|_Q$ for $\ell \in \{1, \dots, N-1\}$. More precisely, by the Markov property, it is sufficient to show that there exist $\mathtt{m} \in (0, 1]$ and $\sigma > 0$ such that for any probabilities $\xi, \xi'$ on $(\mathsf{Z}, \mathcal{Z})$,

$$\|\mathbb{E}_{\xi'}[\mathbf{Y}_\ell]\|_Q^2 \le 1 - \mathtt{m}_\ell \text{ and } \|\mathbf{Y}_\ell - \mathbb{E}_{\xi'}[\mathbf{Y}_\ell]\|_Q \le \sigma_\ell \ , \quad \mathbb{P}_\xi\text{-a.s. .} \tag{60}$$

Such bounds require the blocking procedure, since (60) not necessarily holds with $h = 1$. Setting $h$ as in equation (23), and applying Lemma 12 and Lemma 14, we show that (60) hold with

$$\mathtt{m}_\ell = \Big( \sum_{k=j_{\ell-1}}^{j_\ell - 1} \alpha_k \Big) a/6 \ , \quad \sigma_\ell = \mathrm{C}_\sigma \Big( \sum_{k=j_{\ell-1}}^{j_\ell - 1} \alpha_k \Big) \ ,$$

where $\mathrm{C}_\sigma$ is given in (58). Then, applying Proposition 13 with $q = \log n$, we get

$$\mathbb{E}_\xi^{1/p} \left[ \|\Gamma_{1:n}^{(\alpha)}\|^p \right] \le \mathbb{E}_\xi^{1/q} \left[ \|\Gamma_{1:n}^{(\alpha)}\|^q \right]$$

$$\le \sqrt{\kappa_Q} d^{1/q} e \prod_{\ell=1}^{N-1} \left( 1 - \Big( \sum_{k=j_{\ell-1}}^{j_\ell - 1} \alpha_k \Big) a/6 + (\log n) \, \mathrm{C}_\sigma^2 \Big( \sum_{k=j_{\ell-1}}^{j_\ell - 1} \alpha_k \Big)^2 \right)$$

$$\le \sqrt{\kappa_Q} d^{1/q} e \prod_{\ell=1}^{N-1} \exp\left\{ -\Big( \sum_{k=j_{\ell-1}}^{j_\ell - 1} \alpha_k \Big) a/6 + (\log n) \, \mathrm{C}_\sigma^2 \Big( \sum_{k=j_{\ell-1}}^{j_\ell - 1} \alpha_k \Big)^2 \right\}$$

$$\overset{(a)}{\le} \sqrt{\kappa_Q} e^2 d^{1/q} \exp\left\{ -\Big( \sum_{k=j}^{n} \alpha_k \Big) a/12 \right\} \ .$$

Here in (a) we used the fact that

$$(\log n) \, \mathrm{C}_\sigma^2 \Big( \sum_{k=j_{\ell-1}}^{j_\ell - 1} \alpha_k \Big) \le \frac{a}{12} \ ,$$

which holds due to A3.

### D.3 Proof of Proposition 2

First part of the statement (existence of $Q$) follows from [67, Proposition 1]. For the second part, we note that for any non-zero vector $x \in \mathbb{R}^d$, we have

$$\frac{x^\top (\mathrm{I} - \alpha\bar{\mathbf{A}})^\top Q (\mathrm{I} - \alpha\bar{\mathbf{A}}) x}{x^\top Q x} = 1 - \alpha \frac{x^\top (\bar{\mathbf{A}}^\top Q + Q\bar{\mathbf{A}}) x}{x^\top Q x} + \alpha^2 \frac{x^\top \bar{\mathbf{A}}^\top Q \bar{\mathbf{A}} x}{x^\top Q x}$$

$$= 1 - \alpha \frac{x^\top P x}{x^\top Q x} + \alpha^2 \frac{x^\top \bar{\mathbf{A}}^\top Q \bar{\mathbf{A}} x}{x^\top Q x}$$

$$\le 1 - \alpha \frac{\lambda_{\min}(P)}{\|Q\|} + \alpha^2 \|\bar{\mathbf{A}}\|_Q^2$$

$$\le 1 - \alpha a \ ,$$

where we set

$$a = \frac{1}{2} \frac{\lambda_{\min}(P)}{\|Q\|} \ ,$$

and used the fact that $\alpha \le \alpha_\infty$, where $\alpha_\infty$ is defined in (7).

# E   Proofs of Section 4

## E.1   Proof of Proposition 3

For completeness, we recall the setting considered in the main text. Even though the sequence $\{Z_k\}_{k\in\mathbb{N}}$ forms a Markov chain, the iterates $\{\theta_k\}_{k\in\mathbb{N}}$ do not form the Markov chain. As a consequence, the standard consistency results for overlapping batch means variance estimators cannot be applied directly to (18). In this appendix, we provide the detailed derivations underlying the statement that applying the block bootstrap to the sequence $\{\theta_\ell\}$ is asymptotically equivalent, up to a suitable correction, to applying the same procedure to the latent variables $\{\varepsilon(Z_\ell)\}$.

Note that for any $0 \le t \le n - b_n$, we unroll the LSA recursion only up to $\theta_t - \theta_\star$, rather than all the way to the initial iterate $\theta_0 - \theta_\star$:

$$\theta_{k+t} - \theta^\star = \Gamma_{t+1:t+k}(\theta_t - \theta^\star) - \sum_{\ell=t+1}^{k+t} \alpha_\ell \Gamma_{\ell+1:t+k}\varepsilon(Z_\ell) \ . \tag{61}$$

While one could also unroll the recursion fully to $\theta_0 - \theta_\star$, leading to an alternative representation suitable for the multiplier subsample bootstrap and yielding the same convergence rate, we adopt the form in (61) as it simplifies subsequent analysis and notation.

Using (61), similarly to (13), we can extract from the MSB estimate a linear statistic that depends on $\{\varepsilon(Z_\ell)\}$ and get

$$\bar{\theta}_{n,b_n}(u) = \frac{\sqrt{b_n}}{\sqrt{n - b_n + 1}} \sum_{t=0}^{n-b_n} w_t (\bar{\theta}_{b_n,t} - \bar{\theta}_n)^\top u \tag{62}$$

$$= \frac{\sqrt{b_n}}{\sqrt{n - b_n + 1}} \sum_{t=0}^{n-b_n} w_t (\tilde{W}_{b_n,t} - \tilde{W}_n)^\top u + \frac{\sqrt{b_n}}{\sqrt{n - b_n + 1}} \sum_{t=0}^{n-b_n} w_t (\tilde{D}_{b_n,t} - \tilde{D}_n)^\top u \ ,$$

where we have set

$$\tilde{W}_{b_n,t} = -\frac{1}{b_n} \sum_{k=1}^{b_n-1} Q_{k,t,b_n}\varepsilon(Z_{k+t}), \qquad \tilde{W}_n = -\frac{1}{n} \sum_{k=1}^{n-1} Q_k \varepsilon(Z_k),$$

$$\tilde{D}_{b_n,t} = \frac{1}{b_n} \sum_{k=1}^{b_n-1} \Gamma_{t+1:t+k}(\theta_t - \theta^\star) + \frac{1}{b_n} \sum_{k=1}^{b_n-1} H^{(0)}_{k,t,b_n}\varepsilon(Z_{k+t})$$

$$\tilde{D}_n = \frac{1}{n} \sum_{k=0}^{n-1} \Gamma_{1:k}(\theta_0 - \theta^\star) + \frac{1}{n} \sum_{k=1}^{n-1} H^{(0)}_k \varepsilon(Z_\ell) \ ,$$

and $Q_{k,t,b_n}, H^{(0)}_{k,t,b_n}, J^{(0)}_{k,t,b_n}$ are analogues of $Q_k, H^{(0)}_k, J^{(0)}_k$ for the bootstrap procedure and are defined as follows

$$Q_{k,t,b_n} = \alpha_{k+t} \sum_{\ell=k}^{b_n-1} G_{k+t+1:\ell+t} \ ,$$

$$H^{(0)}_{k,t,b_n} = - \sum_{\ell=t+1}^{k+t} \alpha_\ell \Gamma_{\ell+1:t+k}\tilde{\mathbf{A}}(Z_\ell) J^{(0)}_{\ell-1,t,b_n}$$

$$J^{(0)}_{k,t,b_n} = - \sum_{\ell=t+1}^{k+t} \alpha_\ell G_{\ell+1:k+t}\varepsilon(Z_\ell) \ .$$

Since existing results on the concentration of overlapping batch means variance estimators for Markov chains do not consider weighted sums, we replace the leading terms $\tilde{W}_{b_n,t}$ and $\tilde{W}_n$ with expressions

involving the constant matrix $\bar{\mathbf{A}}^{-1}$ and obtain

$$\bar{\theta}_{n,b_n}(u) = \frac{\sqrt{b_n}}{\sqrt{n-b_n+1}} \sum_{t=0}^{n-b_n} w_t(\bar{\theta}_{b_n,t} - \bar{\theta}_n)^\top u$$

$$= \frac{\sqrt{b_n}}{\sqrt{n-b_n+1}} \sum_{t=0}^{n-b_n} w_t(\bar{W}_{b_n,t} - \bar{W}_n)^\top u + \frac{\sqrt{b_n}}{\sqrt{n-b_n+1}} \sum_{t=0}^{n-b_n} w_t(\bar{D}_{b_n,t} - \bar{D}_n)^\top u \,,$$

where we have set

$$\bar{W}_{b_n,t} = -\frac{1}{b_n} \sum_{k=1}^{b_n-1} \bar{\mathbf{A}}^{-1}\varepsilon(Z_{k+t}), \qquad \bar{W}_n = -\frac{1}{n}\sum_{k=1}^{n-1}\bar{\mathbf{A}}^{-1}\varepsilon(Z_k),$$

$$\bar{D}_{b_n,t} = \frac{1}{b_n}\sum_{k=1}^{b_n-1}\Gamma_{t+1:t+k}(\theta_t - \theta^\star) + \frac{1}{b_n}\sum_{k=1}^{b_n-1}H^{(0)}_{k,t,b_n} - \frac{1}{b_n}\sum_{k=1}^{b_n-1}(Q_{k,t,b_n} - \bar{\mathbf{A}}^{-1})\varepsilon(Z_{k+t})$$

$$\bar{D}_n = \frac{1}{n}\sum_{k=0}^{n-1}\Gamma_{1:k}(\theta_0 - \theta^\star) + \frac{1}{n}\sum_{k=1}^{n-1}H^{(0)}_k - \frac{1}{n}\sum_{\ell=1}^{n-1}(Q_\ell - \bar{\mathbf{A}}^{-1})\varepsilon(Z_\ell) \,,$$

Then we can decompose the variance w.r.t. $\mathbb{P}^{\mathsf{b}}$ of $\bar{\theta}_{n,b_n}(u)$ and extract the variance of block bootstrap procedure applied to the non-observable random variables $\{\varepsilon(Z_\ell)\}$

$$\hat{\sigma}^2_\theta(u) = \frac{b_n}{n-b_n+1}\sum_{t=0}^{n-b_n}((\bar{\theta}_{b_n,t} - \bar{\theta}_n)^\top u)^2$$

$$= \underbrace{\frac{b_n}{n-b_n+1}\sum_{t=0}^{n-b_n}((\bar{W}_{b_n,t} - \bar{W}_n)^\top u)^2}_{\hat{\sigma}^2_\varepsilon(u)} + \mathcal{R}_{var}(u)$$

where we define the remainder term as follows

$$\mathcal{R}_{var}(u) = D^b_1 + D^b_2$$

$$D^b_1 = \frac{b_n}{n-b_n+1}\sum_{t=0}^{n-b_n}((\bar{D}_{b_n,t} - \bar{D}_n)^\top u)^2 \tag{63}$$

$$D^b_2 = \frac{2b_n}{n-b_n+1}\sum_{t=0}^{n-b_n}u^\top(\bar{W}_{b_n,t} - \bar{W}_n)(\bar{D}_{b_n,t} - \bar{D}_n)^\top u$$

To bound $\mathcal{R}_{var}(u)$ it remains to bound $\mathbb{E}^{1/p}_\xi[|D^b_1|^p]$ and $\mathbb{E}^{1/p}_\xi[|D^b_2|^p]$. Hence, from Lemma 18 and Lemma 19, we obtain the following bound for the residual term $\mathcal{R}_{var}(u)$:

$$\mathbb{E}^{1/p}_\xi\big[|\mathcal{R}_{var}(u)|^p\big] \lesssim p b_n^{1/2}n^{\gamma/2-1} + p^4(\log n)b_n^{1/2}n^{-\gamma} \tag{64}$$

$$+ p b_n^{-1/2}n^{\gamma/2} + p^4(\log n)n^{-1} + pn^{2\gamma-2} \,.$$

The version of this results with constants, you can see in (65). Below we present some technical lemmas that are necessary in order to bound the residual term $\mathcal{R}_{var}(u)$.

## E.2 Technical bounds

**Bounds for $\bar{D}_{b_n,t}$ and $\bar{D}_n$.** First, we establish bounds for $\bar{D}_{b_n,t}$ and $\bar{D}_n$. The argument follows the same reasoning as in the proof of the bound for $D_1$ in (27). The main difference lies in the appearance of additional terms arising from the modification of the linear component in the decomposition (62).

**Lemma 15.** *Assume A1, A2 and A3. Then it holds*

$$\mathbb{E}^{1/p}_\xi[|u^\top \bar{D}_{b_n,t}|^p] \lesssim \frac{C^{\mathsf{D,b}}_{1,1}\,p^{1/2}}{b_n\sqrt{\alpha_t}} + \frac{C^{\mathsf{D,b}}_{1,2}}{b_n} + C^{\mathsf{D,b}}_{1,3}\,p^2\sqrt{\log(1/\alpha_{b_n+t-1})}\frac{\sum_{k=1+t}^{b_n+t-1}\alpha_k}{b_n}$$

$$+ C^{\mathsf{D,b}}_{1,4}\frac{p^{1/2}}{b_n}\big(\sum_{k=2}^{b_n-1}(k+k_0+t)^{2\gamma-2}\big)^{1/2} + \frac{C^{\mathsf{D,b}}_{1,5}\,p^{1/2}b_n^{-1}}{\sqrt{\alpha_{t+b_n-2}}} + C^{\mathsf{D,b}}_{1,6}\frac{(k_0+t-1)^{\gamma-1}}{b_n} \,,$$

*where*

$$\mathsf{C}^{\mathsf{D,b}}_{1,1} = \frac{C_\Gamma d^{1/\log n + 1/2}\mathsf{D}^{(\mathrm{M})}_1 t_{\mathrm{mix}}}{a(1-\gamma)}$$

$$\mathsf{C}^{\mathsf{D,b}}_{1,2} = \frac{k_0^\gamma}{ac_0(1-\gamma)}C_\Gamma^2 d^{2/\log n + 1/2}\|\theta_0 - \theta^\star\| + t_{\mathrm{mix}}\|\varepsilon\|_\infty(\sqrt{\kappa_Q}C^{(S)}_{\gamma,a}k_0^{\gamma-1} + \sqrt{\kappa_Q}(\|\bar{\mathbf{A}}^{-1}\| + (8/a)))$$

$$\mathsf{C}^{\mathsf{D,b}}_{1,3} = \mathsf{D}^{(\mathrm{M})}_3 t_{\mathrm{mix}} + t_{\mathrm{mix}}\|\varepsilon\|_\infty\sqrt{\kappa_Q}C^{(S,2)}_{\gamma,a}$$

$$\mathsf{C}^{\mathsf{D,b}}_{1,4} = t_{\mathrm{mix}}\|\varepsilon\|_\infty\sqrt{\kappa_Q}C^{(S)}_{\gamma,a}$$

$$\mathsf{C}^{\mathsf{D,b}}_{1,5} = \frac{\|A^{-1}\|t_{\mathrm{mix}}\|\varepsilon\|_\infty\sqrt{\kappa_Q}}{\sqrt{a}}$$

$$\mathsf{C}^{\mathsf{D,b}}_{1,6} = t_{\mathrm{mix}}\|\varepsilon\|_\infty\sqrt{\kappa_Q}C^{(S)}_{\gamma,a}$$

*Proof.* We split $\bar{D}_{b_n,t}$ into three parts:

$$T_1 = \frac{1}{b_n}\sum_{k=1}^{b_n-1} u^\top \Gamma_{t+1:t+k}(\theta_t - \theta^\star)\,,$$

$$T_2 = \frac{1}{b_n}\sum_{k=1}^{b_n-1} u^\top H^{(0)}_{k,t,b_n}\,,$$

$$T_3 = -\frac{1}{b_n}\sum_{k=1}^{b_n-1} u^\top (Q_{k,t,b_n} - \bar{\mathbf{A}}^{-1})\varepsilon(Z_{k+t})\,.$$

We start from $T_1$. Using Minkowski's and Hölder's inequality, we obtain

$$\mathbb{E}^{1/p}_\xi[|u^\top T_1|^p] \le \frac{1}{b_n}\sum_{k=1}^{b_n-1}(\mathbb{E}^{1/p}_\xi[\|\Gamma_{t+1:t+k}\|^{2p}])^{1/(2p)}(\mathbb{E}^{1/p}_\xi[\|(\theta_t - \theta^\star)\|^{2p}])^{1/(2p)}\,.$$

Applying similar arguments as in (38) together with Proposition 12 and Proposition 8 we get

$$\mathbb{E}^{1/p}_\xi[|u^\top T_1|^p] \lesssim \frac{1}{b_n}\sum_{k=1}^{b_n-1} C_\Gamma d^{1/\log n + 1/2}\exp\left\{-(a/12)\sum_{\ell=t+1}^{t+k}\alpha_\ell\right\}\mathsf{D}^{(\mathrm{M})}_1 t_{\mathrm{mix}}\sqrt{p\alpha_t}$$

$$+ \frac{1}{b_n}\sum_{k=1}^{b_n-1} C_\Gamma^2 d^{2/\log n + 1/2}\exp\left\{-(a/12)\sum_{\ell=1}^{t+k}\alpha_\ell\right\}\|\theta_0 - \theta^\star\|\,.$$

Using Lemma 24, Lemma 31 and that $k_0^{1-\gamma} > \frac{24}{ac_0}$ we get

$$\mathbb{E}^{1/p}_\xi[|u^\top T_1|^p] \lesssim \frac{1}{b_n\sqrt{\alpha_t}a(1-\gamma)}C_\Gamma d^{1/\log n + 1/2}\mathsf{D}^{(\mathrm{M})}_1 t_{\mathrm{mix}}\sqrt{p}$$

$$+ \frac{k_0^\gamma\exp\{-\frac{ac_0}{24(1-\gamma)}t^{1-\gamma}\}}{b_n ac_0(1-\gamma)}C_\Gamma^2 d^{2/\log n + 1/2}\|\theta_0 - \theta^\star\|\,.$$

For $T_2$ we use Proposition 10 and Minkowski's inequality and get

$$\mathbb{E}^{1/p}_\xi[|T_2|^p] \le \frac{1}{b_n}\sum_{k=1}^{b_n-1}\mathbb{E}^{1/p}_\xi[\|u^\top H^{(0)}_{k,t,b_n}\|^p] \le \frac{\mathsf{D}^{(\mathrm{M})}_3 t_{\mathrm{mix}}p^2}{b_n}\sum_{k=1+t}^{b_n+t-1}\alpha_k\sqrt{\log(1/\alpha_k)}$$

$$\lesssim \mathsf{D}^{(\mathrm{M})}_3 t_{\mathrm{mix}}p^2\sqrt{\log(1/\alpha_{b_n+t-1})}\frac{\sum_{k=1+t}^{b_n+t-1}\alpha_k}{b_n}\,.$$

We first use Lemma 4 to show that

$$T_3 = -\frac{1}{b_n}\sum_{\ell=1}^{b_n-1} u^\top S_{\ell,t,b_n}\varepsilon(Z_{\ell+t}) + \frac{1}{b_n}\sum_{\ell=1}^{b_n-1} u^\top \bar{\mathbf{A}}^{-1}G_{\ell+t:b_n+t-1}\varepsilon(Z_{\ell+t}) = T_{31} + T_{32}\,,$$

where $S_{\ell,t,b_n} = \sum_{j=\ell+1}^{b_n-1}(\alpha_{k+t} - \alpha_{j+t})G_{l+t+1:j+t-1}$. It is easy to see that $T_{31}$ is a weighed linear statistics of MC. We apply Markov property together with Lemma 20 and obtain

$$\mathbb{E}_\xi^{1/p}[|T_{31}|^p] \lesssim \frac{t_{\mathrm{mix}}p^{1/2}\|\varepsilon\|_\infty}{b_n}\left(\sum_{k=2}^{b_n-1}\|S_{k,t,b_n}\|^2\right)^{1/2}$$
$$+ \frac{t_{\mathrm{mix}}\|\varepsilon\|_\infty}{b_n}\left(\|S_{1,t,b_n}\| + \|S_{b_n-1,t,b_n}\| + \sum_{k=1}^{b_n-2}\|S_{k+1,t,b_n} - S_{k,t,b_n}\|\right).$$

Applying Lemma 5 and Lemma 7, we get

$$\mathbb{E}_\xi^{1/p}[|T_{31}|^p] \lesssim \frac{t_{\mathrm{mix}}p^{1/2}\|\varepsilon\|_\infty\sqrt{\kappa_Q}C_{\gamma,a}^{(S)}}{b_n}\left(\sum_{k=2}^{b_n-1}(k+k_0+t)^{2\gamma-2}\right)^{1/2}$$
$$+ \frac{t_{\mathrm{mix}}\|\varepsilon\|_\infty}{b_n}\left(\sqrt{\kappa_Q}C_{\gamma,a}^{(S)}((k_0+t+1)^{\gamma-1} + (b_n+k_0+t-1)^{\gamma-1})\right.$$
$$\left.+ \sqrt{\kappa_Q}C_{\gamma,a}^{(S,2)}\sum_{k=1}^{b_n-2}\alpha_{k+1+t}\right)$$

For term $T_{32}$ we also apply Lemma 20 and get

$$\mathbb{E}_\xi^{1/p}[|T_{32}|^p] \lesssim \frac{\|\bar{\mathbf{A}}^{-1}\|t_{\mathrm{mix}}p^{1/2}\|\varepsilon\|_\infty}{b_n}\left(\sum_{k=2}^{b_n-1}\|G_{k+t:b_n+t-1}\|^2\right)^{1/2}$$
$$+ \frac{t_{\mathrm{mix}}\|\varepsilon\|_\infty}{b_n}\left(\|\bar{\mathbf{A}}^{-1}\|(\|G_{1+t:b_n+t-1}\| + \|G_{b_n+t-1:b_n+t-1}\|) + \sum_{k=1}^{b_n-2}\|\bar{\mathbf{A}}^{-1}(G_{k+1+t:b_n+t-1} - G_{k+t:b_n+t-1})\|\right).$$

Using Lemma 8, Lemma 26 and Lemma 28 with $b = a/2$ we get

$$\mathbb{E}_\xi^{1/p}[|T_{32}|^p] \lesssim \frac{\|\bar{\mathbf{A}}^{-1}\|t_{\mathrm{mix}}p^{1/2}\|\varepsilon\|_\infty\sqrt{\kappa_Q}}{\sqrt{ac_0}b_n}(b_n+t+k_0-2)^{\gamma/2}$$
$$+ \frac{t_{\mathrm{mix}}\|\varepsilon\|_\infty}{b_n}\left(\|\bar{\mathbf{A}}^{-1}\|(\alpha_{b_n+t-1}/\alpha_t + \sqrt{\kappa_Q}(1 - (a/2)\alpha_{b_n+t-1})) + \sqrt{\kappa_Q}\sum_{k=1}^{b_n-1}\alpha_{k+t}\prod_{i=k+t+1}^{b_n+t-1}(1 - (a/2)\alpha_i)\right)$$
$$\lesssim \frac{\|\bar{\mathbf{A}}^{-1}\|t_{\mathrm{mix}}p^{1/2}\|\varepsilon\|_\infty\sqrt{\kappa_Q}}{\sqrt{a}b_n\sqrt{\alpha_{b_n+t-2}}}$$
$$+ \frac{t_{\mathrm{mix}}\|\varepsilon\|_\infty\sqrt{\kappa_Q}}{b_n}\left(\|\bar{\mathbf{A}}^{-1}\| + (8/a)\right).$$

$\square$

**Lemma 16.** *Assume A1, A2 and A3. Then it holds*

$$\mathbb{E}_\xi^{1/p}[|u^\top\bar{D}_n|^p] \lesssim \frac{\mathrm{C}_{2,1}^{\mathsf{D,b}}}{n} + \mathrm{C}_{2,2}^{\mathsf{D,b}}p^2n^{-\gamma} + \mathrm{C}_{2,3}^{\mathsf{D,b}}p^{1/2}n^{\gamma-3/2}$$
$$+ \mathrm{C}_{2,4}^{\mathsf{D,b}}p^{1/2}n^{\gamma/2-1} + \mathrm{C}_{2,5}^{\mathsf{D,b}}p^2\sqrt{\log n}n^{-\gamma},$$

*where*

$$\mathrm{C}_{2,1}^{\mathsf{D,b}} = C_\Gamma d^{1/\log n} \frac{k_0^\gamma}{ac_0(1-\gamma)}\|\theta_0 - \theta^\star\| + t_{\mathrm{mix}}\|\varepsilon\|_\infty(\sqrt{\kappa_Q}(\|\bar{\mathbf{A}}^{-1}\| + 1/a) + \sqrt{\kappa_Q}C_{\gamma,a}^{(S)}k_0^{\gamma-1})$$

$$\mathrm{C}_{2,2}^{\mathsf{D,b}} = \frac{\mathsf{D}_3^{(\mathsf{M})}t_{\mathrm{mix}}\sqrt{\log\frac{k_0^\gamma}{c_0}}}{c_0(1-\gamma)} + \frac{\sqrt{\kappa_Q}t_{\mathrm{mix}}\|\varepsilon\|_\infty C_{\gamma,a}^{(S,2)}c_0}{1-\gamma}$$

$$\mathrm{C}_{2,3}^{\mathsf{D,b}} = \frac{t_{\mathrm{mix}}\|\varepsilon\|_\infty\sqrt{\kappa_Q}C_{\gamma,a}^{(S)}}{\sqrt{2\gamma-1}}$$

$$\mathrm{C}_{2,4}^{\mathsf{D,b}} = \frac{\|\bar{\mathbf{A}}^{-1}\|t_{\mathrm{mix}}\|\varepsilon\|_\infty\sqrt{\kappa_Q}k_0^{\gamma/2}}{\sqrt{ac_0}}$$

$$\mathrm{C}_{2,5}^{\mathsf{D,b}} = \frac{\mathsf{D}_3^{(\mathsf{M})}t_{\mathrm{mix}}c_0\sqrt{\gamma}}{1-\gamma}$$

*Proof.* Using (28) and (29) we get

$$\mathbb{E}_\xi^{1/p}[u^\top(\frac{1}{n}\sum_{k=0}^{n-1}\Gamma_{1:k}(\theta_0 - \theta^\star) + \frac{1}{n}\sum_{k=1}^{n-1}H_k^{(0)})|^p] \lesssim C_\Gamma d^{1/\log n}\frac{1}{n}\frac{k_0^\gamma}{ac_0(1-\gamma)}\|\theta_0 - \theta^\star\|$$

$$+ \frac{\mathsf{D}_3^{(\mathsf{M})}t_{\mathrm{mix}}p^2c_0}{1-\gamma}(\sqrt{\gamma\log n} + \sqrt{\log\frac{k_0^\gamma}{c_0}})n^{-\gamma}\ .$$

For simplicity we define

$$T_4 = -u^\top\frac{1}{n}\sum_{\ell=1}^{n-1}(Q_\ell - \bar{\mathbf{A}}^{-1})\varepsilon(Z_\ell)$$

We first use Lemma 4 to show that

$$T_4 = -\frac{1}{n}\sum_{\ell=1}^{n-1}u^\top S_\ell\varepsilon(Z_\ell) + \frac{1}{n}\sum_{\ell=1}^{n-1}u^\top\bar{\mathbf{A}}^{-1}G_{\ell:n-1}\varepsilon(Z_\ell) = T_{41} + T_{42}\ .$$

It is easy to see that $T_{41}$ is a weighed linear statistics of MC. We apply Lemma 20,

$$\mathbb{E}_\xi^{1/p}[|T_{41}|^p] \lesssim \frac{t_{\mathrm{mix}}p^{1/2}\|\varepsilon\|_\infty}{n}(\sum_{k=2}^{n-1}\|S_k\|^2)^{1/2}$$

$$+ \frac{t_{\mathrm{mix}}\|\varepsilon\|_\infty}{n}(\|S_1\| + \|S_{n-1}\| + \sum_{k=1}^{n-2}\|S_{k+1} - S_k\|)\ .$$

Applying Lemma 5 and Lemma 7, we get

$$\mathbb{E}_\xi^{1/p}[|T_{41}|^p] \lesssim \frac{t_{\mathrm{mix}}p^{1/2}\|\varepsilon\|_\infty\sqrt{\kappa_Q}C_{\gamma,a}^{(S)}}{\sqrt{2\gamma-1}}n^{\gamma-3/2}$$

$$+ \frac{t_{\mathrm{mix}}\|\varepsilon\|_\infty}{n}(\sqrt{\kappa_Q}C_{\gamma,a}^{(S)}((k_0+1)^{\gamma-1} + (n+k_0-1)^{\gamma-1}) + \frac{\sqrt{\kappa_Q}C_{\gamma,a}^{(S,2)}c_0}{1-\gamma}n^{1-\gamma})$$

$$\lesssim \frac{t_{\mathrm{mix}}p^{1/2}\|\varepsilon\|_\infty\sqrt{\kappa_Q}C_{\gamma,a}^{(S)}}{\sqrt{2\gamma-1}}n^{\gamma-3/2} + t_{\mathrm{mix}}\|\varepsilon\|_\infty\sqrt{\kappa_Q}C_{\gamma,a}^{(S)}k_0^{\gamma-1}n^{-1}$$

$$+ \frac{\sqrt{\kappa_Q}t_{\mathrm{mix}}\|\varepsilon\|_\infty C_{\gamma,a}^{(S,2)}c_0}{1-\gamma}n^{-\gamma}$$

For term $T_{42}$ we also apply Lemma 20 and get

$$\mathbb{E}_\xi^{1/p}[|T_{42}|^p] \lesssim \frac{\|\bar{\mathbf{A}}^{-1}\|t_{\mathrm{mix}}p^{1/2}\|\varepsilon\|_\infty}{n}(\sum_{k=2}^{n-1}\|G_{k:n-1}\|^2)^{1/2}$$

$$+ \frac{t_{\mathrm{mix}}\|\varepsilon\|_\infty}{n}(\|\bar{\mathbf{A}}^{-1}\|(\|G_{1:n-1}\| + \|G_{n-1:n-1}\|) + \sum_{k=1}^{n-2}\|\bar{\mathbf{A}}^{-1}(G_{k+1:n-1} - G_{k:n-1})\|)\ .$$

Using Lemma 8, Lemma 26 and Lemma 28 with $b = a/2$ we get

$$\mathbb{E}_\xi^{1/p}[|T_{32}|^p] \lesssim \frac{\|\bar{\mathbf{A}}^{-1}\| t_{\text{mix}} p^{1/2} \|\varepsilon\|_\infty \sqrt{\kappa_Q}}{\sqrt{ac_0}n}(n + k_0 - 2)^{\gamma/2}$$

$$+ \frac{t_{\text{mix}} \|\varepsilon\|_\infty}{n}\left(\|\bar{\mathbf{A}}^{-1}\|(\alpha_{n-1}/\alpha_0 + \sqrt{\kappa_Q}(1 - (a/2)\alpha_{n-1})) + \sqrt{\kappa_Q}(8/a)\right)$$

$$\lesssim \frac{\|\bar{\mathbf{A}}^{-1}\| t_{\text{mix}} p^{1/2} \|\varepsilon\|_\infty \sqrt{\kappa_Q} k_0^{\gamma/2}}{\sqrt{ac_0}} n^{\gamma/2 - 1}$$

$$+ \frac{t_{\text{mix}} \|\varepsilon\|_\infty (\|\bar{\mathbf{A}}^{-1}\| + \sqrt{\kappa_Q})}{n} .$$

$\square$

**Bounds for $\bar{W}_{b_n,t}$ and $\bar{W}_n$.** Note that the terms $\bar{W}b_n, t$ and $\bar{W}n$ are linear statistics of a Markov chain. Their moments can therefore be bounded directly using our Rosenthal-type inequality (see Lemma 20).

**Lemma 17.** *Assume A1, A2 and A3. Then in holds*

$$\mathbb{E}_\xi^{1/p}[|u^\top \bar{W}_n|^p] \lesssim \mathrm{C}_{3,1}^{\mathsf{D,b}} p^{1/2} n^{-1/2} + \frac{\mathrm{C}_{3,2}^{\mathsf{D,b}}}{n} ,$$

*and*

$$\mathbb{E}_\xi^{1/p}[|u^\top \bar{W}_{b_n,t}|^p] \lesssim \mathrm{C}_{3,1}^{\mathsf{D,b}} p^{1/2} b_n^{-1/2} + \frac{\mathrm{C}_{3,2}^{\mathsf{D,b}}}{b_n} ,$$

*where*

$$\mathrm{C}_{3,1}^{\mathsf{D,b}} = t_{\text{mix}} \|\bar{\mathbf{A}}^{-1}\| \|\varepsilon\|_\infty$$

$$\mathrm{C}_{3,2}^{\mathsf{D,b}} = t_{\text{mix}} \|\bar{\mathbf{A}}^{-1}\| .$$

*Proof.* Using Lemma 20, we get

$$\mathbb{E}_\xi^{1/p}[|u^\top \bar{W}_n|^p] \lesssim \frac{t_{\text{mix}} p^{1/2} \|\bar{\mathbf{A}}^{-1}\| \|\varepsilon\|_\infty}{\sqrt{n}} + \frac{t_{\text{mix}} \|\bar{\mathbf{A}}^{-1}\|}{n}$$

The boundary for $\bar{W}_{b_n,t}$ is obtained similarly. $\square$

**Bounds for $D_1^{\mathsf{b}}$ and $D_2^{\mathsf{b}}$** In this section, the previously obtained bounds for $\bar{W}_{b_n,t}$, $\bar{D}_{b_n,t}$, $\bar{W}_n$, and $\bar{D}_n$ are combined to establish corresponding bounds for $D_1^{\mathsf{b}}$ and $D_2^{\mathsf{b}}$ in $\mathcal{R}_{var}(u)$.

**Lemma 18.** *Assume A1, A2 and A3. Then it holds*

$$\mathbb{E}_\xi^{1/p}[|D_1^{\mathsf{b}}|^p] \lesssim M_{1,1} pn^\gamma b_n^{-1} + M_{1,2} b_n^{-1} + M_{1,3} p^4 (\log n) n^{-1} + M_{1,4} pn^{2\gamma - 2}$$

$$+ M_{2,1} b_n n^{-2} + M_{2,2} p^4 b_n (\log n) n^{-2\gamma} + M_{2,3} pb_n n^{2\gamma - 3} + M_{2,4} pb_n n^{\gamma - 2} ,$$

*where*

$$M_{1,1} = \frac{(2k_0)^{\gamma + 1}((\mathrm{C}_{1,1}^{\mathsf{D,b}})^2 + (\mathrm{C}_{1,5}^{\mathsf{D,b}})^2)}{c_0(\gamma + 1)}$$

$$M_{1,2} = (\mathrm{C}_{1,2}^{\mathsf{D,b}})^2 + (\mathrm{C}_{1,6}^{\mathsf{D,b}})^2$$

$$M_{1,3} = (\mathrm{C}_{1,3}^{\mathsf{D,b}})^2 (\gamma + \log \frac{k_0^\gamma}{c_0}) \frac{c_0^2}{2\gamma - 1}$$

$$M_{1,4} = (\mathrm{C}_{1,4}^{\mathsf{D,b}})^2 \frac{2^{2\gamma - 1} - 1}{2\gamma - 1}$$

$$M_{2,1} = (\mathrm{C}_{2,1}^{\mathsf{D,b}})^2$$

$$M_{2,2} = (\mathrm{C}_{2,2}^{\mathsf{D,b}})^2 + (\mathrm{C}_{2,5}^{\mathsf{D,b}})^2$$

$$M_{2,3} = (\mathrm{C}_{2,3}^{\mathsf{D,b}})^2$$

$$M_{2,4} = (\mathrm{C}_{2,4}^{\mathsf{D,b}})^2 .$$

*Proof.* Note that using Minkowski's inequality and Hölder's inequality, we have

$$\mathbb{E}_\xi^{1/p}[|D_1^b|^p] \le \frac{b_n}{n-b_n+1} \sum_{t=0}^{n-b_n} \left( \mathbb{E}_\xi^{1/p}[|u^\top \bar{D}_{b_n,t}|^{2p} + \mathbb{E}_\xi^{1/p}[|u^\top \bar{D}_b|^{2p} \right.$$

$$\left. + 2\mathbb{E}_\xi^{1/(2p)}[|u^\top \bar{D}_{b_n,t}|^{2p}\mathbb{E}_\xi^{1/(2p)}[|u^\top \bar{D}_b|^{2p}] \right)$$

$$\le \frac{2b_n}{n-b_n+1} \sum_{t=0}^{n-b_n} \left\{ \mathbb{E}_\xi^{1/p}[|u^\top \bar{D}_{b_n,t}|^{2p} + \mathbb{E}_\xi^{1/p}[|u^\top \bar{D}_b|^{2p} \right\}$$

Using Lemma 15 and Lemma 24 we obtain

$$\frac{2b_n}{n-b_n+1} \sum_{t=0}^{n-b_n} \mathbb{E}_\xi^{1/p}[|u^\top \bar{D}_{b_n,t}|^{2p} \lesssim \frac{b_n}{n-b_n+1} \left\{ \sum_{t=0}^{n-b_n} \frac{(\mathsf{C}_{1,1}^{\mathsf{D,b}})^2 p}{b_n^2 \alpha_t} + \frac{(\mathsf{C}_{1,2}^{\mathsf{D,b}})^2}{b_n^2} \right.$$

$$+ (\mathsf{C}_{1,3}^{\mathsf{D,b}})^2 p^4 \log(1/\alpha_{b_n+t-1}) \frac{(\sum_{k=1+t}^{b_n+t-1} \alpha_k)^2}{b_n^2}$$

$$\left. + (\mathsf{C}_{1,4}^{\mathsf{D,b}})^2 \frac{p}{b_n^2} \sum_{k=2}^{b_n-1} (k+k_0+t)^{2\gamma-2} + \frac{(\mathsf{C}_{1,5}^{\mathsf{D,b}})^2 p b_n^{-2}}{\alpha_{t+b_n-2}} + (\mathsf{C}_{1,6}^{\mathsf{D,b}})^2 \frac{(k_0+t-1)^{2\gamma-2}}{b_n^2} \right\}$$

$$\lesssim M_{1,1} p n^\gamma b_n^{-1} + M_{1,2} b_n^{-1} + M_{1,3} p^4 \log n n^{-1} + M_{1,4} p n^{2\gamma-2}$$

Using Lemma 16, we get

$$\frac{2b_n}{n-b_n+1} \sum_{t=0}^{n-b_n} \mathbb{E}_\xi^{1/p}[|u^\top \bar{D}_b|^{2p} \lesssim M_{2,1} b_n n^{-2} + M_{2,2} p^4 b_n (\log n) n^{-2\gamma} + M_{2,3} p b_n n^{2\gamma-3} + M_{2,4} p b_n n^{\gamma-2} .$$

$$\square$$

**Lemma 19.** *Assume A1, A2 and A3. Then it holds*

$$\mathbb{E}_\xi^{1/p}[|D_2^b|^p] \lesssim M_{3,1} p^{1/2} b_n^{1/2} n^{-1} + M_{3,2} p b_n^{1/2} n^{\gamma-3/2} + M_{3,3} p b_n^{1/2} n^{\gamma/2-1} + M_{3,4} p^{5/2} \sqrt{\log n} n^{-\gamma} b_n^{1/2}$$

$$+ M_{3,5} p^{1/2} b_n^{-1/2} + M_{3,6} p^{5/2} \sqrt{\log n} b_n^{1/2} n^{-\gamma} + M_{3,7} p b_n^{-1/2} n^{\gamma-1/2} + M_{3,8} p b_n^{-1/2} n^{\gamma/2} ,$$

*where*

$$M_{3,1} = \mathsf{C}_{2,1}^{\mathsf{D,b}}(\mathsf{C}_{3,1}^{\mathsf{D,b}} + \mathsf{C}_{3,2}^{\mathsf{D,b}})$$

$$M_{3,2} = \mathsf{C}_{2,3}^{\mathsf{D,b}}(\mathsf{C}_{3,1}^{\mathsf{D,b}} + \mathsf{C}_{3,2}^{\mathsf{D,b}})$$

$$M_{3,3} = \mathsf{C}_{2,4}^{\mathsf{D,b}}(\mathsf{C}_{3,1}^{\mathsf{D,b}} + \mathsf{C}_{3,2}^{\mathsf{D,b}})$$

$$M_{3,4} = (\mathsf{C}_{2,5}^{\mathsf{D,b}} + \mathsf{C}_{2,2}^{\mathsf{D,b}})(\mathsf{C}_{3,1}^{\mathsf{D,b}} + \mathsf{C}_{3,2}^{\mathsf{D,b}})$$

$$M_{3,5} = (\mathsf{C}_{1,2}^{\mathsf{D,b}} + \mathsf{C}_{1,6}^{\mathsf{D,b}})(\mathsf{C}_{3,1}^{\mathsf{D,b}} + \mathsf{C}_{3,2}^{\mathsf{D,b}})$$

$$M_{3,6} = \frac{\mathsf{C}_{1,3}^{\mathsf{D,b}}}{1-\gamma}(\sqrt{\gamma} + \sqrt{\log \frac{k_0^\gamma}{c_0}})(\mathsf{C}_{3,1}^{\mathsf{D,b}} + \mathsf{C}_{3,2}^{\mathsf{D,b}})$$

$$M_{3,7} = \frac{\mathsf{C}_{1,4}^{\mathsf{D,b}}(2k_0)^{\gamma+1/2}}{\sqrt{2\gamma-1}(\gamma+1/2)}(\mathsf{C}_{3,1}^{\mathsf{D,b}} + \mathsf{C}_{3,2}^{\mathsf{D,b}})$$

$$M_{3,8} = \frac{(2k_0)^{\gamma/2+1}(\mathsf{C}_{1,5}^{\mathsf{D,b}} + \mathsf{C}_{1,1}^{\mathsf{D,b}})}{\sqrt{c_0}(\gamma/2+1)}(\mathsf{C}_{3,1}^{\mathsf{D,b}} + \mathsf{C}_{3,2}^{\mathsf{D,b}})$$

*Proof.* Note that using Minkowski's inequality together with Hölder's inequality we get

$$\mathbb{E}_\xi^{1/p}[|D_2^b|^p] \lesssim \frac{b_n}{n-b_n+1} \sum_{t=0}^{n-b_n} \mathbb{E}_\xi^{1/(2p)}[|u^\top \bar{W}_{b_n,t}|^{2p}](\mathbb{E}_\xi^{1/(2p)}[|u^\top \bar{D}_{b_n,t}|^{2p}] + \mathbb{E}_\xi^{1/(2p)}[|u^\top \bar{D}_n|^{2p}])$$

$$+ \frac{b_n}{n-b_n+1} \sum_{t=0}^{n-b_n} \mathbb{E}_\xi^{1/(2p)}[|u^\top \bar{W}_n|^{2p}](\mathbb{E}_\xi^{1/(2p)}[|u^\top \bar{D}_{b_n,t}|^{2p}] + \mathbb{E}_\xi^{1/(2p)}[|u^\top \bar{D}_n|^{2p}])$$

Note that bound on $\mathbb{E}_\xi^{1/(2p)}[|u^\top \bar{W}_{b_n,t}|^{2p}]$, $\mathbb{E}_\xi^{1/(2p)}[|u^\top \bar{W}_n|^{2p}]$, $\mathbb{E}_\xi^{1/(2p)}[|u^\top \bar{D}_n|^{2p}]$ does not depend upon $t$, hence, we can rewrite formula above as follows

$$\mathbb{E}_\xi^{1/p}[|D_2^b|^p] \lesssim \frac{b_n}{n - b_n + 1}\left\{\sum_{t=0}^{n-b_n} \mathbb{E}_\xi^{1/(2p)}[|u^\top \bar{D}_{b_n,t}|^{2p}]\right\}(\mathbb{E}_\xi^{1/(2p)}[|u^\top \bar{W}_{b_n,t}|^{2p}] + \mathbb{E}_\xi^{1/(2p)}[|u^\top \bar{W}_n|^{2p}])$$
$$+ b_n \mathbb{E}_\xi^{1/(2p)}[|u^\top \bar{D}_n|^{2p}](\mathbb{E}_\xi^{1/(2p)}[|u^\top \bar{W}_{b_n,t}|^{2p}] + \mathbb{E}_\xi^{1/(2p)}[|u^\top \bar{W}_n|^{2p}])$$

Applying Lemma 15 and Lemma 24 we get

$$\frac{b_n}{n - b_n + 1}\sum_{t=0}^{n-b_n} \mathbb{E}_\xi^{1/(2p)}[|(u^\top \bar{D}_{b_n,t})|^{2p}] \lesssim \frac{b_n}{n - b_n + 1}\sum_{t=0}^{n-b_n}\left\{\frac{C_{1,1}^{\mathsf{D,b}}\, p^{1/2}}{b_n\sqrt{\alpha_t}} + \frac{C_{1,2}^{\mathsf{D,b}}}{b_n}\right.$$
$$+ C_{1,3}^{\mathsf{D,b}}\, p^2\sqrt{\log(1/\alpha_{b_n+t-1})}\frac{\sum_{k=1+t}^{b_n+t-1}\alpha_k}{b_n}$$
$$\left. + C_{1,4}^{\mathsf{D,b}}\frac{p^{1/2}}{b_n}(\sum_{k=2}^{b_n-1}(k+k_0+t)^{2\gamma-2})^{1/2} + \frac{C_{1,5}^{\mathsf{D,b}}\, p^{1/2}b_n^{-1}}{\sqrt{\alpha_{t+b_n-2}}} + C_{1,6}^{\mathsf{D,b}}\frac{(k_0+t-1)^{\gamma-1}}{b_n}\right\}$$

Using Lemma 16 and Lemma 17 we get

$$\mathbb{E}_\xi^{1/p}[|D_2^b|^p] \lesssim \left\{C_{2,1}^{\mathsf{D,b}}\, b_n n^{-1} + C_{2,3}^{\mathsf{D,b}}\, p^{1/2}b_n n^{\gamma-3/2} + C_{2,4}^{\mathsf{D,b}}\, p^{1/2}b_n n^{\gamma/2-1}\right.$$
$$+ (C_{2,5}^{\mathsf{D,b}} + C_{2,2}^{\mathsf{D,b}})p^2\sqrt{\log n}\, n^{-\gamma}b_n + C_{1,2}^{\mathsf{D,b}} + \frac{C_{1,3}^{\mathsf{D,b}}}{1-\gamma}p^2(\sqrt{\gamma} + \sqrt{\log\frac{k_0^\gamma}{c_0}})\sqrt{\log n}\, b_n n^{-\gamma}$$
$$\left. + \frac{C_{1,4}^{\mathsf{D,b}}(2k_0)^{\gamma+1/2}}{\sqrt{2\gamma-1}(\gamma+1/2)}p^{1/2}n^{\gamma-1/2} + \frac{(2k_0)^{\gamma/2+1}(C_{1,5}^{\mathsf{D,b}} + C_{1,1}^{\mathsf{D,b}})p^{1/2}(n-b_n+1)^{\gamma/2}}{\sqrt{c_0}(\gamma/2+1)}\right\}.$$
$$\cdot (C_{3,1}^{\mathsf{D,b}} + C_{3,2}^{\mathsf{D,b}})p^{1/2}(n^{-1/2} + b_n^{-1/2})$$

$\square$

**Version of** (64) **with constants.** Finally, from Lemma 18 and Lemma 19, we obtain the following bound for the residual term $\mathcal{R}_{var}(u)$:

$$\mathbb{E}_\xi^{1/p}\big[|\mathcal{R}_{var}(u)|^p\big] \lesssim M_1 p b_n^{1/2} n^{\gamma/2-1} + M_2 p^4(\log n)b_n^{1/2}n^{-\gamma} \tag{65}$$
$$+ M_3 p b_n^{-1/2}n^{\gamma/2} + M_4 p^4(\log n)n^{-1} + M_5 p n^{2\gamma-2}\,,$$

where

$$\begin{aligned}
M_1 &= M_{3,1} + M_{3,2} + M_{3,3} + M_{2,3} + M_{2,4} \\
M_2 &= M_{3,4} + M_{3,6} + M_{2,2} \\
M_3 &= M_{3,5} + M_{3,7} + M_{3,8} + M_{1,1} + M_{1,2} \\
M_4 &= M_{1,3} + M_{2,1} \\
M_5 &= M_{1,4}\,.
\end{aligned} \tag{66}$$

### E.3 Proof of Corollary 2

The proof of Corollary 2 follows directly from From Proposition 3 and Proposition 4. Indeed, note that

$$
\begin{aligned}
\mathbb{E}_\xi^{1/p}[|\hat{\sigma}_\theta^2(u) - \sigma^2(u)|^p] &\leq \mathbb{E}_\xi^{1/p}[|\hat{\sigma}_\varepsilon^2(u) - \sigma^2(u)|^p] + \mathbb{E}_\xi^{1/p}[|\mathcal{R}_{var}(u)|^p] \\
&\lesssim \frac{p t_{\mathrm{mix}}^3 \|\varepsilon\|_\infty^2}{\sqrt{n}} + \frac{p^2 t_{\mathrm{mix}}^2 \sqrt{b_n} \|\varepsilon\|_\infty^2}{\sqrt{n}} + \frac{p t_{\mathrm{mix}}^2 \|\varepsilon\|_\infty^2}{\sqrt{b_n}} \\
&\quad + M_1 p b_n^{1/2} n^{\gamma/2 - 1} + M_2 p^4 (\log n) b_n^{1/2} n^{-\gamma} \\
&\quad + M_3 p b_n^{-1/2} n^{\gamma/2} + M_4 p^4 (\log n) n^{-1} + M_5 p n^{2\gamma - 2} \\
&\lesssim p^4 (t_{\mathrm{mix}}^2 \|\varepsilon\|_\infty^2 + M_2 + M_1)(\log n) \frac{b_n^{1/2}}{\sqrt{n}} + p^4 (t_{\mathrm{mix}}^2 \|\varepsilon\|_\infty^2 + M_3) \frac{n^{\gamma/2}}{\sqrt{b_n}} \\
&\quad + p^4 (t_{\mathrm{mix}}^3 \|\varepsilon\|_\infty^2 + M_4) \frac{(\log n)}{\sqrt{n}} + p^4 M_5 n^{2\gamma - 2}
\end{aligned}
$$

Hence, applying Lemma 21 with $p = \log n$ we obtain that with probability $1 - \frac{1}{n}$ it holds

$$
\begin{aligned}
\left|\hat{\sigma}_\theta^2(u) - \sigma^2(u)\right| &\lesssim (t_{\mathrm{mix}}^2 \|\varepsilon\|_\infty^2 + M_2 + M_1)(\log n)^5 \frac{b_n^{1/2}}{\sqrt{n}} \\
&\quad + (\log n)^4 (t_{\mathrm{mix}}^2 \|\varepsilon\|_\infty^2 + M_3) \frac{n^{\gamma/2}}{\sqrt{b_n}} \\
&\quad + (t_{\mathrm{mix}}^3 \|\varepsilon\|_\infty^2 + M_4) \frac{(\log n)^5}{\sqrt{n}} \\
&\quad + (\log n)^4 M_5 n^{2\gamma - 2}
\end{aligned}
\tag{67}
$$

By setting $b_n = b^\beta$, we can optimize the inequality above by $\gamma$ and $\beta$. And by putting $\beta = 3/4$ and $\gamma = 1/2 + \varepsilon$ where $\varepsilon < 1/\log n$ we finally get that with probability $1 - \frac{1}{n}$ it holds

$$
\begin{aligned}
&\left|\hat{\sigma}_\theta^2(u) - \sigma^2(u)\right| \\
&\lesssim (t_{\mathrm{mix}}^2 \|\varepsilon\|_\infty^2 + M_2 + M_1 + t_{\mathrm{mix}}^2 \|\varepsilon\|_\infty^2 + M_3 + t_{\mathrm{mix}}^3 \|\varepsilon\|_\infty^2 + M_4 + M_5)(\log n)^5 n^{-1/8 + \varepsilon/2}
\end{aligned}
$$

### E.4 Detailed version of the proof of Theorem 2

We preface the proof with a Gaussian comparison inequality due to [21, Theorem 1.3]. It states that for $\xi_i \sim \mathcal{N}(0, \sigma_i^2)$, $i = 1, 2$,

$$
\sup_{x \in \mathbb{R}} |\mathbb{P}(\xi_1 \leq x) - \mathbb{P}(\xi_2 \leq x)| \leq (3/2)|\sigma_1^2/\sigma_2^2 - 1|.
\tag{68}
$$

Now we proceed with the proof of Theorem 2. Using (67) together with the inequality $\sigma^{-2}(u) \leq 1/\lambda_{\min}(\Sigma_\infty)$ we get that with probability $1 - 1/n$ it holds

$$
\begin{aligned}
\left|\hat{\sigma}_\theta^2(u) - \sigma^2(u)\right|/\sigma^2(u) &\lesssim \frac{\mathrm{e}}{\lambda_{\min}(\Sigma_\infty)} (t_{\mathrm{mix}}^2 \|\varepsilon\|_\infty^2 + M_2 + M_1)(\log n)^5 \frac{b_n^{1/2}}{\sqrt{n}} \\
&\quad + \frac{\mathrm{e}}{\lambda_{\min}(\Sigma_\infty)} (\log n)^4 (t_{\mathrm{mix}}^2 \|\varepsilon\|_\infty^2 + M_3) \frac{n^{\gamma/2}}{\sqrt{b_n}} \\
&\quad + \frac{\mathrm{e}}{\lambda_{\min}(\Sigma_\infty)} (t_{\mathrm{mix}}^3 \|\varepsilon\|_\infty^2 + M_4) \frac{(\log n)^5}{\sqrt{n}} \\
&\quad + \frac{\mathrm{e}}{\lambda_{\min}(\Sigma_\infty)} (\log n)^4 M_5 n^{2\gamma - 2}
\end{aligned}
$$

Now we apply (68) and obtain that with probability $1 - 1/n$ it holds

$$
\begin{aligned}
\mathsf{d}_K\big(\mathcal{N}(0,\hat{\sigma}_\theta^2(u)),\mathcal{N}(0,\sigma^2(u))\big) &\lesssim \frac{1}{\lambda_{\min}(\Sigma_\infty)}(t_{\mathrm{mix}}^2\|\varepsilon\|_\infty^2 + M_2 + M_1)(\log n)^5 \frac{b_n^{1/2}}{\sqrt{n}} \\
&\quad + \frac{1}{\lambda_{\min}(\Sigma_\infty)}(\log n)^4(t_{\mathrm{mix}}^2\|\varepsilon\|_\infty^2 + M_3)\frac{n^{\gamma/2}}{\sqrt{b_n}} \\
&\quad + \frac{1}{\lambda_{\min}(\Sigma_\infty)}(t_{\mathrm{mix}}^3\|\varepsilon\|_\infty^2 + M_4)\frac{(\log n)^5}{\sqrt{n}} \\
&\quad + \frac{1}{\lambda_{\min}(\Sigma_\infty)}(\log n)^4 M_5 n^{2\gamma-2} \;.
\end{aligned}
$$

Combining this inequality with Corollary 1 we obtain with probability $1 - 1/n$

$$
\begin{aligned}
\sup_{x\in\mathbb{R}} |\mathbb{P}(\sqrt{n}(\bar{\theta}_n - \theta^\star)^\top u \le x) - \mathbb{P}^\flat(\bar{\theta}_{n,b_n}(u) \le x)| &\lesssim (\mathrm{C}_{K,1} + \mathrm{C}_{K,2} + \mathrm{C}_1^{\mathsf{D}}\|\theta_0 - \theta^\star\| + \mathrm{C}_2^{\mathsf{D}})\frac{\log n}{n^{1/4}} \\
&\quad + (\mathrm{C}_4^{\mathsf{D}} + \mathrm{C}_3^{\mathsf{D}})\frac{(\log n)^{5/2}}{n^{\gamma-1/2}} \\
&\quad + \frac{1}{\lambda_{\min}(\Sigma_\infty)}(t_{\mathrm{mix}}^2\|\varepsilon\|_\infty^2 + M_2 + M_1)(\log n)^5 \frac{b_n^{1/2}}{\sqrt{n}} \\
&\quad + \frac{1}{\lambda_{\min}(\Sigma_\infty)}(\log n)^4(t_{\mathrm{mix}}^2\|\varepsilon\|_\infty^2 + M_3)\frac{n^{\gamma/2}}{\sqrt{b_n}} \\
&\quad + \frac{1}{\lambda_{\min}(\Sigma_\infty)}(t_{\mathrm{mix}}^3\|\varepsilon\|_\infty^2 + M_4)\frac{(\log n)^5}{\sqrt{n}} \\
&\quad + \frac{1}{\lambda_{\min}(\Sigma_\infty)}(\log n)^4 M_5 n^{2\gamma-2}
\end{aligned}
$$

To complete the proof, it remains to optimize the bound above. Setting $b_n = \lceil n^{4/5}\rceil$ and $\gamma = 3/5$, we obtain with probability $1 - 1/n$

$$
\begin{aligned}
\sup_{x\in\mathbb{R}} |\mathbb{P}(\sqrt{n}(\bar{\theta}_n - \theta^\star)^\top u \le x) - \mathbb{P}^\flat(\bar{\theta}_{n,b_n}(u) \le x)| &\lesssim (\mathrm{C}_{K,1} + \mathrm{C}_{K,2} + \mathrm{C}_1^{\mathsf{D}}\|\theta_0 - \theta^\star\| + \mathrm{C}_2^{\mathsf{D}})\frac{\log n}{n^{1/4}} \\
&\quad + \frac{1}{\lambda_{\min}(\Sigma_\infty)}(t_{\mathrm{mix}}^3\|\varepsilon\|_\infty^2 + M_4)\frac{(\log n)^5}{\sqrt{n}} \\
&\quad + \left(\mathrm{C}_3^{\mathsf{D}} + \mathrm{C}_4^{\mathsf{D}} + \frac{t_{\mathrm{mix}}^3\|\varepsilon\|_\infty^2 + M_1 + M_2 + M_3 + M_5}{\lambda_{\min}(\Sigma_\infty)}\right)(\log n)^5 n^{-1/10} \;.
\end{aligned}
$$

## F   Probability inequalities

Denote by $\Phi$ the c.d.f. of a standard Gaussian random variable and set
$$
\mathsf{d}_K\big(X\big) = \sup_{x\in\mathbb{R}} |\mathbb{P}(X \le x) - \Phi(x)|.
$$

**Proposition 14.** *For any random variables $X, Y$, and any $p \ge 1$,*
$$
\mathsf{d}_K\big(X + Y\big) \le \mathsf{d}_K\big(X\big) + 2\mathbb{E}^{1/(p+1)}[|Y|^p] \;. \tag{69}
$$

*Proof.* Let $t \ge 0$. By Markov's inequality

$$
\begin{aligned}
\mathbb{P}(X + Y \le x) &\le \mathbb{P}(X + Y \le x, |Y| \le t) + \frac{1}{t^p}\mathbb{E}[|Y|^p] \\
&\le \mathbb{P}(X \le x + t) - \Phi(x + t) + \Phi(x + t) + \frac{1}{t^p}\mathbb{E}[|Y|^p] \\
&\le \Phi(x) + \sup_{x\in\mathbb{R}} |\mathbb{P}(X \le x) - \Phi(x)| + \frac{t}{\sqrt{2\pi}} + \frac{1}{t^p}\mathbb{E}[|Y|^p].
\end{aligned}
$$

Choosing $t = \mathbb{E}^{1/(p+1)}[|Y|^p]$ we obtain

$$\sup_{x \in \mathbb{R}}(\mathbb{P}(X + Y \leq x) - \Phi(x)) \leq \sup_{x \in \mathbb{R}} |\mathbb{P}(X \leq x) - \Phi(x)| + 2\mathbb{E}^{1/(p+1)}[|Y|^p]$$

Similarly, we may estimate $\sup_{x \in \mathbb{R}}(\Phi(x) - \mathbb{P}(X + Y \leq x))$. Hence, (69) holds. $\qquad\square$

**Remark 2.** *The result similar to Proposition 14 was previously obtained in [44, Lemma 1]. It states that for random variables $X$ and $Y$ and any $p \geq 1$, it holds that*

$$\mathsf{d}_K(X + Y) \leq 2\mathsf{d}_K(X) + 3\|\mathbb{E}[|Y|^{2p}|X]\|_1^{1/(2p+1)}.$$

## F.1 Rosenthal and Burkholder inequalities

We begin this section with a version of Rosenthal inequality (see the original paper [61] and the Pinelis version of the Rosenthal inequality [56]). Let $f : \mathsf{Z} \to \mathbb{R}$ be a bounded function with $\|f\|_\infty < \infty$ and define

$$\bar{S}_n = \sum_{k=0}^{n-1} \{f(Z_k) - \pi(f)\} .$$

Then the following bound holds:

**Lemma 20.** *Assume A 1. Then for any $p \geq 2$ and $f : \mathsf{Z} \to \mathbb{R}^d$ with $\|f\|_\infty < \infty$, any initial distribution $\xi$ on $(\mathsf{Z}, \mathcal{Z})$, and any $u \in \mathbb{S}_{d-1}, A_i \in \mathbb{R}^{d \times d}$, it holds that*

$$\mathbb{E}_\xi^{1/p}\Big[\Big|\sum_{k=1}^{n} u^\top A_k (f(Z_k) - \pi(f))\Big|^p\Big] \leq (16/3) t_{\mathrm{mix}} p^{1/2} \|f\|_\infty \Big(\sum_{k=2}^{n} \|A_k\|^2\Big)^{1/2}$$

$$+ (8/3) t_{\mathrm{mix}}\Big(\|A_1\| + \|A_n\| + \sum_{k=1}^{n-1} \|A_{k+1} - A_k\|\Big)\|f\|_\infty .$$

*Proof.* Under A1 the Poisson equation

$$g(z) - \mathrm{P}g(z) = f(z) - \pi(f)$$

has a unique solution for any bounded $f$, which is given by the formula

$$g(z) = \sum_{k=0}^{\infty} \{\mathrm{P}^k f(z) - \pi(f)\} .$$

Thus, using A1, we obtain that $g(z)$ is also bounded with

$$\|g(z)\| \leq \sum_{k=0}^{+\infty} \|\mathrm{P}^k f(z) - \pi(f)\| \leq 2\|f\|_\infty \sum_{k=0}^{+\infty} (1/4)^{\lfloor k/t_{\mathrm{mix}}\rfloor} \leq (8/3) t_{\mathrm{mix}} \|f\|_\infty . \tag{70}$$

Hence, we can represent

$$\sum_{k=1}^{n} u^\top A_k (f(Z_k) - \pi(f)) = \underbrace{\sum_{k=2}^{n} u^\top A_k (g(Z_k) - \mathrm{P}g(Z_{k-1}))}_{T_1}$$

$$+ \underbrace{\sum_{k=1}^{n-1} u^\top (A_{k+1} - A_k)\mathrm{P}g(Z_k) + u^\top A_1 g(Z_1) - u^\top A_n \mathrm{P}g(Z_n)}_{T_2} .$$

The term $T_2$ can be controlled using Minkowski's inequality:

$$\mathbb{E}_\xi^{1/p}[|T_2|^p] \leq (8/3) t_{\mathrm{mix}}\Big(\|A_1\| + \|A_n\| + \sum_{k=1}^{n-1} \|A_{k+1} - A_k\|\Big)\|f\|_\infty .$$

Now we proceed with $T_1$. Since $\mathbb{E}^{\mathcal{F}_{k-1}}[g(Z_k) - Pg(Z_{k-1})] = 0$ a.s. and $|u^\top A_k(g(Z_k) - Pg(Z_{k-1}))| \leq (16/3)\|A_k\|t_{\text{mix}}\|f\|_\infty$, we get, using the Azuma-Hoeffding inequality [78, Corollary 3.9], that

$$\mathbb{P}_\xi[|T_1| \geq t] \leq 2\exp\left\{-\frac{2t^2}{(16/3)^2 t_{\text{mix}}^2 \|f\|_\infty^2 \sum_{k=2}^n \|A_k\|^2}\right\}.$$

Hence, applying [24, Lemma 7], we get

$$\mathbb{E}_\xi^{1/p}[|T_1|^p] \leq (16/3)p^{1/2} t_{\text{mix}}\|f\|_\infty (\sum_{k=2}^n \|A_k\|^2)^{1/2}.$$

$\square$

The lemma below is a simple technical statement used to switch between $p$-th moment bounds and high-probability bounds.

**Lemma 21.** *Fix $\delta \in (0, 1/e^2)$ and let $Y$ be a positive random variable, such that*

$$\mathbb{E}^{1/p}[Y^p] \leq p^\upsilon C_1$$

*for any $2 \leq p \leq \log(1/\delta)$. Then it holds with probability at least $1 - \delta$, that*

$$Y \leq eC_1(\log(1/\delta))^\upsilon.$$

*Proof.* Applying Markov's inequality, for any $t \geq 0$ we get that

$$\mathbb{P}(Y \geq t) \leq \frac{\mathbb{E}[Y^p]}{t^p} \leq \frac{(C_1 p^\upsilon)^p}{t^p}.$$

Now we set $p = \log(1/\delta)$, $t = eC_1(\log(1/\delta))^\upsilon$, and aim to check that

$$\frac{(C_1(\log(1/\delta))^\upsilon)^{\log(1/\delta)}}{(eC_1(\log(1/\delta))^\upsilon)^{\log(1/\delta)}} \leq \delta.$$

Taking logarithms from both sides, the latter inequality is equivalent to

$$-\log(1/\delta) \leq \log\delta,$$

which turns into exact equality. $\square$

# G  Auxiliary results on the sequences of step sizes $\{\alpha_k\}_{k\in\mathbb{N}}$ under A3

In this section we provide the auxiliary results on the sequences of step sizes $\{\alpha_k\}_{k\in\mathbb{N}}$ under A3.

For simplicity, we denote

$$g_{n:m} = \sum_{k=n}^m k^{-\gamma}, \ n \leq m.$$

**Lemma 22.** *Let $n \leq m$. Then*

$$\frac{(m+1)^{1-\gamma} - n^{1-\gamma}}{1-\gamma} \leq g_{n:m} \leq \frac{m^{1-\gamma} - (n-1)^{1-\gamma}}{1-\gamma}.$$

**Lemma 23.** *Let $b, c_0 > 0$ and $\alpha_\ell = c_0(\ell + k_0)^{-\gamma}$ for $\gamma \in (1/2, 1)$, $k_0 \geq 0$. Assume that $bc_0 < 1$ and $k_0^{1-\gamma} \geq 2/(bc_0)$. Then*

$$\sum_{k=\ell}^{n-1} \alpha_\ell \prod_{j=\ell+1}^k (1 - b\alpha_j) \leq \mathcal{L}_b,$$

*where we set*

$$\mathcal{L}_b = c_0 + \frac{2}{b(1-\gamma)}.$$

*Proof.* Note that

$$\sum_{k=\ell}^{n-1} \prod_{j=\ell+1}^{k} (1 - b\alpha_j) \le \sum_{k=\ell}^{n-1} \exp\left\{ -b \sum_{j=l+1}^{k} \alpha_j \right\} \le \sum_{k=\ell+k_0}^{n+k_0-1} \exp\left\{ -\frac{bc_0}{2(1-\gamma)} (k^{1-\gamma} - (l+k_0)^{1-\gamma}) \right\}$$

Applying Lemma 31 with $k_0^{1-\gamma} \ge 2/(bc_0)$, we finish the proof. □

**Lemma 24.** *Assume A3. Then the following bounds holds:*

1.

$$\sum_{i=1}^{k} \alpha_i \le \frac{c_0}{1-\gamma} ((k+k_0)^{1-\gamma} - k_0^{1-\gamma})$$

2. *for any $p \ge 2$*

$$\sum_{i=1}^{k} \alpha_i^p \le \frac{c_0^p}{p\gamma - 1} ,$$

3. *for any $m \in \{0, \ldots, k\}$*

$$\sum_{i=m+1}^{k} \alpha_i \ge \frac{c_0}{2(1-\gamma)} ((k+k_0)^{1-\gamma} - (m+k_0)^{1-\gamma}) ,$$

*Proof.* To proof 1, note that

$$\sum_{i=1}^{k} \alpha_i \le c_0 \int_{k_0}^{k+k_0} \frac{\mathrm{d}x}{x^\gamma} \le \frac{c_0}{1-\gamma} ((k+k_0)^{1-\gamma} - k_0^{1-\gamma}) ,$$

To proof 2, note that

$$\sum_{i=1}^{k} \alpha_i^p \le c_0^p \int_{1}^{+\infty} \frac{\mathrm{d}x}{x^{p\gamma}} \le \frac{c_0^p}{p\gamma - 1} ,$$

To proof 3, note that for any $i \ge 1$ we have $2(i+k_0)^{-\gamma} \ge (i+k_0-1)^{-\gamma}$. Hence,

$$\sum_{i=m+1}^{k} \alpha_i \ge \frac{1}{2} \sum_{i=m}^{k-1} \alpha_i \ge \frac{c_0}{2} \int_{m+k_0}^{k+k_0} \frac{\mathrm{d}x}{x^\gamma} = \frac{c_0}{2(1-\gamma)} ((k+k_0)^{1-\gamma} - (m+k_0)^{1-\gamma}) .$$

□

**Lemma 25** (Lemma 24 in [26]). *Let $b > 0$ and $\{\alpha_k\}_{k \ge 0}$ be a non-increasing sequence such that $\alpha_1 \le 1/b$. Then*

$$\sum_{j=1}^{k} \alpha_j \prod_{l=j+1}^{k} (1 - \alpha_l b) = \frac{1}{b} \left\{ 1 - \prod_{l=1}^{k} (1 - \alpha_l b) \right\}$$

*Proof.* The proof is given in [26]. □

**Lemma 26.** *Let $b > 0$ and $\alpha_k = \frac{c_0}{(k_0+k)^\gamma}$ be a non-increasing sequence such that $c_0 \le 1/b$ and $k_0 \ge \{\frac{\gamma}{2bc_0}\}^{1/(1-\gamma)}$. Then it holds*

$$\alpha_j \prod_{l=j+1}^{k} (1 - \alpha_l b) \le \alpha_k$$

*Proof.* Note that

$$\alpha_j \prod_{l=j+1}^{k} (1 - \alpha_l b) = \alpha_k \prod_{l=j+1}^{k} \frac{\alpha_{l-1}}{\alpha_l} (1 - \alpha_l b) .$$

It remains to note that,

$$\frac{\alpha_{l-1}}{\alpha_l}(1-\alpha_l b) = \left\{\frac{k_0+l}{k_0+l-1}\right\}^\gamma - \frac{bc_0}{(k_0+l-1)^\gamma} \leq 1 + \frac{\gamma}{k_0+l-1} - \frac{bc_0}{(k_0+l-1)^\gamma} \leq 1 - (b/2)\alpha_{l-1} \,,$$

where the last inequality holds since $k_0 \geq \{\frac{2\gamma}{bc_0}\}^{1/(1-\gamma)}$. $\qquad\square$

**Lemma 27.** *Let $b > 0$, and let $\alpha_k = c_0/(k+k_0)^\gamma$, $\gamma \in (0;1)$, such that $c_0 \leq 1/b$ and $k_0 \geq \{\frac{2\gamma}{rc_0}\}^{1/1-\gamma}$ with some constant $r > 0$. Then it holds that*

$$\frac{\alpha_k}{\alpha_{k+1}} \leq 1 + r\alpha_{k+1}$$

*Proof.* Note that

$$\frac{\alpha_k}{\alpha_{k+1}} \leq \left(1 + \frac{1}{k+k_0}\right)^\gamma \leq 1 + \frac{2\gamma}{k_0+k+1} \leq 1 + \frac{rc_0}{(k_0+k+1)^\gamma} \,,$$

where the last inequality holds since $k_0 \geq \{\frac{2\gamma}{rc_0}\}^{1/1-\gamma}$. $\qquad\square$

**Lemma 28.** *Let $b > 0$, and let $\alpha_k = c_0/(k+k_0)^\gamma$, $\gamma \in (0;1)$, such that $c_0 b \leq 1/2$ and $k_0 \geq \{\frac{8\gamma}{bc_0}\}^{1/1-\gamma}$. Then for any $q \in (1;3]$, it holds that*

$$\sum_{j=1}^k \alpha_j^q \prod_{\ell=j+1}^k (1-\alpha_\ell b) \leq \frac{4}{b}\alpha_k^{q-1} \,.$$

*Proof.* Using Lemma 27, we obtain that

$$\sum_{j=1}^k \alpha_j^q \prod_{\ell=j+1}^k (1-\alpha_\ell b) = \alpha_k^{q-1} \sum_{j=1}^k \alpha_j \prod_{\ell=j+1}^k \left(\frac{\alpha_{\ell-1}}{\alpha_\ell}\right)^{q-1} (1-\alpha_\ell b)$$

$$\leq \alpha_k^{q-1} \sum_{j=1}^k \alpha_j \prod_{\ell=j+1}^k (1+r\alpha_\ell)^{q-1}(1-\alpha_\ell b) \,.$$

We set $r = \frac{b}{2(q-1)}$. If $q \in (1,2)$ then we use Bernoulli's inequality and obtain

$$\sum_{j=1}^k \alpha_j^q \prod_{\ell=j+1}^k (1-\alpha_\ell b) \leq \alpha_k^{q-1} \sum_{j=1}^k \alpha_j \prod_{\ell=j+1}^k (1+b\alpha_\ell/2)(1-\alpha_\ell b)$$

$$\leq \alpha_k^{q-1} \sum_{j=1}^k \alpha_j \prod_{\ell=j+1}^k (1-b\alpha_\ell/2) \overset{(a)}{\leq} \frac{2}{b}\alpha_k^{q-1} \,,$$

where in (a) we used Lemma 25. If $q \in [2,3]$, using that $1-\alpha_\ell b \leq \left(1-b/(q-1)\alpha_\ell\right)^{q-1}$, we obtain

$$\sum_{j=1}^k \alpha_j^q \prod_{\ell=j+1}^k (1-\alpha_\ell b) \leq \alpha_k^{q-1} \sum_{j=1}^k \alpha_j \prod_{\ell=j+1}^k \left(1+\frac{b}{2(q-1)}\alpha_\ell\right)^{q-1}\left(1-\frac{b}{q-1}\alpha_\ell\right)^{q-1}$$

$$\leq \alpha_k^{q-1} \sum_{j=1}^k \alpha_j \prod_{\ell=j+1}^k \left(1-\frac{b}{2(q-1)}\alpha_\ell\right)^{q-1}$$

$$\leq \alpha_k^{q-1} \sum_{j=1}^k \alpha_j \prod_{\ell=j+1}^k \left(1-\frac{b}{2(q-1)}\alpha_\ell\right) \leq \frac{2(q-1)}{b}\alpha_k^{q-1} \,,$$

and the statement follows. $\qquad\square$

We conclude with a technical statement on the coefficients $\{\alpha_j\}_{j\in\mathbb{N}}$ under A3.

**Lemma 29.** *Let $b > 0$, and let $\alpha_\ell = c_0/\{\ell + k_0\}^\gamma$, $\gamma \in (1/2; 1)$, such that $c_0 \leq 1/b$. Then, for any $k_0$ satisfying*

$$k_0^{1-\gamma} \geq \frac{2}{c_0 b}\left(\log\{\frac{c_0}{b(2\gamma - 1)2^\gamma}\} + \gamma \log\{k_0\}\right),\tag{71}$$

*any $s \in (1; 2]$, $q \in (0; 1]$, and $k \in \mathbb{N}$, it holds that*

$$\sum_{j=1}^{k} \alpha_j^s\Big(\sum_{\ell=j+1}^{k} \alpha_\ell^2\Big)^q \prod_{\ell=j+1}^{k}(1 - \alpha_\ell b) \leq C(s, q, b)\alpha_k^{s+q-1}/b^{1+q},\tag{72}$$

*where $C(s, q, b) = 12 \cdot 3^{\gamma(s+q-1)}\left(\frac{4(s-1)\gamma}{bc_0}\right)^{2\gamma(s-1)/(1-\gamma)}$.*

*Proof.* We denote $t = \lfloor(k + k_0)/2\rfloor$ and split the sum in (72) into two parts:

$$\sum_{j=1}^{k}\Big(\frac{c_0}{(j + k_0)^\gamma}\Big)^s\Big(\sum_{\ell=j+1}^{k}\Big(\frac{c_0}{(l + k_0)^\gamma}\Big)^2\Big)^q \prod_{\ell=j+1}^{k}(1 - \frac{c_0}{(l + k_0)^\gamma}b)$$

$$= \sum_{j=1+k_0}^{k+k_0}\Big(\frac{c_0}{j^\gamma}\Big)^s\Big(\sum_{\ell=j+1}^{k+k_0}\Big(\frac{c_0}{l^\gamma}\Big)^2\Big)^q \prod_{\ell=j+1}^{k+k_0}(1 - \frac{c_0}{l^\gamma}b)$$

$$\leq \underbrace{\sum_{j=1}^{t}\Big(\frac{c_0}{j^\gamma}\Big)^s\Big(\sum_{\ell=j+1}^{k+k_0}\Big(\frac{c_0}{l^\gamma}\Big)^2\Big)^q \prod_{\ell=j+1}^{k+k_0}(1 - \frac{c_0}{l^\gamma}b)}_{T_1}$$

$$+ \underbrace{\sum_{j=t+1}^{k+k_0}\Big(\frac{c_0}{j^\gamma}\Big)^s\Big(\sum_{\ell=j+1}^{k+k_0}\Big(\frac{c_0}{l^\gamma}\Big)^2\Big)^q \prod_{\ell=j+1}^{k+k_0}(1 - \frac{c_0}{l^\gamma}b)}_{T_2}.$$

For the term $T_2$ we notice that $\frac{c_0}{j^\gamma} \leq 2^\gamma\alpha_k$ for $j \in \{t + 1, \ldots, k + k_0\}$, hence, we can upper bound $T_2$ as follows:

$$T_2 \leq 2^{\gamma(s+2q)}\sum_{j=t+1}^{k+k_0}\alpha_k^{s+2q}(k + k_0 - j)^q \exp\{-b(k + k_0 - j)\alpha_k\}$$

$$\leq 2^{\gamma(s+2q)+1}\alpha_k^{s+2q}\int_0^{+\infty} x^q \exp\{-b\alpha_k x\}\mathrm{d}x$$

$$\leq 2^{\gamma(s+2q)+1}\alpha_k^{s+q-1}/b^{1+q}\int_0^{+\infty} u^q \exp\{-u\}\mathrm{d}u$$

$$\leq 2^{\gamma(s+2q)+1}\alpha_k^{s+q-1}/b^{1+q},$$

where we have used that $\Gamma(q + 1) \leq 1$ for $q \in (0, 1)$. Now it remains to provide an upper bound for $T_1$. Since $k_0$ satisfies (71), we get that, for $j \leq t$,

$$\Big(\sum_{\ell=j+1}^{k+k_0}\Big(\frac{c_0}{\ell^\gamma}\Big)^2\Big)^q \prod_{\ell=j+1}^{k+k_0}(1 - \frac{c_0}{\ell^\gamma}b) \leq (\frac{c_0}{bt^\gamma})^q \prod_{\ell=j+1}^{t}(1 - \frac{c_0}{\ell^\gamma}b).\tag{73}$$

Then, applying (73), we get that

$$T_1 \leq (\frac{c_0}{bt^\gamma})^q\sum_{j=1}^{t}\Big(\frac{c_0}{j^\gamma}\Big)^s \prod_{\ell=j+1}^{t}(1 - \frac{c_0}{\ell^\gamma}b) \overset{(a)}{\leq} (4/b^{q+1})\left(\frac{4(s - 1)\gamma}{bc_0}\right)^{2\gamma(s-1)/(1-\gamma)}\Big(\frac{c_0}{t^\gamma}\Big)^{q+s-1}$$

$$\leq (4/b^{1+q})\left(\frac{4(s - 1)\gamma}{bc_0}\right)^{2\gamma(s-1)/(1-\gamma)} 3^{\gamma(s+q-1)}\alpha_k^{q+s-1},$$

where in (a) we have additionally used Lemma 30 and in (b) we used $\frac{c_0}{t^\gamma} \leq 3^\gamma \alpha_k$. It remains to check the relation (73), that is, it is enough to obtain an upper bound

$$\Big(\sum_{\ell=2}^{k+k_0} \frac{c_0^2}{\ell^{2\gamma}}\Big)^q \prod_{\ell=t+1}^{k+k_0} (1 - \frac{c_0}{\ell^\gamma} b) \leq (\frac{c_0}{bt^\gamma})^q \ . \tag{74}$$

Since

$$\sum_{\ell=2}^{k+k_0} \frac{c_0^2}{\ell^{2\gamma}} \leq c_0^2 \int_2^{k+k_0} \frac{\mathrm{d}x}{x^{2\gamma}} \leq \frac{c_0^2}{2\gamma - 1} \ .$$

Hence, (74) will follow from

$$\frac{c_0^2}{2\gamma - 1} \exp\Big\{ -\frac{(k + k_0 - t)\alpha_k b}{q} \Big\} \leq \frac{c_0}{t^\gamma b} \ ,$$

which is guaranteed by relations (71). $\qquad\square$

**Lemma 30.** *Let $b > 0$, and let $\alpha_k = c_0/(k)^\gamma$, $\gamma \in (1/2; 1)$, such that $c_0 b \leq 1/2$. Then for any $q \in [1; 2]$, it holds that*

$$\sum_{j=1}^k \alpha_j^q \prod_{\ell=j+1}^k (1 - \alpha_\ell b) \leq C_q \alpha_k^{q-1} \ ,$$

*where $C_q = \frac{4}{b}\Big(\frac{4(q-1)\gamma}{bc_0}\Big)^{2\gamma(q-1)/(1-\gamma)}$.*

*Proof.* Note that

$$\sum_{j=1}^k \alpha_j^q \prod_{\ell=j+1}^k (1 - \alpha_\ell b) = \alpha_k^{q-1} \sum_{j=1}^k \alpha_j \prod_{\ell=j+1}^k \Big(\frac{\alpha_{\ell-1}}{\alpha_\ell}\Big)^{q-1} (1 - \alpha_\ell b)$$

$$\leq \alpha_k^{q-1} \sum_{j=1}^k \alpha_j \prod_{\ell=j+1}^k \Big(1 + \frac{1}{\ell - 1}\Big)^{\gamma(q-1)} (1 - \alpha_\ell b)$$

$$\leq \alpha_k^{q-1} \sum_{j=1}^k \alpha_j \exp\Big\{ \sum_{\ell=j+1}^k \Big\{ \frac{\gamma(q-1)}{\ell - 1} - \frac{bc_0}{\ell^\gamma} \Big\} \Big\} \ .$$

Define $\ell_0 = \lceil \big(\frac{4(q-1)\gamma}{bc_0}\big)^{1/(1-\gamma)} \rceil \geq 2$, then for any $l > l_0$ we have $\frac{\gamma(q-1)}{\ell-1} - \frac{bc_0}{\ell^\gamma} \leq -\frac{bc_0}{2\ell^\gamma}$. Hence, we get

$$\sum_{j=1}^k \alpha_j^q \prod_{\ell=j+1}^k (1 - \alpha_\ell b) \leq \alpha_k^{q-1} \sum_{j=1}^k \alpha_j \exp\Big\{ \sum_{\ell=j+1}^k \Big\{ -\frac{bc_0}{2\ell^\gamma} \Big\} \Big\} \exp\Big\{ \sum_{\ell=2}^{\ell_0} \Big\{ \frac{\gamma(q-1)}{\ell - 1} \Big\} \Big\} \ .$$

Therefore, Lemma 25 together with the elementary inequality $\mathrm{e}^{-x} \leq 1 - x/2$ for $x \in (0; 1/2)$, implies that

$$\sum_{j=1}^k \alpha_j^q \prod_{\ell=j+1}^k (1 - \alpha_\ell b) \leq \alpha_k^{q-1} \sum_{j=1}^k \alpha_j \prod_{\ell=j+1}^k (1 - \frac{b}{4}\alpha_\ell) \exp\Big\{ \gamma(q-1)(\log(\ell_0 - 1) + 1) \Big\}$$

$$\leq \alpha_k^{q-1} \frac{4}{b}\Big(\frac{4(q-1)\gamma}{bc_0}\Big)^{2\gamma(q-1)/(1-\gamma)} \ .$$

$\qquad\square$

**Lemma 31.** *For any $A > 0$, any $0 \leq i \leq n - 1$ and any $\gamma \in (1/2, 1)$ it holds*

$$\sum_{j=i}^{n-1} \exp\Big\{ -A(j^{1-\gamma} - i^{1-\gamma}) \Big\} \leq \begin{cases} 1 + \exp\{\frac{1}{1-\gamma}\} \frac{1}{A^{1/(1-\gamma)}(1-\gamma)} \Gamma(\frac{1}{1-\gamma}) \ , & \text{if } Ai^{1-\gamma} \leq \frac{1}{1-\gamma} \text{ and } i \geq 1 \ ; \\ 1 + \frac{1}{A(1-\gamma)^2} i^\gamma \ , & \text{if } Ai^{1-\gamma} > \frac{1}{1-\gamma} \text{ and } i \geq 1 \ ; \\ 1 + \frac{1}{A^{1/(1-\gamma)}(1-\gamma)} \Gamma(\frac{1}{1-\gamma}) \ , & \text{if } i = 0 \ . \end{cases}$$

*Proof.* The proof is given in [70, Lemma 16]. $\qquad\square$

# H   Applications to the TD learning algorithm

Recall that the TD learning algorithm within the framework of linear stochastic approximation (LSA) can be written as

$$\theta_k = \theta_{k-1} - \alpha_k(\mathbf{A}_k\theta_{k-1} - \mathbf{b}_k) , \qquad (75)$$

where the matrices $\mathbf{A}_k$ and vectors $\mathbf{b}_k$ are defined by

$$\mathbf{A}_k = \varphi(s_k)\{\varphi(s_k) - \lambda\varphi(s_{k+1})\}^\top ,$$
$$\mathbf{b}_k = \varphi(s_k)\mathcal{R}(s_k) .$$

Our primary objective is to estimate the agent's *value function*, defined as

$$V(s) = \mathbb{E}[\textstyle\sum_{k=0}^\infty \lambda^k \mathcal{R}(S_k) \,|\, S_0 = s] ,$$

where $S_{k+1} \sim \mathrm{P}(\cdot \,|\, S_k)$ for all $k \in \mathbb{N}$. In this context, the TD learning updates (75) correspond to approximating solutions to the deterministic linear system $\bar{\mathbf{A}}\theta^\star = \bar{\mathbf{b}}$ (see [77]), where the matrix $\bar{\mathbf{A}}$ and right-hand side $\bar{\mathbf{b}}$ are given by

$$\bar{\mathbf{A}} = \mathbb{E}_{s\sim\mu, s'\sim\mathrm{P}(\cdot|s)} \left[ \varphi(s)\{\varphi(s) - \lambda\varphi(s')\}^\top \right]$$
$$\bar{\mathbf{b}} = \mathbb{E}_{s\sim\mu} \left[ \varphi(s)\mathcal{R}(s) \right] .$$

The rest of the proof of Proposition 5 reduces to checking the properties of $\bar{\mathbf{A}}$ and direct verification of conditions of Proposition 2, which is done by the lines of [67, Proposition 2]

**Numerical experiments.**   We consider the simple instance of the Garnet problem [4, 32]. This problem is characterized by the number of states $N_s$, number of actions $a$, and branching factor $b$. Here $b$ corresponds to the number of states $s'$, that can be reached when starting from a given state-action pair $(s, a)$. The reward $r(s, a) \in [0, 1]$ is a deterministic function. We set the hyperparameter values $N_s = 6$, $a = 2$, $b = 3$, feature dimension $d = 2$, and discount factor $\lambda = 0.8$. We aim to evaluate the value function of policy $\pi(\cdot|s)$, which is given, for any $a \in \mathcal{A} = \{1, 2\}$, by the expression

$$\pi(a|s) = \frac{U_a^{(s)}}{\sum_{i=1}^{|\mathcal{A}|} U_i^{(s)}} ,$$

where the $U_i^{(s)}$ are i.i.d. observations with $\mathcal{U}[0, 1]$. In this case we can suppose that $\mathcal{R}(s) = \mathbb{E}_{a\sim\pi(a|s)}r(s, a)$ is a random variable which depends on state $s$ and some independent random variable. We consider the problem of policy evaluation in this MDP using the TD learning algorithm with randomly generated feature mapping, that is, we generate the matrix

$$\Phi \in \mathbb{R}^{N_s \times d}$$

with i.i.d. $\mathcal{N}(0, 1)$ entries, and then take $\phi(s)$, $s \in \{1, \ldots, |S|\}$, to its $s$-th row, normalized by its euclidean norm: $\phi(s) = \Phi_s/\|\Phi_s\|$. We run the procedure (1) with the learning rates $\alpha_k = c_0/(k_0 + k)^\gamma$ with $\gamma = 2/3$ with appropriately chosen $c_0$ and $k_0$. We generate random vector $u$ form unitary sphere, and compute coverage probabilities for $u^\top\theta^\star$ for confidence levels $\{0.8, 0.9, 0.95\}$. The detailed setting of the experiments follows [29]. Results are given in Table 1 and illustrates the consistency of multiplier subsample bootstrap procedure applied on the Garnet problem.

Table 1: Coverage probabilities of OBM estimation for the empirical distribution.

| $n \mid b_n$ | **0.95** | | **0.9** | | **0.8** | | stddev $\times 10^3$ |
|---|---|---|---|---|---|---|---|
| | $\hat{\sigma}_\theta^2(u)$ | $\sigma^2(u)$ | $\hat{\sigma}_\theta^2(u)$ | $\sigma^2(u)$ | $\hat{\sigma}_\theta^2(u)$ | $\sigma^2(u)$ | |
| 20480 \| 250 | 0.873 | 0.881 | 0.773 | 0.805 | 0.641 | 0.662 | 10.89 |
| 204800 \| 1200 | 0.935 | 0.945 | 0.880 | 0.892 | 0.768 | 0.784 | 3.49 |
| 1024000 \| 3600 | 0.942 | 0.948 | 0.887 | 0.897 | 0.769 | 0.788 | 1.56 |

Code to reproduce experiments is given in `https://github.com/svsamsonov/markov_lsa_normal_approximation`. Our experiments were conducted on a single Intel Xeon Gold 6248R CPU (48 cores, 3.0–4.0 GHz), 768 GB RAM, and 240 GB SSD storage, without GPU accelerators.

