# OpenReview forum: "Statistical inference for Linear Stochastic Approximation with Markovian Noise"
_NeurIPS.cc/2025/Conference — NeurIPS 2025 poster_

### Official Review · Reviewer_sZzp · 2025-06-24

**Clarity:** 4
**Significance:** 3
**Originality:** 3
**Rating:** 4
**Confidence:** 3

**Summary:**

This paper derives Berry–Esseen bounds for the averaged iterates of linear stochastic approximation under ergodic Markovian noise. Moreover, the authors obtain non-asymptotic guarantees for the multiplier subsample bootstrap estimator, including explicit error rates for estimating the asymptotic covariance matrix and for the coverage probabilities of $\theta^*$.

**Questions:**

I hope the following questions would help the authors improve the paper.

Q1. This paper appears to extend the results of [67] by relaxing the i.i.d. noise assumption to ergodic Markovian noise. It would help readers if the authors could clarify (i) the main technical challenges in moving from the i.i.d. to the Markovian setting and (ii) the key distinctions between this work and [67].

Q2. On line 222 the authors state that [81] derives a Berry–Esseen bound for TD learning under Markov noise with the similar rate $\tilde{\mathcal{O}}(1/n^{1/4})$. Could you elaborate on why the techniques in [81] do not carry over directly to the general linear stochastic approximation with Markovian noise?

Q3. In the introduction the authors note two standard approaches for constructing confidence intervals based on asymptotic normality—plug-in and batch-mean estimators. In the i.i.d. setting, [80] employs a plug-in estimator and attains an \$\mathcal{O}(n^{-1/3})\$ approximation rate for \$\theta^\ast\$ in TD learning, outperforming the \$\mathcal{O}(n^{-1/4})\$ rate typically achieved by batch-mean estimators for general linear SA. In this paper, the authors adopt a batch-mean method and derives an \$\mathcal{O}(n^{-1/10})\$ rate for linear SA with Markovian data. Could you clarify (i) why plug-in methods were not considered for the Markovian setting and (ii) whether the \$\mathcal{O}(n^{-1/10})\$ rate is believed to be optimal?

Q4. In this paper, the results are derived for projected iterates. Could the authors clarify whether analogous guarantees would hold for the unprojected algorithm, and what specific technical obstacles would need to be overcome to establish such an extension?

Q5. The manuscript introduces many definitions—see, for instance, Lines 182 and 266—without offering the intuition behind them. As a result, readers may struggle to recall what each term represents and to follow the authors’ line of reasoning.

Q6. Throughout the paper, the time index alternates between $k$ and $n$. For clarity and consistency, it would be preferable to choose a single symbol and use it uniformly.

**Ethical Concerns:**

["NO or VERY MINOR ethics concerns only"]

**Final Justification:**

After reading the rebuttal, I am convinced that extending the analysis from the i.i.d. setting to the Markovian setting introduces substantial technical challenges, which this paper addresses very effectively. However, because the convergence rate of \$\mathcal{O}(n^{-1/10})\$ is not optimal and the results are limited to projected algorithms—as noted in my Q3 and Q4—I am raising my score only from 3 to 4.

**Limitations:**

Yes.

**Paper Formatting Concerns:**

I didn’t notice any major formatting issues in this paper.

**Quality:**

4

**Strengths And Weaknesses:**

Strengths:

S1. To the best of my knowledge, the Berry–Esseen bounds the authors derive—expressed in Kolmogorov distance—for averaged linear stochastic-approximation iterates under ergodic Markovian noise are novel. The analysis also makes particularly effective use of the Poisson decomposition for Markov chains.

S2. This is a theoretically solid work. The proof seems correct, although I did not check line by line.

Weaknesses:

Please see questions below.

---

> ### Author Rebuttal · Authors · 2025-07-31
>
> We thank the reviewer for their feedback. Below we address some points raised as weaknesses and provide responses to the listed questions.
>
> Q1 ***(i) the main technical challenges in moving from the i.i.d. to the Markovian setting and (ii) the key distinctions between this work and [67].*** The detailed answer to this question is given in our reply to Q3 to the reviewer ***QGKs***. We briefly repeat key points here. The main  technical difference is that the tools used in [67], notably the Berry–Esseen-type bounds for nonlinear statistics (see [69]), require independence and are not applicable in our setting. Both our work and \[67] rely on a decomposition of the form $\sqrt{n}(\bar\theta_n - \theta^*) = W + D$, where $W$ is a linear statistic and $D$ is a nonlinear remainder term. In the i.i.d. setting considered in [67], it is sufficient to control the second moment of $D$. However, in the Markovian setting, we must instead control higher-order moments, typically up to the $p$-th moment, where $p \simeq \log n$.
> Moreover, [67] investigates the validity of the multiplier bootstrap for constructing confidence intervals. This technique, however, is not applicable under Markovian noise due to sample dependence. In fact, [41, Proposition 1] shows that the limiting distributions of the multiplier bootstrap SGD and the original SGD can differ, implying that this bootstrap method cannot be  used for constracting confidence intervals in the Markovian case.
> To address this issue, we propose an alternative procedure—multiplier subsample bootstrap—which requires a more technically involved analysis.
> In summary, while both works share the goal of providing non-asymptotic bounds for LSA algorithm, the Markovian setting introduces significant technical challenges.
>
> Q2 ***Could you elaborate on why the techniques in [81] do not carry over directly to the general linear stochastic approximation with Markovian noise?***
> We highlight that the results of [81] hold only for specific problem, namely for TD leaning, at the same time we consider general LSA problem. It is not clear if the obtained results can be directly translated to the setting of general LSA, since they rely on a particular properties of the design matrix, which are specific to the TD learning. Moreover, the authors of [81] do not provide an algorithm for inference and constructing the confidence intervals, contrary to our submission.
>
> Also, we would like to mention that the results in [81] appeared recently - just about three months before the NeurIPS deadline. It is in fact close to be a concurrent submission according to NeurIPS guidelines, and, to our knowledge, it is not yet published. At the same time, the key element of their analysis involves a non-asymptotic CLT for vector-valued martingales, which is a complicated result, which requires to be carefully checked. In fact, a version of such a martingale CLT has been provided in a recent paper [59], but later the authors of [81] found a missing term in their analysis. This comment is not to criticize the paper [59], but rather to acknowledge that the field of martingale CLTs relies of technically invlved arguments, where it is extremely easy to make a mistake.
>
> Our analysis was carried out independently of [81] and provides a result based on peer-reviewed and well-established results [11] and [27]. We do not claim that the results in [81] are incorrect; rather, due to the recency of the work and the technical complexity of the methods involved, we were not in a position to fully verify their applicability in our setting within the submission timeframe.
>
> Furthermore, applying the results from [81] would require the quadratic characteristic of the  martingale to be almost surely constant, which is not the case of our submission. Extending their approach to our setting would therefore require additional nontrivial modifications, which are similar to those which are incorporated in our paper when building upon the Bolthausen's result [11]. We hope this clarifies our choice of probabilistic tools used in our results.
>
> Q3 ***Could you clarify (i) why plug-in methods were not considered for the Markovian setting & clarifying rates of convergence*** To the best of our knowledge, indeed, there are not result of this type in the literature in Markovain setting. Moreover, a complete non-asymptotic analysis of plug-in methods even for general SGD (not LSA) and linear stochastic approximation (LSA) with Markovian noise does not exist. First of all, this is due to the fact that one needs to estimate the noise covariance matrix $\Sigma_{\varepsilon}$ defined in our submission, defined in lines 147-148. This step is computationally intensive, and ,ight require additional nested procedure to estimate covariances.
>
> Moreover, existing results for plug-in or batch-mean methods for SA in the literature typically provide only in-expectation bounds of the form $\mathbb{E}[\|\hat \Sigma_n - \Sigma_\infty\|],$ which are not sufficient to provide on Gaussian approximation and do not take into account the convergence rate in CLT. In particular, in-expectation bounds do not guarantee accurate distributional approximations between
> $N(0, \hat \Sigma_n)$ and
> $N(0, \Sigma_{\infty})$ on events with high probability. Such guarantees are essential for establishing a Gaussian comparison bound. In contrast, our work provides a high-probability bound on
> $|u^\top \hat \Sigma_n u - u^\top \Sigma_\infty u|$, with rate $\mathcal{O}(n^{-1/8})$. Notably, a similar rate appears in [63], which studies batch mean estimators for stochastic gradient descent (SGD) with Markovian noise, though only in expectation.
>
> ***(ii) whether the ${\mathcal{O}}(n^{-1/10})$ rate is believed to be optimal?*** This is a very intersring question, please see our response to the Weakness 3 raised by reviewer ***nTAM***.
>
> Q4 ***Could the authors clarify whether analogous guarantees would hold for the unprojected algorithm, and what specific technical obstacles would need to be overcome to establish such an extension?*** Extending our results to the unprojected algorithm is an important direction, but it introduces several technical challenges. Specifically, two key technical obstacles arise:
>
> a. Establishing a Gaussian approximation for sums of martingale-difference sequences;
>
> b. Obtaining concentration bounds for the overlapping batch mean estimator of the covariance matrix.
>
> The step (a) is especially involved, and associated issues are discussed above in response to ***Reviewer 1bdC*** (Weakness 1). The step (b) is also challenging, and there is no ready-to-use which can be applied at this point. Both steps involve significant additional complexity, and we leave this extension for future work.
>
> Q5  ***The manuscript introduces many definitions—see, for instance, Lines 182 and 266—without offering the intuition behind them. As a result, readers may struggle to recall what each term represents and to follow the authors’ line of reasoning*** Thank you for the suggestion. These notations were introduced primarily to simplify the exposition and streamline the technical arguments. In the final version of the paper, we will do our best to improve clarity by adding more intuition and brief reminders when key definitions are reused. At the same time, our results rely on sophisticated tools for the proofs of the main results, such as martingale concentration inequalities, and given this it is not possible to provide fully elementary exposition.
>
> Q6 ***Throughout the paper, the time index alternates between $k$ and $n$. For clarity and consistency, it would be preferable to choose a single symbol and use it uniformly.*** Thank you for the comment. We use the indices $k$ and $n$ to distinguish between their different roles: $n$ denotes the total number of iterations of the LSA algorithm, while $k$ refers to a generic intermediate iteration index. We use index $k$ to present results which hold true for any intermediate iteration index $k$. We will clarify this in the notation section to avoid confusion.
>
> ***References***
>
> [11]E. Bolthausen. Exact Convergence Rates in Some Martingale Central Limit Theorems. The Annals of Probability, 10(3):672 – 688, 1982.
>
> [27]Xiequan Fan. Exact rates of convergence in some martingale central limit theorems. Journal of Mathematical Analysis and Applications, 469(2):1028–1044, 2019.
>
> [41] Ruiqi Liu, Xi Chen, and Zuofeng Shang. Statistical inference with Stochastic Gradient Methods under  $\phi$-mixing Data. arXiv preprint arXiv:2302.12717, 2023.
>
> [59] Pratik Ramprasad, Yuantong Li, Zhuoran Yang, Zhaoran Wang, Will Wei Sun, and Guang Cheng. Online bootstrap inference for policy evaluation in reinforcement learning. J. Amer. Statist. Assoc., 118(544):2901–2914, 2023.
>
> [63] Abhishek Roy and Krishnakumar Balasubramanian. Online covariance estimation for stochastic gradient descent under Markovian sampling. arXiv preprint arXiv:2308.01481, 2023.
>
> [67] Sergey Samsonov, Eric Moulines, Qi-Man Shao, Zhuo-Song Zhang, and Alexey Naumov. Gaussian Approximation and Multiplier Bootstrap for Polyak-Ruppert Averaged Linear Stochastic Approximation with Applications to TD Learning. In Advances in Neural Information Processing Systems, volume 37, pages 12408–12460. Curran Associates, Inc., 2024.
>
> [69] Qi-Man Shao and Zhuo-Song Zhang. Berry–Esseen bounds for multivariate nonlinear statistics with applications to M-estimators and stochastic gradient descent algorithms. Bernoulli, 28(3):1548–1576, 2022.
>
> [81] Weichen Wu, Yuting Wei, and Alessandro Rinaldo. Uncertainty quantification for Markov chains with application to temporal difference learning. arXiv preprint arXiv:2502.13822, 2025.

---

> > ### Comment · Reviewer_sZzp · 2025-08-05
> >
> > The authors have done a good job in responding to my previous comments. I find their responses clear and detailed. I have no further questions and will raise the score to 4.

---

### Official Review · Reviewer_QGKs · 2025-06-24

**Clarity:** 4
**Significance:** 3
**Originality:** 3
**Rating:** 5
**Confidence:** 3

**Summary:**

The paper establishes finite-sample inference for Polyak–Ruppert averaged linear stochastic approximation under uniformly geometrically ergodic Markov noise by (1) proving a non-asymptotic Berry–Esseen bound of order $O(n^{-1/4})$ for one-dimensional projections of $\sqrt{n}(\bar\theta_n - \theta^*)$, and (2) developing a multiplier block bootstrap with $O(n^{-1/10})$ finite-sample coverage error (up to logs). The analysis, based on Poisson-equation decompositions, martingale Berry–Esseen techniques, and overlapping batch means for variance estimation, is illustrated in the context of temporal-difference learning with linear function approximation.

**Questions:**

The questions follow the order of the weaknesses listed above.

1. Could the authors clarify how Markovian noise actually arises in real-world SA applications? In particular, could they provide concrete instances to substantiate the statement “many practical SA applications involve dependent noise, often forming a Markov chain” (line 21)? Addressing this would help improve the **clarity** and **significance** scores.
2. Could the authors provide empirical mean-squared error decay curves (for example, log-log plots of
   $\mathbb{E}\|\bar\theta_n - \theta^*\|^2$ versus $n$) on a nontrivial benchmark and compare the observed slope to the $O(n^{-1/4})$ rate predicted by Theorem 1? This would directly assess whether the theory captures real-world behavior. Addressing this would help improve the **significance** score.

3. Although the related works section (Section 2) discusses many results under i.i.d. noise, could the authors comment on the key differences between the i.i.d. and Markovian noise settings, particularly regarding the theoretical tools and techniques required for analysis? Addressing this would help improve the **significance** score.

**Ethical Concerns:**

["NO or VERY MINOR ethics concerns only"]

**Final Justification:**

I am satisfied with the authors’ clarifications and am happy to maintain my original score.

**Limitations:**

Yes, the authors acknowledge potential limitations.

**Quality:**

4

**Strengths And Weaknesses:**

# Strengths:

* The paper is very well organized, with thorough background and motivation, a comprehensive related works section, clear motivations before each theorem, and insightful discussions after each result, all of which greatly aid reader comprehension.
* Provides the first non-asymptotic coverage error bounds for bootstrap methods under dependent (Markovian) SA iterates, which is valuable for online learning and RL applications.
* The theoretical analysis is rigorous and solid, combining Poisson equation techniques for Markov chains, martingale Berry–Esseen bounds, and batch means/multiplier block bootstrap analysis under dependence, with sound logic and careful handling of technical conditions.

# Weaknesses:

Please note that the following weaknesses do not undermine the core contributions, but addressing them would make the paper even stronger:

* Since the paper’s central focus is on analysis under Markovian noise, the unsubstantiated claim that “many practical SA applications involve dependent noise, often forming a Markov chain” (line 21) leaves readers without the concrete motivation needed to appreciate why handling this specific dependency structure is both relevant and necessary.
* The experimental validation in Appendix G is limited to synthetic and relatively small-scale examples, so it is unclear how the proposed theoretical bounds and bootstrap intervals hold up on larger or real-world problems.
* The related works section (Section 2) reviews many results under i.i.d. noise but does not explain how the theoretical tools and proof techniques must change in the Markovian setting, making it hard to see what new challenges the paper overcomes.

---

> ### Author Rebuttal · Authors · 2025-07-31
>
> We thank the reviewer for their  positive  feedback and are happy to provide further details in response to the questions they raised.
>
> Q1  ***Could the authors clarify how Markovian noise actually arises in real-world SA applications?***
>
> This setting naturally arises for Reinforcement Learning algorithms, since it is typical that the sequence of agent's states form a Markov chain in this case. Below, we provide concrete examples from RL domain where Markovian noise naturally appears in LSA:
>
> ***Temporal-Difference (TD) Learning with linear function approximation***
> TD(0) and its variants are widely used for policy evaluation. These methods update parameters using observed transitions $(s_n, r_n, s_{n+1})$, which are generated by following a fixed policy in a Markov Decision Process (MDP). One can find details about this setting, for example, in the recent paper [68].
>
> **Gradient Temporal Difference Learning (GTD, GTD2):** GTD algorithms, designed for stable off-policy learning, also rely on linear updates based on sequences of transitions $(s_n, a_n, r_n, s_{n+1})$ generated by a behavior policy in an MDP.
> In all these examples, the data used for parameter updates come from trajectories in an MDP and thus form a Markov chain. Since the update rules are linear in the parameters and the stochastic noise arises from a Markovian process, these algorithms are naturally modeled as instances of LSA with Markovian noise.
>
> Q2 ***Could the authors provide empirical mean-squared error decay curves (for example, log-log plots of $\mathbb{E}|\bar\theta_{n} - \theta^\star|^2$ versus $n$) on a nontrivial benchmark and compare the observed slope to the $O(n^{-1/4})$  rate predicted by Theorem 1?***
>
> We would like to clarify, that the main focus of our work is not on to bound the mean-squared error $\mathbb{E}\|\bar\theta_n - \theta^\star\|^2$, but to establish a rate of convergence to the Gaussian limit in the Kolmogorov distance:
> $$
> \sup_{x \in \mathbb{R}} \| \mathbb{P} \left( \sqrt{n} u^\top (\bar\theta_n - \theta^\star) \leq x \right) - \Phi(x) \|,
> $$
> where $\Phi(x)$ denotes the standard normal cumulative distribution function. This object is significantly more complicated, as it captures distributional convergence rather than second-moment decay. Moreover, empirical estimation of the Kolmogorov distance on a nontrivial benchmark is highly computationally intensive. It requires accurately estimating the distribution of $\sqrt{n} u^\top (\bar\theta_n - \theta^*)$, which in turn necessitates a large number of independent trajectories to obtain empirical cumulative distribution function. At the same time, we acknowledge the importance of lower bounds for the rates obtained in Theorem 1.
>
> Q3  ***Although the related works section (Section 2) discusses many results under i.i.d. noise, could the authors comment on the key differences between the i.i.d. and Markovian noise settings, particularly regarding the theoretical tools and techniques required for analysis?*** The works [67] and [80] consider the LSA problem with i.i.d noise. The main  technical difference is that the tools used in [67] and [80], notably the Berry–Esseen type bounds for nonlinear statistics by Shao and Zhang (see [69]), are applicable only for i.i.d. random variables. Our work, as well as [67] and [80], relies on a decomposition of the form
> $$
> \sqrt{n}(\bar\theta\_k - \theta^\star) = W + D
> $$
> where $W$ is a linear statistic and $D$ is a nonlinear remainder term. In the i.i.d. setting considered in [67] and [80], it is sufficient to control the second moment of $D$. However, in the Markovian setting, we must instead control higher-order moments, typically up to the $p$-th moment, where $p \simeq \log n$. In addition, we would like to highlight that the results of [80] hold only for specific problem, namely for TD leaning, at the same time we consider general LSA problem.
>
> Another issue with this decomposition arises already from the central limit theorem for the linear statistic $W$. While in the i.i.d. setting, there are considerable number of papers which study this problem and the dependence of convergence upon the problem dimension $d$ (e.g. [6]), setting of Markov chains and martingales if far less studied. Typically published in peer-reviews journals results only cover the 1-dimensional setting, or are not explicit with respect to mixing time of the underlying Markov chain.
>
> One more methodological difference is that [67] investigates the validity of the multiplier bootstrap procedure for constructing confidence intervals for general LSA problem with i.i.d. noise, which is different from the procedure outlined in our paper. This is due to the fact, that this procedure is not applicable under Markovian noise due to sample dependence. In fact, [41, Proposition 1] shows that the limiting distributions of the multiplier bootstrap SGD and the original SGD can differ, implying that this bootstrap method is not even asymptotically valid. To address this issue, we propose an alternative procedure— multiplier subsample bootstrap —which requires completely different analysis.
>
> ***References***
>
> [6] V. Bentkus. On the dependence of the Berry–Esseen bound on dimension. Journal of Statistical Planning and Inference, 113(2):385–402, 2003.
>
> [41] Ruiqi Liu, Xi Chen, and Zuofeng Shang. Statistical inference with Stochastic Gradient Methods under -mixing Data. arXiv preprint arXiv:2302.12717, 2023.
>
> [67] Sergey Samsonov, Eric Moulines, Qi-Man Shao, Zhuo-Song Zhang, and Alexey Naumov. Gaus- sian Approximation and Multiplier Bootstrap for Polyak-Ruppert Averaged Linear Stochastic Approximation with Applications to TD Learning. In Advances in Neural Information Process- ing Systems, volume 37, pages 12408–12460. Curran Associates, Inc., 2024.
>
> [68] Sergey Samsonov, Daniil Tiapkin, Alexey Naumov, and Eric Moulines. Improved High Probability Bounds for the Temporal Difference Learning Algorithm via Exponential Stability. In Shipra Agrawal and Aaron Roth, editors, Proceedings of Thirty Seventh Conference on Learning Theory, volume 247 of Proceedings of Machine Learning Research, pages 4511–4547. PMLR, 30 Jun–03 Jul 2024.
>
> [69] Qi-Man Shao and Zhuo-Song Zhang. Berry–Esseen bounds for multivariate nonlinear statistics with applications to M-estimators and stochastic gradient descent algorithms. Bernoulli, 28(3):1548–1576, 2022.
>
> [80] Weichen Wu, Gen Li, Yuting Wei, and Alessandro Rinaldo. Statistical Inference for Temporal Difference Learning with Linear Function Approximation. arXiv preprint arXiv:2410.16106, 2024.

---

> ### Comment · Reviewer_QGKs · 2025-08-05
>
> Thank you for your thoughtful and thorough rebuttal. Your responses clarified the key points I raised, especially regarding real-world relevance of Markovian noise in applications like TD learning and GTD algorithms. I also appreciate the explanation of your focus on convergence in distribution rather than mean-squared error. The additional discussion on how results differ between the i.i.d. and Markovian settings was helpful and addressed my earlier concerns. I'm satisfied with your clarifications and am happy to maintain my original score.

---

### Official Review · Reviewer_nTAM · 2025-06-30

**Clarity:** 2
**Significance:** 2
**Originality:** 3
**Rating:** 4
**Confidence:** 4

**Summary:**

The paper studies the statistical inference properties of the projected Polyak-Ruppert averaged iterates in Linear Stochastic Approximation (LSA) algorithms under a Markovian noise model. The authors establish non-asymptotic bounds for the convergence of the averaged iterates to a Gaussian distribution in the Kolmogorov distance, with a rate of $O(n^{-1/4})$. They also validate the multiplier subsample bootstrap (MSB) procedure and demonstrate its consistency, allowing the probabilities for the actual parameter $\theta^\star$ to be approximated at a rate of $O(n^{-1/10})$. Furthermore, the paper recovers an $O(n^{-1/8})$ rate (up to logarithmic factors) for estimating the asymptotic variance of the iterates. The theoretical results are finally applied to the TD learning setting, which becomes a specific instance of LSA under appropriate assumptions.

**Questions:**

The strengths and weaknesses section already provides some suggestions and questions. Here are additional questions:
- Remark 1 states that "Corollary 1 predicts the optimal error of ..." Could the authors clarify what "optimal" refers to?
- On page 6, it is claimed that the essential part of the analysis "can be generalized to the $d$-dimensional case." Given this, what are the main challenges in extending the full result to multi-dimensional settings?
- Are there technical tools used in the paper that are novel and worth highlighting? Or is the technical contribution mostly about applying existing results and their variants with a new proof strategy?

**Ethical Concerns:**

["NO or VERY MINOR ethics concerns only"]

**Final Justification:**

A score of 4 shows my support for accepting the paper. The main points I raised in the initial review and during the rebuttal periods are to understand if the paper's results and methodologies extend to scopes that would interest a much broader community. In particular, I seek confirmation from the authors on the extensions to:
- Higher dimensions
- Settings with (much) fewer assumptions
- More practically relevant bounds, unless the existing rates are not far from optimal.

Though unable to make such extensions, the authors were able to present challenges and difficulties that one might face on the path. Hence, I kept my recommendation.

**Limitations:**

This is a theoretical paper and does not directly present societal concerns. As mentioned earlier, it is recommended to include a dedicated section that discusses the limitations of the assumptions (for practical applications) and compares them with relevant work.

**Paper Formatting Concerns:**

The paper is clearly written and technically well-formatted. There are no major concerns regarding the formatting.

**Quality:**

3

**Strengths And Weaknesses:**

Strengths:
- The paper provides clear and rigorous theoretical contributions with non-asymptotic bounds for both approximation and bootstrap procedures, an area that has received less attention for Markovian settings compared to IID.
- The Kolmogorov convergence rate of $O(n^{-1/4})$ seems to be novel and tighter than previous results, such as the Wasserstein bound of $O(n^{-1/6})$ in [72], which is $O(n^{-1/12})$ under Kolmogorov distance.
- The authors carefully utilize the Poisson equation/decomposition and martingale CLTs for their analysis. The derivation is detailed and appears to be technically interesting.
- The block bootstrap method is rigorously justified and analyzed non-asymptotically, which is novel within the LSA setting with Markov noise.

Weaknesses:
- The analysis is limited to one-dimensional projections of the estimation error. While the authors acknowledge this, extending the results to multivariate bounds remains an open question. Additionally, the results of [74] apply in a $d$-dimensional setting.
- Assumptions such as those related to step size and tuning may be restrictive and impact the significance of the results. It is advisable to include a dedicated section discussing these limitations and comparing them with relevant work.
- The bootstrap bound of $O(n^{-1/10})$, although novel, does not provide guarantees regarding its tightness or optimality. Could the authors offer insights into the best possible upper bound that might be achieved?
- The results are primarily theoretical. Although the TD learning application is mentioned, there are few empirical plots or data tables throughout the entire submission (including supplementary materials) to demonstrate empirical convergence rates and algorithm performance. Some technical notations and assumptions may also hinder accessibility for non-expert audiences.

---

> ### Author Rebuttal · Authors · 2025-07-31
>
> We thank the reviewer for their positive feedback and are happy to provide further details in response to their concerns. Below we provide our responses to weaknesses and questions (Q) raised by the reviewer.
>
> Weakness 1 and Q2 ***The analysis is limited to one-dimensional projections of the estimation error. While the authors acknowledge this, extending the results to multivariate bounds remains an open question. &  On page 6, it is claimed that the essential part of the analysis "can be generalized to the $d$-dimensional case." Given this, what are the main challenges in extending the full result to multi-dimensional settings?***
>
> Extending our results to the unprojected algorithm is an important direction, but it introduces several technical challenges. Specifically, two key technical obstacles arise:
>
> a. Establishing a Gaussian approximation for sums of martingale-difference sequences;
>
> b. Obtaining concentration bounds for the overlapping batch mean estimator of the covariance matrix.
>
> The step (a) is especially involved, and associated issues are discussed above in response to ***Reviewer 1bdC*** (Weakness 1). The step (b) is also challenging, and there is no ready-to-use which can be applied at this point. Both steps involve significant additional complexity, and we leave this extension for future work. At the same time, in our proofs  we rely on the $p$-th moment bounds on the error terms $\mathbb{E}[\|\theta_k - \theta^{\star}\|]$, which we derive in the current submission, and which are instrumental both for the martingale CLT (when dealing with non-constant quadratic characteristic), and for the subsequent analysis of the bootstrap procedure.
>
> Weakness 2 ***Assumptions such as those related to step size and tuning may be restrictive and impact the significance of the results. It is advisable to include a dedicated section discussing these limitations and comparing them with relevant work.*** We thank the reviewer for this comment and would include a more detailed discussion in the revised paper. We would like to emphasize that the step-size schedule $\alpha_k = c_0 / (k_0 + k)^\gamma$ is a standard and widely adopted choice in the stochastic approximation literature (see, e.g., \[25, 70, 80]). The values of $c_0$ and $k_0$ indeed depend on problem-specific characteristics, and in our work, we provide explicit constraints on these constants to guarantee the validity of the non-asymptotic CLT bound.  The dependence of the parameters upon the total number of iterations $n$ is discussed in more details  in our answer to the reviewer ***3fTA***, see the corresponding Weakness 1. The key conclusion from our bounds is that the parameter $k_0$ needs to be chosen sufficiently large.
>
> Weakness 3 ***The bootstrap bound of $O(n^{-1/10})$ , although novel, does not provide guarantees regarding its tightness or optimality. Could the authors offer insights into the best possible upper bound that might be achieved?***
>
> To the best of our knowledge, our work provides the first non-asymptotic rate for constructing confidence intervals in the linear stochastic approximation (LSA) setting with Markovian noise. As such, there are currently no known lower bounds for this problem that would allow us to assess the optimality of the $O(n^{-1/10})$ rate. We believe that this rate may not be optimal, and can be improved.  In particular, we highlight that the available rates for estimating the covariance matrix $\Sigma_{\infty}$ in SA with Markov noise are also pessimistic. As stated in the paper, recent work [63] obtained rate $O(n^{-1/8})$ for such a problem, albeit also without lower bounds. At the same time, [63] derive only in-expectation bounds for estimating the error $\mathbb{E}[\|\hat{\Sigma}_n - \Sigma_{\infty}\|] for the suggested estimator $\hat{\Sigma}_n$ for the batch-mean procedure. It is possible, that the same rate can be obtained in our inference problem, yet this would require a different proof strategy. We see this as an interesting direction for future research.
>
> Weakness 4 ***The results are primarily theoretical. Although the TD learning application is mentioned, there are few empirical plots or data tables throughout the entire submission (including supplementary materials) to demonstrate empirical convergence rates and algorithm performance.***
>
> The primary aim of the paper is theoretical, and it is true that some additional experimental settings can be added. While it is complicated (if not impossible) to trace numerically the convergence rate $n^{-1/10}$ for our inference procedure, in the revised version of the paper we can incorporate some additional examples for TD learning on more challenging environments, such as the ones from the recent paper [59]. Unfortunately, given the timeline, we are not able to provide these additional results before the rebuttal period ends.
>
>  ***Some technical notations and assumptions may also hinder accessibility for non-expert audiences.***
>
> Thank you for this comment, in the revised version of the paper we will do our best to simplify exposition, at the same time, our results demand sophisticated tools for the proofs of the main results, such as martingale concentration inequalities, and given this it is not possible to provide fully elementary exposition.
>
> Q1  ***Remark 1 states that "Corollary 1 predicts the optimal error of ..." Could the authors clarify what "optimal" refers to? and the best possible upper bound that might be achieved?***
>
> We thank the reviewer for pointing out the ambiguity in the use of the term "optimal error" in Remark 1. In this context, by “optimal” we means simply the best possible bound, which can be inferred from Corollary 1 after optimizing its right-hand side over $\gamma$. Through this optimization, we obtain a convergence rate $\mathcal{O}(n^{-1/4})$ in the Kolmogorov distance. We do not claim that this rate is optimal, as no lower bounds are currently known for the general LSA problem with Markovian noise. However, we would like to note that the $\mathcal{O}(n^{-1/4})$ rate we obtain in the general LSA setting matches the best-known bound established in [81] for particular instance of the LSA problem, namely, the temporal-difference (TD) learning with Markovian noise. We will revise the statement of Remark 1 to make this point clearer.
>
> Q3 ***Are there technical tools used in the paper that are novel and worth highlighting? Or is the technical contribution mostly about applying existing results and their variants with a new proof strategy?***
>
> Our work introduces two technical contributions that we believe are of independent interest:
>
> a. We show how the bootstrap procedure for LSA iterates with Markovian noise, where the bootstrap iterates themselves do not form a Markov chain, can be effectively reduced to a bootstrap procedure for linear statistics of Markov chains. In particular, we relate the bootstrap variance estimation in LSA to the problem of estimating the asymptotic variance of a certain linear statistic of a Markov chain, which generates noise in the LSA procedure. This reduction allows us to leverage concentration tools developed for the setting of linear statistics.
>
> b. Our proof of Theorem 1 relies on a new version of martingale Central Limit Theorem (CLT) applicable for martingale, which quadratic characteristic is not almost surely constant. This result of Lemma 21, provided in Appendix, provides such a result based on previously established theorems due to Bolthausen [11] and Fan [27]. This result provides a fully non-asymptotic bound in the Kolmogorov distance for scalar-valued martingales with explicit dependence upon $p$ and other problem characteristics.
>
>
> ***References****
>
> [11] E. Bolthausen. Exact Convergence Rates in Some Martingale Central Limit Theorems. The Annals of Probability, 10(3):672 – 688, 1982.
>
> [25] Alain Durmus, Eric Moulines, Alexey Naumov, Sergey Samsonov, and Hoi-To Wai. On the stability of random matrix product with markovian noise: Application to linear stochastic approximation and td learning. In Mikhail Belkin and Samory Kpotufe, editors, Proceedings of Thirty Fourth Conference on Learning Theory, volume 134 of Proceedings of Machine Learning Research, pages 1711–1752. PMLR, 15–19 Aug 2021.
>
> [27] Xiequan Fan. Exact rates of convergence in some martingale central limit theorems. Journal of Mathematical Analysis and Applications, 469(2):1028–1044, 2019.
>
> [59] Pratik Ramprasad, Yuantong Li, Zhuoran Yang, Zhaoran Wang, Will Wei Sun, and Guang Cheng. Online bootstrap inference for policy evaluation in reinforcement learning. J. Amer. Statist. Assoc., 118(544):2901–2914, 2023.
>
> [63] Abhishek Roy and Krishnakumar Balasubramanian. Online covariance estimation for stochastic gradient descent under Markovian sampling. arXiv preprint arXiv:2308.01481, 2023.
>
> [70] Marina Sheshukova, Sergey Samsonov, Denis Belomestny, Eric Moulines, Qi-Man Shao, Zhuo-Song Zhang, and Alexey Naumov. Gaussian approximation and multiplier bootstrap for stochastic gradient descent. arXiv preprint arXiv:2502.06719, 2025.
>
> [80] Weichen Wu, Gen Li, Yuting Wei, and Alessandro Rinaldo. Statistical Inference for Temporal Difference Learning with Linear Function Approximation. arXiv preprint arXiv:2410.16106, 2024.
>
> [81] Weichen Wu, Yuting Wei, and Alessandro Rinaldo. Uncertainty quantification for Markov chains with application to temporal difference learning. arXiv preprint arXiv:2502.13822, 2025.

---

> > ### Comment · Reviewer_nTAM · 2025-08-07
> >
> > Thank you for the detailed response. Please see my comments below.
> >
> > ### 1. The analysis is limited to one-dimensional projections
> > > Extending our results to the unprojected algorithm is an important direction, but it introduces several technical challenges.
> > > Both steps involve significant additional complexity, and we leave this extension for future work.
> >
> > Thanks for clarifying. I acknowledge that extension to higher dimensions could be challenging. For example, in statistical learning of distributions, 1D closed intervals could often be efficiently partitioned to facilitate estimation, while such partitions are frequently infeasible or not efficiently computable.
> >
> > On the other hand, real-world learning tasks are usually in higher dimensions. So I believe it's fair to say that the "one-dimensional" analysis limited the generality and significance of the paper, hence it is a weakness. That being said, please incorporate some of the rebuttal comments in the paper to help future researchers understand the path ahead and challenges they might face.
> >
> > ###  2. Assumptions such as those related to step size and tuning may be restrictive and impact the significance of the results.
> >
> > To clarify, "step size and tuning" are not the only assumptions made in the paper. I understand that meaningful results may not be attainable without assumptions. However, the main point of raising this weakness is that, given the number of assumptions, the authors should carefully explain how restrictive each is. It is possible that a single assumption, either non-realistic or uncommon, could greatly facilitate the proofs while (implicitly) weakening the value of the results.
> >
> > ### 3. The bootstrap bound of $O(n^{-1/10})$, although novel, does not provide guarantees regarding its tightness or optimality.
> > > There are currently no known lower bounds for this problem that would allow us to assess the optimality
> >
> > Similar to above, I believe it's fair to view this as a "weakness" of the paper, regardless of whether some related works have established optimality. In the field of statistical learning, it is desired to establish the optimality of the (bounds on) sample complexity, expected error, learning rate, etc. So I would expect the authors to at least show some form of lower bounds, which don't have to match the upper bound(s).
> >
> > ### 4. The results are primarily theoretical ... there are few empirical plots ... to demonstrate empirical convergence rates and algorithm performance.
> > > The primary aim of the paper is theoretical.
> > > Unfortunately, given the timeline, we are not able to provide these additional results before the rebuttal period ends.
> >
> > I acknowledge that the paper's contributions are mainly on the theory side. I had considered this weakness as a minor one.
> >
> > I'd also like to thank the authors for the detailed response to my questions. If appropriate, please find ways to incorporate these comments/statements into the paper or the supplementary.

---

> > > ### Author Response · Authors · 2025-08-08
> > >
> > > We would like to thank the reviewer for their comment and will do our best to incorporate the discussion on the suggested weaknesses. At the same time, we would like to further tighten some of the points raised in our rebuttal and in the reviewer’s comment.
> > >
> > > 1. ***The analysis is limited to one-dimensional projections***
> > >
> > > We agree, that the high-dimensional results are important and a natural step for further investigations. At the same time, while it is true that high-dimensional CLT allows control of convergence over a richer collection of sets, it is known that the dependence on dimension in this case is rather pessimistic (see the mentioned reference [Shao and Zhang, 2022], which considers the i.i.d. noise setting). Thus, the question here is more about the collection of sets over which we want to control the convergence rate. The results are known to be fundamentally different for convex sets (Shao and Zhang or Wu, Wei, and Rinaldo) or rectangles (e.g. [[Chernozhukov et al, 2017]). Thus, apart from the technical difficulties, there is an additional question of how to choose the appropriate collection of sets for a particular application. In this sense, projections might be even more useful; see, e.g., Remark 1 in [Samsonov et al., 2024].
> > >
> > > 2. ***Assumptions***
> > >
> > > We agree that the paper could benefit from further discussion of the assumptions. At the same time, those not related to the choice of step size are rather standard (Hurwitzness of the system matrix and uniform geometric ergodicity of the noise chain) and are imposed in a number of theoretical papers concerning the linear stochastic approximation. We will add a more detailed bibliography to the revised version of the paper.
> > >
> > > 3. ***Lower bounds***
> > >
> > > We agree with the reviewer that the question of lower bounds is extremely important. Unfortunately, to the best of our knowledge, there are no available lower bounds even for the simpler (yet related) problem of online estimation of the asymptotic covariance matrix for SA algorithms; see, e.g., [Chen et al., 2023]. Thus, we believe it is more appropriate to consider lower bounds as a challenging direction for a separate submission, rather than a weakness of the current paper.
> > >
> > > ***References***
> > >
> > > [Shao and Zhang, 2022] Qi-Man Shao and Zhuo-Song Zhang. Berry–Esseen bounds for multivariate nonlinear statistics with applications to M-estimators and stochastic gradient descent algorithms. Bernoulli, 28(3):1548–1576, 2022.
> > >
> > > [Chernozhukov et al, 2017] Victor Chernozhukov, Denis Chetverikov, and Kengo Kato. Central limit theorems and bootstrap in high dimensions. Annals of Probability, 45(4):2309–2352, 2017.
> > >
> > > [Samsonov et al, 2024] Sergey Samsonov, Eric Moulines, Qi-Man Shao, Zhuo-Song Zhang, and Alexey Naumov. Gaussian Approximation and Multiplier Bootstrap for Polyak-Ruppert Averaged Linear Stochastic Approximation with Applications to TD Learning. In Advances in Neural Information Processing Systems, volume 37, pages 12408–12460. Curran Associates, Inc., 2024.
> > >
> > > [Chen et al, 2023] Xi Chen, Wanrong Zhu, and Wei Biao Wu. Online Covariance Matrix Estimation in Stochastic Gradient Descent. Journal of the American Statistical Association, 118(541):393–404, 2023.

---

### Official Review · Reviewer_3fTA · 2025-07-01

**Clarity:** 3
**Significance:** 2
**Originality:** 3
**Rating:** 4
**Confidence:** 3

**Summary:**

The paper studies the problem of Linear Stochastic Approximation (LSA) where
the noise is Markovian. In this setting, the authors focus on establishing
non-asymptotic Berry-Essen bounds for estimators that employ Polyak-Ruppert
averaging. Over the course of the paper, the authors establish $O(n^{-1/4})$
rates of convergence to the limiting Gaussian distribution w.r.t. the
Kolmogorov distance.

Moreover, the paper studies the block bootstrap for estimating the variance
of their estimator $\overline{\theta}_n$ of interest, showing it is consistent under the
assumption of Markovian noise. Under mild assumptions, the paper recommends
a bootstrap growth rate $b_n$ and stepsize growth rate $\alpha_k$ which
yield a $O(n^{-1/8})$ convergence rate w.r.t. the Kolmogorov distance for
any projected version of $\sqrt{n}(\overline{\theta}_n - \theta^*)$.

The paper concludes by offering a concrete application of their theory for
the temporal-difference learning algorithm applied to Markov decision
processes.

**Questions:**

[Q1] Most of the random quantites studied in the paper are scalars, as they
usually are the dot product between the estimator and some random vector
$u$. Can the results here be generalized to the multivariate domain,
eschewing the need to choose these fixed vectors $u$ in advance?

**Ethical Concerns:**

["NO or VERY MINOR ethics concerns only"]

**Final Justification:**

I think the work seems well argued and written. That being said, b/c of the similar work that has already been published, I have slightly reduced my score in significance and overall. I am still in favor of submission.

**Limitations:**

No, the authors should explain whether there is any negative societal impact from doing this theoretical work.

**Quality:**

3

**Strengths And Weaknesses:**

Strengths:

Originality:

[+] The approach taken by the paper appears to borrow from many different
previous works, e.g., by using the Poisson decomposition of Markov chains to
help derive its non-asymptotic bounds. While the results accord with other
works, the proof strategy appears to be unique.

[+] There is accompanying theory on the block bootstrap that can be used
to estimate the non-asymptotic variance of the desired parameter
$\theta$. This proof appears to be novel and the first to do so in the LSA
setting with Markov noise.

Quality:

[+] The paper is very thorough and takes great care in defining all
notation, salient assumptions and related work. While I have not checked
everything in the appendix, the paper seems sensible and does not contain
any obvious flaws.

Clarity:

[+] The paper is very clear and detail oriented. Most proofs are pushed to
the appendix so that the core ideas can be discussed in the main portion of
the text. While there is quite a bit of notation, the general ideas in the
paper are well conveyed.

Significance:

[+] The core significance of this paper is mostly
theoretical. Its convergence rate of asymptotic variance matches other works
but is more general as these techniques apply to Markovian noise rather than
simply independent noise samples.

[+] The paper specifies a specific learning rate and bootstrap growth rate
in the case of the bootstrap. Thus while theoretical, this provides some
practical input for those practicing with this technique.

Weaknesses:

Significance:

[-] As discussed in the paper, in order to obtain these non-asymptotic error
bounds, one must know the final value $n$ in advance. While this is not a
huge deal, it does impose some structure on the algorithms for which these
bounds can be derived.

[-] The work does not improve upon previous bounds; that being said, it does
generalize the previously shown bounds to a non-asymptotic regime with
Markovian noise, which does appear novel.

---

> ### Author Rebuttal · Authors · 2025-07-31
>
> We thank the reviewer for their positive feedback and are happy to provide further details in response to their concerns.
>
> Weakness 1 ***As discussed in the paper, in order to obtain these non-asymptotic error bounds, one must know the final value $n$ in advance. While this is not a huge deal, it does impose some structure on the algorithms for which these bounds can be derived.***  Our proof technique requires to control the $p$-th moment of the last iterate, with $p \approx \log n$, in order to derive Theorem 1 about Gaussian approximation. This introduces the dependence of the step size upon the number of iterations $n$. It is known that, at least in case of constant step-size $\alpha$, bounds on the $p$-th moments of the error require to choose $\alpha \approx 1/p$, as shown in [23]. Consequently, the total number of iterations $n$ must be known in advance to set the appropriate step-size. It is not immediately clear, if the same conclusion applies to the products of random matrices with decreasing step sizes, at the same time, there are some evidences, which support this claim. In particular, consider theorem 3.1 in the related paper [80]. The results are derived for the particular setting of TD(0) learning, yet they require to scale $c_0$ in the step size with the confidence parameter $\delta$, which needs to be known in advance. Since we set $\delta = 1/n$ in the current paper, this result reveals the same phenomenon about the total number of iterations.
>
> We also note that if an extension of Theorem 2.1 from [69] were available for dependent sequences, such as Markov chain, then one could potentially avoid bounding high-order moments and rely instead on bounds for the second and fourth moments only. Developing such an extension is an interesting direction for future work.
>
> Weakness 2 ***The work does not improve upon previous bounds; that being said, it does generalize the previously shown bounds to a non-asymptotic regime with Markovian noise, which does appear novel.*** We would like to clarify that our work provides a high-probability bound on
> $$
> |u^\top \hat \Sigma_n u - u^\top \Sigma_\infty u|,
> $$
> with rate $\mathcal{O}(n^{-1/8})$, where $\hat \Sigma_n$ is multiplier subsample bootstrap estimator for  $\Sigma_\infty$. It is ture that similar rate was previously obtained in [63] for batch-mean estimators $\tilde \Sigma_n$ in the context of stochastic gradient descent (SGD) with Markovian noise. At the same, we highlight that this result holds only in expectation, that is,
> $$
> \mathbb{E}[\| \tilde \Sigma_n - \Sigma_\infty \|].
> $$
> Our high-probability bounds are strictly stronger than bounds in expectation. Moreover, expectation-based bounds are insufficient for proving Gaussian comparison results, such as approximating the distribution of $\mathcal{N}(0, \hat\Sigma_n)$ by $\mathcal{N}(0, \Sigma_\infty)$ on events with high probability. Moreover, the authors of [63] do not take into account the rates of normal approximation for the original SGD estimate, which makes their final results purely asymptotic. Contrary, our works establishes the non-asymptotic CLT bounds in Kolmogorov distance for general linear stochastic approximation (LSA) with Markovian noise, and relies on them to show the non-asymptotic error of the suggested bootstrap approximation. Thus, it is not entirely correct to compare this rate with the one derived in [63].
>
> Q1. ***Most of the random quantites studied in the paper are scalars, as they usually are the dot product between the estimator and some random vector $u$. Can the results here be generalized to the multivariate domain, eschewing the need to choose these fixed vectors $u$ in advance?***
>
> Extending our results to the unprojected algorithm is an important direction, but it introduces several technical challenges. Specifically, two key technical obstacles arise:
>
> a. Establishing a Gaussian approximation for sums of martingale-difference sequences;
>
> b. Obtaining concentration bounds for the overlapping batch mean estimator of the covariance matrix.
>
> The step (a) is especially involved, and associated issues are discussed above in response to  ***Reviewer 1bdC*** (Weakness 1). The step (b) is also challenging, and there are no ready-to-use results which can be applied at this point. Both steps involve significant additional complexity, and we leave this extension for future work. We will add a more comprehensive overview of these difficulties to the revised version of the paper.
>
> ***References***
>
> [23] Alain Durmus, Eric Moulines, Alexey Naumov, Sergey Samsonov, Kevin Scaman, and Hoi-To Wai. Tight high probability bounds for linear stochastic approximation with fixed stepsize. In M. Ranzato, A. Beygelzimer, K. Nguyen, P. S. Liang, J. W. Vaughan, and Y. Dauphin, editors, Advances in Neural Information Processing Systems, volume 34, pages 30063–30074. Curran Associates, Inc., 2021.
>
> [63] Abhishek Roy and Krishnakumar Balasubramanian. Online covariance estimation for stochastic gradient descent under Markovian sampling. arXiv preprint arXiv:2308.01481, 2023.
>
> [69] Qi-Man Shao and Zhuo-Song Zhang. Berry–Esseen bounds for multivariate nonlinear statistics with applications to M-estimators and stochastic gradient descent algorithms. Bernoulli, 28(3):1548–1576, 2022.
>
> [80] Weichen Wu, Gen Li, Yuting Wei, and Alessandro Rinaldo. Statistical Inference for Temporal Difference Learning with Linear Function Approximation. arXiv preprint arXiv:2410.16106, 2024.

---

> > ### Comment · Reviewer_3fTA · 2025-08-07
> >
> > I thank the authors for their response. The similar work mentioned by Reviewer 1bdC from Wu, Wei, and Rinaldo (2025) does limit the significance of this work and thus I have slightly reduced my score for significance. That being said, I still am in favor of accepting the paper at this conference.

---

> ### Author Response · Authors · 2025-08-08
>
> We would like to thank the reviewer for their comment. While we agree that the paper of Wei, Wu, and Rinaldo [81] is closely related to our submission, there are several key differences. First, the mentioned paper relies on particular properties of the design matrix that are specific to TD learning, and it is unclear whether the result applies to general LSA. Moreover, the authors of [81] do not provide an algorithm for inference or for constructing confidence intervals, in contrast to our submission. We emphasize this result as the second core contribution of our paper.
>
> We hope the reviewer will take this argument into account when evaluating the significance of our submission.

---

### Official Review · Reviewer_1bdC · 2025-07-01

**Clarity:** 4
**Significance:** 3
**Originality:** 3
**Rating:** 4
**Confidence:** 2

**Summary:**

The authors study the problem of deriving central limit theorems and bootstrap consistency results for iterates in linear stochastic approximation (LSA) problems under Markovian noise.   They derive a Berry-Esseen type result for Polyak–Ruppert averaged iterates projected onto the unit sphere with a fixed vector. The convergence rate $n^{-1/4}$ matches the sharpest available bounds in the literature for TD learning.  They also establish rate of convergence for a Gaussian block-multiplier bootstrap procedure.

**Questions:**

- If you generalize your approach to avoid a fixed projection vector $u$, would the rate of convergence remain the same?  I expect that this would be the case, but if the authors could explain the main technical difficulty with considering the $d$-dimensional case, it would be appreciated.
-  Along similar lines, would be possible to consider a growing $d$ regime with appropriate moment bounds?

**Ethical Concerns:**

["NO or VERY MINOR ethics concerns only"]

**Final Justification:**

I still have some concerns related to the fixed projection vector and closeness of the results to the existing work of Wu, Wei, and Rinaldo (2025).  However, the authors have pointed out some differences in the conditions between these results, and have elaborated on the difficulty of generalizing to other distances.  I am willing to slightly increase my score.

**Limitations:**

Yes

**Quality:**

3

**Strengths And Weaknesses:**

Strengths
- The proof of the central limit theorem uses interesting technical tools related to Poisson approximation of Markov chains, and the resulting rate matches other results derived for similar problems.
- The authors derive a result involving the bootstrap for LSA problems, which appears to be the first such result in the literature.
- The bootstrap result involves an interesting approximation argument that handles the fact that the iterates are not themselves a Markov chain.
- The paper is well-written overall and clear.

Weaknesses
- Results involving a fixed projection are weaker than those involving the convex distance in $\mathbb{R}^d$.  The closely related work of Wu, Wei, and Rinaldo (2025) derive such a result, and attain the same rate of convergence.  Moreover, the above result appears to allow $d$ to grow, whereas the arguments in the current paper appear to require $d$ to be fixed.
- The rate of convergence for the bootstrap is slightly disappointing.  The authors give some intuition as to why the rate of convergence is slower. It appears to arise from choosing tuning parameters to balance the rate of convergence for the central limit theorem with variance estimation. It also appears to be related to the choice of target variance.

---

> ### Author Rebuttal · Authors · 2025-07-31
>
> We thank the reviewer for acknowledging the theoretical contribution of the paper and the fact that it is well-written. Below we respond to the weaknesses and questions raised by the reviewer.
>
> Weakness 1. ***Results involving a fixed projection are weaker than those involving the convex distance in $\mathbb{R}^d$. The closely related work of Wu, Wei, and Rinaldo (2025) [81] derive such a result, and attain the same rate of convergence. Moreover, the above result appears to allow $d$ to grow, whereas the arguments in the current paper appear to require $d$ to be fixed.***
>
> The reviewer is correct to mention the paper [81] deals with the convex distance in $\mathbb{R}^d$. At the same time, we highlight that the results of [81] hold only for specific problem, namely for TD leaning, at the same time we consider general LSA problem. It is not clear if the obtained results can be directly translated to the setting of general LSA, since they rely on a particular properties of the design matrix, which are specific to the TD learning. Moreover, the authors of [81] do not provide an algorithm for inference and constructing the confidence intervals, contrary to our submission.
>
> Also, we would like to mention that the results in [81] appeared recently - just about three months before the NeurIPS deadline. It is in fact close to be a concurrent submission according to NeurIPS guidelines, and, to our knowledge, it is not yet published. At the same time, the key element of their analysis involves a non-asymptotic CLT for vector-valued martingales, which is a complicated result, which requires to be carefully checked. In fact, a version of such a martingale CLT has been provided in a recent paper [59], but later the authors of [81] found a missing term in their analysis. This comment is not to criticize the paper [59], but rather to acknowledge that the field of martingale CLTs relies of technically invlved arguments, where it is extremely easy to make a mistake.
>
> Our analysis was carried out independently of [81] and provides a result based on peer-reviewed and well-established results [11] and [27]. We do not claim that the results in [81] are incorrect; rather, due to the recency of the work and the technical complexity of the methods involved, we were not in a position to fully verify their applicability in our setting within the submission timeframe.
>
> Furthermore, applying the results from [81] would require the quadratic characteristic of the  martingale to be almost surely constant, which is not the case of our submission. Extending their approach to our setting would therefore require additional nontrivial modifications, which are similar to those which are incorporated in our paper when building upon the Bolthausen's result [11]. We hope this clarifies our choice of probabilistic tools used in our results.
>
> ***Moreover, the above result appears to allow $d$ to grow, whereas the arguments in the current paper appear to require $d$ to be fixed*** We would appreciate if the reviewer could clarify what it means that the dimension $d$ of the parameter $\theta$ grows during the execution of the algorithm. Typically when considering the LSA problem, the problem dimension is already fixed and it is not clear, how one could interpret adding new lines/columns to the system matrix $\bar{A}$ or adding new features when considering the TD learning algorithm.
>
> Weakness 2. ***The rate of convergence for the bootstrap is slightly disappointing. The authors give some intuition as to why the rate of convergence is slower. It appears to arise from choosing tuning parameters to balance the rate of convergence for the central limit theorem with variance estimation. It also appears to be related to the choice of target variance.***
>
>  We agree that the obtained rate may appear suboptimal. However, to the best of our knowledge, this is the first result providing a fully non-asymptotic analysis for constructing confidence intervals in the linear stochastic approximation (LSA) setting with Markovian noise.  Furthermore, if we focus on the accuracy of the variance approximation—i.e., how well our bootstrap estimator approximates the asymptotic covariance —we achieve a high-probability bound of order $\mathcal{O}(n^{-1/8})$. This matches the rate established in [63] for a different estimator (batch mean method) in the context of stochastic gradient descent (SGD) with Markovian noise, though their result holds only in expectation, that is for $\mathbb{E}[\|\hat \Sigma_n - \Sigma_\infty\|]$. The latter rate is, so the best of our knowledge, the best one which is currently in the literature.
>
> Q1. ***If you generalize your approach to avoid a fixed projection vector $u$, would the rate of convergence remain the same? I expect that this would be the case, but if the authors could explain the main technical difficulty with considering the $d$-dimensional case, it would be appreciated.***
>
> Extending our results to the unprojected algorithm is an important direction, but it introduces several technical challenges. Specifically, two key technical obstacles arise:
>
> a. Establishing a Gaussian approximation for sums of martingale-difference sequences;
>
> b. Obtaining concentration bounds for the overlapping batch mean estimator of the covariance matrix.
>
> The step (a) is especially involved, and associated issues are discussed above in response to Weakness 1. The step (b) is also challenging, and there is no ready-to-use results which can be applied at this point. Both steps involve significant additional complexity, and we leave this extension for future work.
>
> Q2. ***Along similar lines, would be possible to consider a growing $d$ regime with appropriate moment bounds?*** We agree that our theoretical results are obtained under the condition of a fixed parameter dimension $d$. However, we would like to note that it is not clear how to formulate the statement for the LSA algorithm in the case when the dimension $d$ of the parameter $\theta$ changes during its execution. We would be grateful if the reviewer could clarify or specify this comment.
>
> ***References***
>
> [11] E. Bolthausen. Exact Convergence Rates in Some Martingale Central Limit Theorems. The Annals of Probability, 10(3):672 – 688, 1982.
>
> [27] Xiequan Fan. Exact rates of convergence in some martingale central limit theorems. Journal of Mathematical Analysis and Applications, 469(2):1028–1044, 2019.
>
> [59] Pratik Ramprasad, Yuantong Li, Zhuoran Yang, Zhaoran Wang, Will Wei Sun, and Guang Cheng. Online bootstrap inference for policy evaluation in reinforcement learning. J. Amer. Statist. Assoc., 118(544):2901–2914, 2023.
>
> [81] Weichen Wu, Yuting Wei, and Alessandro Rinaldo. Uncertainty quantification for Markov chains with application to temporal difference learning. arXiv preprint arXiv:2502.13822, 2025.

---

> ### Comment · Reviewer_1bdC · 2025-08-06
>
> I would like to thank the authors for their detailed response.  Regarding the authors' question about growing $d$, while it does not make sense to literally increase dimension of the iterate as $n$ grows, one can consider a triangular array setup, in which each row of the triangular array has the same dimension. Such a setup would allow $d$ to appear in the convergence rate, which would provide some guidance as to how large $d$ can be for a Normal/bootstrap approximation to be reasonable.
>
> While I still feel that considering projections is a bit restrictive, in light of the authors' explanation of the technical difficulties associated with generalizing in this direction, I am willing to slightly increase my score.

---

> ### Author Response · Authors · 2025-08-08
>
> We appreciate your comment about the technical difficulties of the $d$-dimensional setting.
>
> The conclusions about the validity of the bootstrap approximation (for a particular collection of sets) can be inferred from the current constants—in fact, the dependence on dimension is completely explicit in our calculations and is milder compared to the $d$-dimensional setting (see, e.g., [Shao and Zhang, 2022], Theorem 3.4). Taking a fixed projection vector $u$ in our setting of Theorem 1 allows, for example, estimation of the rate of normal approximation for the estimate of the value function at a particular state $s$, if we speak about the particular TD learning example. While it is true that high-dimensional CLT allows control of convergence over a richer collection of sets, it is known that the dependence on dimension in this case is rather pessimistic (see the mentioned reference [Shao and Zhang, 2022], which considers the i.i.d. noise setting). Thus, the question here is more about the collection of sets over which we want to control convergence. The results are known to be fundamentally different for convex sets (Shao and Zhang or Wu, Wei, and Rinaldo) or rectangles (e.g. [[Chernozhukov et al, 2017]). Thus, apart from the technical difficulties, there is an additional question of how to choose the appropriate collection of sets for a particular application. In this sense, projections might be even more useful; see, e.g., Remark 1 in [Samsonov et al., 2024].
>
> ***References***
>
> [Shao and Zhang, 2022] Qi-Man Shao and Zhuo-Song Zhang. Berry–Esseen bounds for multivariate nonlinear statistics with applications to M-estimators and stochastic gradient descent algorithms. Bernoulli, 28(3):1548–1576, 2022.
>
> [Chernozhukov et al, 2017] Victor Chernozhukov, Denis Chetverikov, and Kengo Kato. Central limit theorems and bootstrap in high dimensions. Annals of Probability, 45(4):2309–2352, 2017.
>
> [Samsonov et al, 2024] Sergey Samsonov, Eric Moulines, Qi-Man Shao, Zhuo-Song Zhang, and Alexey Naumov. Gaussian Approximation and Multiplier Bootstrap for Polyak-Ruppert Averaged Linear Stochastic Approximation with Applications to TD Learning. In Advances in Neural Information Processing Systems, volume 37, pages 12408–12460. Curran Associates, Inc., 2024.

---

### Decision · Program_Chairs · 2025-09-17

**Decision:**

Accept (poster)

**Comment:**

The paper establishes non-asymptotic Berry–Esseen bounds for Polyak–Ruppert averaged Linear Stochastic Approximation (LSA) under Markovian noise, providing convergence rates to the Gaussian limit. It also validates a multiplier block bootstrap for confidence intervals, giving the first non-asymptotic guarantees for bootstrap inference in this setting and recovering classical variance estimation rates up to logarithmic factors.

The contributions of this paper are interesting and makes a significant contributions to the growing area of SGD inference with Markovian data. Hence, the paper is recommend for acceptance.